# Energy Discrepancies: A Score-Independent Loss for Energy-Based Models

**Tobias Schröder**[1*]  **Zijing Ou**[1†]  **Jen Ning Lim**[2],
**Yingzhen Li**[1]  **Sebastian J. Vollmer**[3]  **Andrew B. Duncan**[1,4]
[1]Imperial College London  [2]University of Warwick  [3]DFKI and RPTU Kaiserslautern
[4]The Alan Turing Institute
{t.schroeder21, z.ou22, yingzhen.li, a.duncan}@imperial.ac.uk
jen-ning.lim@warwick.ac.uk  sebastian.vollmer@dfki.de

## Abstract

Energy-based models are a simple yet powerful class of probabilistic models, but their widespread adoption has been limited by the computational burden of training them. We propose a novel loss function called Energy Discrepancy (ED) which does not rely on the computation of scores or expensive Markov chain Monte Carlo. We show that energy discrepancy approaches the explicit score matching and negative log-likelihood loss under different limits, effectively interpolating between both. Consequently, minimum energy discrepancy estimation overcomes the problem of nearsightedness encountered in score-based estimation methods, while also enjoying theoretical guarantees. Through numerical experiments, we demonstrate that ED learns low-dimensional data distributions faster and more accurately than explicit score matching or contrastive divergence. For high-dimensional image data, we describe how the manifold hypothesis puts limitations on our approach and demonstrate the effectiveness of energy discrepancy by training the energy-based model as a prior of a variational decoder model.

## 1 Introduction

Energy-Based Models (EBMs) are a class of parametric unnormalised probabilistic models of the general form $p_{\text{ebm}} \propto \exp(-U)$ originally inspired by statistical physics. EBMs can be flexibly modelled through a wide range of neural network functions which, in principle, permit the modelling of any positive probability density. Through sampling and inference on the learned energy function, the EBM can then be used as a generative model or in numerous other downstream tasks such as improving robustness in classification or anomaly detection (Grathwohl et al., 2020), simulation-based inference (Glaser et al., 2022), or learning neural set functions (Ou et al., 2022).

Despite their flexibility, EBMs are limited in machine learning applications by the difficulty of their training. The normalisation constant of EBMs, also known as the partition function, is typically intractable making standard techniques such as Maximum Likelihood Estimation (MLE) infeasible. For this reason, EBMs are commonly trained with an approximate maximum likelihood method called Contrastive Divergence (CD) (Hinton, 2002) which approximates the gradient of the log-likelihood using short runs of a Markov chain Monte Carlo (MCMC) method. However, contrastive divergence with short run MCMC leads to malformed estimators of the energy function (Nijkamp et al., 2020a), even for relatively simple restricted Boltzmann-machines (Carreira-Perpiñán & Hinton, 2005). This can, in part, be attributed to the fact that contrastive divergence is not the gradient of any fixed

---

[*]Correspondence to: Tobias Schröder, t.schroeder21@imperial.ac.uk
[†]Code: https://github.com/J-zin/energy-discrepancy

37th Conference on Neural Information Processing Systems (NeurIPS 2023).

objective function (Sutskever & Tieleman, 2010), which severely limits the theoretical understanding of CD and motivated various adjustments of the algorithm (Du et al., 2021; Yair & Michaeli, 2021).

Score-based methods such as Score Matching (SM) (Hyvärinen & Dayan, 2005; Vincent, 2011; Song et al., 2020) and Kernel Stein Discrepancy (KSD) estimation (Liu et al., 2016; Chwialkowski et al., 2016; Gorham & Mackey, 2017; Barp et al., 2019) are a family of competing approaches which offer tractable loss functions and are, by construction, independent of the normalising constant of the distribution. However, such methods suffer from *nearsightedness* as they fail to resolve global features in the distribution without vast amounts of data. In particular, both SM and KSD estimators are unable to capture the mixture weights of two well-separated Gaussians (Song & Ermon, 2019; Zhang et al., 2022; Liu et al., 2023).

We propose a new loss functional for energy-based models called Energy Discrepancy (ED) which compares the data distribution and the energy-based model via two contrasting energy contributions. By definition, energy discrepancy only depends on the energy function and is independent of the scores or MCMC samples from the energy-based model. In our theoretical section, we show that this leads to a loss functional that can be defined on general measure spaces without Euclidean structure and demonstrate its close connection to score matching and maximum likelihood estimation in the Euclidean case. In our practical section, we focus on a simple implementation of energy discrepancy on Euclidean space which requires less evaluations of the energy-based model than the parameter update of contrastive divergence or score-matching. We demonstrate that the Euclidean energy-discrepancy alleviates the problem of nearsightedness of score matching and approximates maximum-likelihood estimation with better theoretical guarantees than contrastive divergence.

On high-dimensional image data, energy-based models face the additional challenge that under the manifold hypothesis (Bengio et al., 2013), the data distribution is not a positive probability density and does, strictly speaking, not permit a representation of the form of an energy-based model. Energy discrepancy is particularly sensitive to singular data supports and requires the transformation of the data distribution to a positive density. We approach this problem by training latent energy-based priors (Pang et al., 2020) which employ a lower-dimensional latent representation in which the data distribution is positive.

Our contributions are the following: 1) We present energy discrepancy, a new estimation loss for the training of energy-based models that can be computed without MCMC or spatial gradients of the energy function. 2) We show that, as a loss function, ED interpolates between the losses of score matching and maximum-likelihood estimation and overcomes the nearsightedness of score-based methods. 3) We introduce a novel variance reduction trick called $w$-stabilisation that drastically reduces the computational cost of approximating energy discrepancy stably.

## 2   Training of Energy-Based Models

In the following, let $p_{\mathrm{data}}(\mathbf{x})$ be an unknown data distribution which we are trying to estimate from independently distributed data $\{\mathbf{x}^i\} \sim p_{\mathrm{data}}$. Energy-based models are parametric distributions of the form

$$p_\theta(\mathbf{x}) \propto \exp(-E_\theta(\mathbf{x}))$$

for which we want to find the scalar energy function $E_\theta$ such that $p_\theta \approx p_{\mathrm{data}}$. Typically, energy-based models are trained with *contrastive divergence* which estimates the gradient of the log-likelihood

$$\nabla_\theta \log p_\theta(\mathbf{x}) = \mathbb{E}_{p_\theta(\mathbf{y})}[\nabla_\theta E_\theta(\mathbf{y})] - \nabla_\theta E_\theta(\mathbf{x}). \tag{1}$$

using Markov chain Monte Carlo methods to approximate the expectation for $p_\theta$. For computational efficiency, the Markov chain is only run for a small number of steps. As a result, contrastive divergence does not learn the maximum-likelihood estimator and can produce malformed estimates of the energy function (Nijkamp et al., 2020a).

Alternatively, the discrepancy between data distribution and energy-based model can be measured by comparing their score functions $\nabla_\mathbf{x} \log p_{\mathrm{data}}$ and $\nabla_\mathbf{x} \log p_\theta$ which, by definition, are independent of the normalising constant. The comparison of the scores is achieved with the *Fisher divergence*. After applying an integration-by-parts and discarding constants with respect to $\theta$, this leads to the *score-matching* loss (Hyvärinen & Dayan, 2005)

$$\mathrm{SM}(p_{\mathrm{data}}, E_\theta) := \mathbb{E}_{p_{\mathrm{data}}(\mathbf{x})}\left[-\Delta_\mathbf{x} E_\theta(\mathbf{x}) + \frac{1}{2}\|\nabla_\mathbf{x} E_\theta(\mathbf{x})\|^2\right]. \tag{2}$$

As this only requires the computation of expectations with respect to $p_{\text{data}}$, the requirement that the data distribution attains a density can be relaxed, yielding a loss function for $p_\theta$ which can be readily approximated. Score-based methods are nearsighted as the score function only contributes local information to the loss. In a mixture of well-separated distributions, the score matching loss decomposes into a sum of objective functions that only see the local mode and are not capable of resolving the weights of the mixture components (Song & Ermon, 2019; Zhang et al., 2022).

## 3 Energy Discrepancies

To illustrate the idea behind the proposed objective function, we start by motivating energy discrepancy from the perspective of explicit score matching. In the following, we will denote the EBM as $p_{\text{ebm}} \propto \exp(-U)$, where $U$ is the energy function that is learned. The nearsightedness of score matching arises due to the presence of large regions of low probability which are separating the modes of the data distribution. To increase the probability mass in these regions, we follow (Zhang et al., 2020) and perturb $p_{\text{data}}$ and $p_{\text{ebm}}$ through a convolution with a Gaussian kernel $\gamma_s(\mathbf{y} - \mathbf{x}) \propto \exp(-\|\mathbf{y} - \mathbf{x}\|^2/2s)$, i.e.

$$p_s(\mathbf{y}) := \int \gamma_s(\mathbf{y} - \mathbf{x}) p_{\text{data}}(\mathbf{x}) d\mathbf{x},$$

$$\exp(-U_s(\mathbf{y})) := \int \gamma_s(\mathbf{y} - \mathbf{x}) \exp(-U(\mathbf{x})) d\mathbf{x}.$$

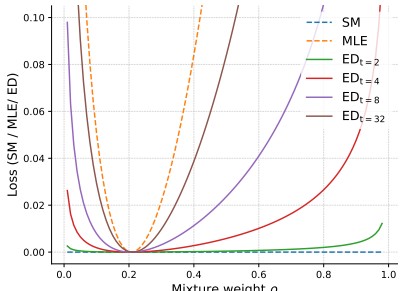

Figure 1: Loss of energy discrepancy, score matching, and maximum likelihood estimation on the task of estimating the weight in a mixture of two Gaussians. For details, see Appendix D.1.

The resulting perturbed divergence $\widetilde{\text{SM}}(p_{\text{data}}, U) := \text{SM}(p_s, U_s)$ retains its unique optimum at $\exp(-U^*) \propto p_{\text{data}}$ (Zhang et al., 2020) but alleviates the nearsightedness as the data distribution is more spread out. The perturbation with $\gamma_s$ simultaneously makes the two distributions more similar which comes at a potential loss of discriminatory power. We mitigate this by integrating the score matching objectives over $s$ over an interval of noise-scales $[0, t]$. It turns out that this integral can be evaluated as the difference of two contrasting energy-contributions:

$$\int_0^t \text{SM}(p_s, U_s) ds = \mathbb{E}_{p_{\text{data}}(\mathbf{x})}[U(\mathbf{x})] - \mathbb{E}_{p_{\text{data}}(\mathbf{x})} \mathbb{E}_{\gamma_t(\mathbf{x}_t - \mathbf{x})}[U_t(\mathbf{x}_t)]. \tag{3}$$

The proof is given in Appendix A.3. The contrasting expression on the right-hand is now *independent* of the score and normalisation of the EBM. We argue that such constructed objective functions are useful losses for energy-based modelling.

### 3.1 A contrastive approach to learning the energy

We lift the idea of learning the energy-based distribution through the contrast of two energy-contributions to a general estimation loss called *Energy Discrepancy*. We will show that energy discrepancy can be defined independently of an underlying perturbative process and is well-posed even on non-Euclidean measure-spaces:

**Definition 1** (Energy Discrepancy). *Let $p_{\text{data}}$ be a positive density on a measure space $(\mathcal{X}, d\mathbf{x})$[1] and let $q(\cdot|\mathbf{x})$ be a conditional probability density. Define the* contrastive potential *induced by $q$ as*

$$U_q(\mathbf{y}) := -\log \int q(\mathbf{y}|\mathbf{x}) \exp(-U(\mathbf{x})) d\mathbf{x}. \tag{4}$$

*We define the* energy discrepancy *between $p_{\text{data}}$ and $U$ induced by $q$ as*

$$\text{ED}_q(p_{\text{data}}, U) := \mathbb{E}_{p_{\text{data}}(\mathbf{x})}[U(\mathbf{x})] - \mathbb{E}_{p_{\text{data}}(\mathbf{x})} \mathbb{E}_{q(\mathbf{y}|\mathbf{x})}[U_q(\mathbf{y})]. \tag{5}$$

In this paper, we shall largely focus on the case where the data is Euclidean, *i.e.* $\mathcal{X} = \mathbb{R}^d$, and the base-distribution $d\mathbf{x}$ is the standard Lebesgue measure. However, this framework also admits $\mathcal{X}$ being discrete spaces like spaces of graphs, or continuous spaces with non-trivial base measures such as manifolds. Specifically, the validity of our approach is characterised by the following non-parametric estimation result:

---

[1]The integrals should be interpreted as Lebesgue integrals, *i.e.,* if $\mathcal{X}$ is discrete, the integral will be a sum.

**Theorem 1.** *Let $p_{\mathrm{data}}$ be a positive probability density on $(\mathcal{X}, \mathrm{d}\mathbf{x})$, and let $q(\cdot|\mathbf{x})$ be a conditional probability density. Under mild assumptions on $q$ and the set of admissible energy functions $U$, energy discrepancy $\mathrm{ED}_q$ is functionally convex in $U$ and has, up to additive constants, a unique global minimiser $U^* = \arg\min \mathrm{ED}_q(p_{\mathrm{data}}, U)$ with $\exp(-U^*) \propto p_{\mathrm{data}}$.*

The assumption on the conditional density $q$ describes that $\mathbf{y} \sim q(\cdot|\mathbf{x})$ involves some loss of information. The assumption that $p_{\mathrm{data}}$ has full support turns out to be critical when scaling energy-discrepancy to high-dimensional data. The proof of Theorem 1 is given in Appendix A.1.

### 3.2 Choices for the conditional distribution $q$

Def. 1 offers a wide range of possible choices for the perturbation distribution $q$. In Appendix D.3 we discuss a possible choice in the discrete space $\{0,1\}^d$. For the remainder of this paper, however, we will focus on Euclidean data $\mathbf{x} \in \mathbb{R}^d$ and hope that the generality of our result inspires future work. Our requirements on $q$ are that simulating from the conditional distribution $\mathbf{y} \sim q(\mathbf{y}|\mathbf{x})$ and computing the convolution that defines $U_q$ are numerically tractable. On continuous spaces, a natural candidate for $q$ is the transition density of a diffusion process which arises as solution to a stochastic differential equations of the form $\mathrm{d}\mathbf{x}_t = \mathbf{a}(\mathbf{x}_t)\mathrm{d}t + \mathrm{d}\mathbf{w}_t$ with drift $\mathbf{a}$ and standard Brownian motion $\mathbf{w}_t$ (see Øksendal, 2003). The conditional density $q_t(\cdot|\mathbf{x})$ represents the probability density of the perturbed particle $\mathbf{x}_t$ that was initialised at $\mathbf{x}_0 = \mathbf{x}$. The resulting transition density then satisfies both of our requirements by employing the Feynman-Kac formula as we line out in Appendix B.2. Although this approach makes the choice of $q$ flexible, the following interpolation result stresses that not much is lost when choosing a Gaussian transition density $\gamma_t \propto \exp(-\|\mathbf{x} - \mathbf{y}\|^2/2t)$:

**Theorem 2.** *Let $q_t$ be the transition density of the diffusion process $\mathrm{d}\mathbf{x}_t = \mathbf{a}(\mathbf{x}_t)\mathrm{d}t + \mathrm{d}\mathbf{w}_t$, let $p_{\mathrm{ebm}} \propto \exp(-U)$ be the energy-based distribution and $p_t$ the data-distribution convolved with $q_t$.*

1. *The energy discrepancy is given by a multi-noise scale score matching loss*

$$\mathrm{ED}_{q_t}(p_{\mathrm{data}}, U) = \int_0^t \mathbb{E}_{p_s(\mathbf{x}_s)}\left[-\Delta U_{q_s}(\mathbf{x}_s) + \frac{1}{2}\|\nabla U_{q_s}(\mathbf{x}_s)\|^2\right]\mathrm{d}s + \mathrm{const}.$$

2. *If $\mathbf{a} = 0$, i.e. $q_t$ is the Gaussian transition density $\gamma_t$, the energy discrepancy converges to the loss of maximum likelihood estimation a linear rate in time*

$$\left|\mathrm{ED}_{\gamma_t}(p_{\mathrm{data}}, U) + \mathbb{E}_{p_{\mathrm{data}}(\mathbf{x})}\left[\log p_{\mathrm{ebm}}(\mathbf{x})\right] - c(t)\right| \leq \frac{1}{2t}\mathbb{W}_2^2(p_{\mathrm{data}}, p_{\mathrm{ebm}})$$

*where $c(t)$ is independent of $U$ and $\mathbb{W}_2$ denotes the Wasserstein distance[2].*

Theorem 2 has two main messages: All diffusion-based energy discrepancies behave like a multi-noise scale score matching loss, which is independent of the drift $\mathbf{a}$. In fact, we show in Appendix A.2 that for a linear drift $\alpha\mathbf{x}_t$, the induced energy discrepancy is always equivalent to the energy discrepancy based on a Gaussian perturbation. Furthermore, estimation with a Gaussian-based energy discrepancy approximates maximum likelihood estimation for large $t$, thus enjoying its attractive asymptotic properties provided $\mathrm{ED}_{\gamma_t}(p_{\mathrm{data}}, U)$ can be approximated with low variance. We demonstrate the result in a mixture model in Figure 1 and Appendix D.1 and give a proof of above theorem in Appendices A.3 and A.4.

**Connection to Contrastive Divergence.** Due to the generality of our result we can also make a direct connection between energy discrepancy and contrastive divergence. To this end, suppose that for $\theta$ fixed, $q$ satisfies the detailed balance relation $q(\mathbf{y}|\mathbf{x})\exp(-E_\theta(\mathbf{x})) = q(\mathbf{x}|\mathbf{y})\exp(-E_\theta(\mathbf{y}))$. In this case, energy discrepancy becomes the loss function

$$\mathrm{ED}_q(p_{\mathrm{data}}, E_\theta) = \mathbb{E}_{p_{\mathrm{data}}(\mathbf{x})}\left[E_\theta(\mathbf{x})\right] - \mathbb{E}_{p_{\mathrm{data}}(\mathbf{x})}\mathbb{E}_{q(\mathbf{y}|\mathbf{x})}\left[E_\theta(\mathbf{y})\right] \tag{6}$$

which, after taking gradients in $\theta$ yields the contrastive divergence update[3]. See Appendix A.5 for details. The non-parametric estimation result from Theorem 1 holds true for the contrastive objective in (6). This means that each step of contrastive divergence optimises an objective function with minimum at $p_{\mathrm{data}} \approx p_\theta$. However, the objective function is adjusted in each step of the algorithm.

---

[2]For a reference on Wasserstein distances, see Peyré & Cuturi (2019)

[3]The implicit dependence of $q$ on $\theta$ is ignored when taking the gradient. Notice that CD is not the gradient of any fixed objective function (Sutskever & Tieleman, 2010).

## 4 Training Energy-Based Models with Energy Discrepancy

In sight of Theorem 2, we will discuss how to approximate ED from samples for Euclidean data $\{\mathbf{x}^i\} \subset \mathbb{R}^d$ and a Gaussian conditional distribution $\gamma_t(\mathbf{y} - \mathbf{x}) \propto \exp(-\|\mathbf{y} - \mathbf{x}\|^2/2t)$. First, the outer expectations on the right-hand of (5) can be computed as plug-in estimators by simulating the Gaussian perturbation $\mathbf{x}_t^i = \mathbf{x}^i + \sqrt{t}\boldsymbol{\xi}$ for $\boldsymbol{\xi} \sim \mathcal{N}(0, \mathbf{I})$ and averaging $\{U(\mathbf{x}^i)\}$ and $\{U_{\gamma_t}(\mathbf{x}_t^i)\}$. The critical step is then finding a low-variance estimate of the contrastive potential $U_{\gamma_t}$ itself.

Due to the symmetry of the Gaussian transition density, we can interpret $\gamma_t$ as the law of a Gaussian random variable with mean $\mathbf{x}_t^i$ and variance $t$, i.e. the conditioned random variable $\mathbf{x}_t^i + \sqrt{t}\boldsymbol{\xi}'|\mathbf{x}_t^i$ for $\boldsymbol{\xi}' \sim \mathcal{N}(0, \mathbf{I})$ has the density $\gamma_t(\mathbf{x}_t^i - \mathbf{x})$. Hence, we can write $U_{\gamma_t}$ as the expectation

$$U_{\gamma_t}(\mathbf{x}_t^i) = -\log \mathbb{E}_{\gamma_1(\boldsymbol{\xi}')}[\exp(-U(\mathbf{x}_t^i + \sqrt{t}\boldsymbol{\xi}'))|\mathbf{x}_t^i]$$

for $i = 1, 2, \ldots, N$. The conditioning expresses that we keep $\mathbf{x}_t^i$ fixed when taking the expectation with respect to $\boldsymbol{\xi}'$. It is then possible to calculate the expectation by sampling $\boldsymbol{\xi}'^{i,j} \sim \mathcal{N}(0, \mathbf{I})$ and calculating the mean as for the outer expectations. However, we find that this approximation is not sufficient as it is biased due to the logarithm and prone to numerical instabilities because of missing bounds on the value of $U_{\gamma_t}(\mathbf{x}_t^i)$. To stabilise training, we augment the Monte Carlo estimate of $U_{\gamma_t}(\mathbf{x}_t^i)$ by an additional term $w/M \exp(-U(\mathbf{x}^i))$ which we call

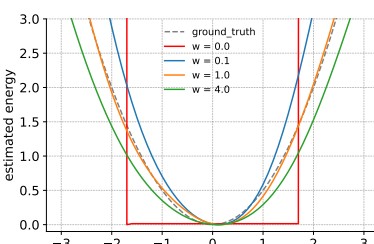

Figure 2: Estimated energy functions for Gaussian data with various choices of $w$. Increasing $w$ leads to flatter energy-landscapes, and training becomes unstable if $w = 0$. For details, see Appendices B.1 and D.2.

$w$-stabilisation. This results in the following approximation of the contrastive potential:

$$U_{\gamma_t}(\mathbf{x}_t^i) \approx -\log\left(\frac{w}{M}\exp(-U(\mathbf{x}^i)) + \frac{1}{M}\sum_{j=1}^{M}\exp(-U(\mathbf{x}_t^i + \sqrt{t}\boldsymbol{\xi}'^{i,j}))\right) \qquad \boldsymbol{\xi}'^{i,j} \sim \mathcal{N}(0, \mathbf{I})$$

The $w$-stabilisation dampens the contributions of contrastive samples $\mathbf{x}_t^i + \sqrt{t}\boldsymbol{\xi}'^{i,j}$ whose energy is higher than the energy of the data point $\mathbf{x}^i$. This provides a deterministic upper bound for the approximate contrastive potential in (4) and reduces the variance of the estimation. We find that the $w$-stabilisation drastically reduces the number of samples $M$ needed for the stable training of deep energy-based models. We illustrate the effect of the stabilisation in Figures 2 and 24 and discuss our reasoning in more details in Appendix B.1. The full loss is now formed for $U := E_\theta$ with $\boldsymbol{\xi}^i \sim \mathcal{N}(0, \mathbf{I})$, $\boldsymbol{\xi}'^{i,j} \sim \mathcal{N}(0, \mathbf{I})$ and tunable hyperparameters $t, M, w$ as

$$\mathcal{L}_{t,M,w}(\theta) := \frac{1}{N}\sum_{i=1}^{N}\log\left(\frac{w}{M} + \frac{1}{M}\sum_{j=1}^{M}\exp(E_\theta(\mathbf{x}^i) - E_\theta(\mathbf{x}^i + \sqrt{t}\boldsymbol{\xi}^i + \sqrt{t}\boldsymbol{\xi}'^{i,j}))\right).$$

The loss is evaluated using the numerically stabilised logsumexp function. The justification of the approximation is given by the following theorem:

**Theorem 3.** *Assume that* $\mathbf{x} \mapsto \exp(-E_\theta(\mathbf{x}))$ *is uniformly bounded. Then, for every* $\varepsilon > 0$ *there exist* $N$ *and* $M(N)$ *such that* $\left|\mathcal{L}_{t,M(N),w}(\theta) - \mathrm{ED}_{\gamma_t}(p_{\mathrm{data}}, E_\theta)\right| < \varepsilon$ *almost surely.*

We give the proof in Appendix B.1. This result forms the basis for proofs of asymptotic consistency of our estimators which we leave for future work. It is noteworthy that an analogous implementation of energy discrepancy for other perturbation kernels and other spaces is possible. For example, we construct a similar loss for discrete spaces using a Bernoulli perturbation in Appendix D.3. Similarly to the Gaussian case, the $w$-stabilisation provides a useful variance reduction for the training with Bernoulli-based energy discrepancy.

### 4.1 Training Energy-Based Models under the Manifold Hypothesis

LeCun (2022) suggests that maximum-likelihood-based training of energy-based models can lead to the formation of undesirable canyon shaped energy-functions. Indeed, energy discrepancy yields an

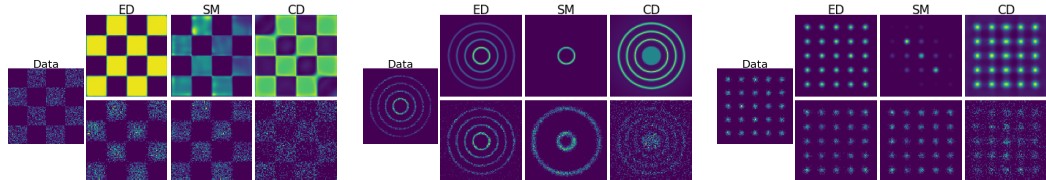

Figure 3: Comparison of energy discrepancy, score matching and contrastive divergence on density estimation. The 1st and 2nd rows are the estimated density and synthesised samples, respectively.

energy with low values on the data-support and rapidly diverging values outside of it when being used on image data, directly. Such learned energies fail to represent the distribution between data-points and are not suitable for inference or image generation. We attribute this phenomenon to the manifold hypothesis, which states that the data concentrates in the vicinity of a low-dimensional manifold (Bengio et al., 2013). In this case, the data distribution is not a positive density and can not be written as an energy-based model as $\log p_{\text{data}}$ is not well-defined. Additionally, Gaussian perturbations of a data point $\tilde{\mathbf{x}} := \mathbf{x} + \boldsymbol{\xi}$ are orthogonal to the data manifold with high-probability and the negative samples in the contrastive term are not informative.

To resolve this problem, we will work with a lower-dimensional latent representation of the data distribution in which positivity can be ensured. We follow Pang et al. (2020) and define a variational decoder network $p_\phi(\mathbf{x}|\mathbf{z})$ and an energy-based tilting of a Gaussian prior $p_\theta(\mathbf{z}) = \exp(-E_\theta(\mathbf{z}))p_0(\mathbf{z})$ on latent space, resulting in the so-called latent energy-based prior models (LEBMs) $p_{\phi,\theta}(\mathbf{x}) \propto \int p_\phi(\mathbf{x}|\mathbf{z})p_\theta(\mathbf{z})\mathrm{d}\mathbf{z}$. To obtain the latent representation of data we sample from the posterior $p_{\phi,\theta}(\mathbf{z}|\mathbf{x}) \propto p_\phi(\mathbf{x}|\mathbf{z}) \exp(-E_\theta(\mathbf{z}))p_0(\mathbf{z})$ using a Langevin sampler. For the training of model, the contrastive divergence algorithm is replaced with energy discrepancy (for details see appendix C). LEBMs provide an interesting benchmark to compare energy discrepancy with contrastive divergence in high dimensions. However, other ways to tackle the manifold problem are elements of ongoing research.

## 5 Empirical Studies

To support our theoretical discussion, we evaluate the performance of energy discrepancy on low-dimensional datasets as well as for the training of latent EBMs on high-dimensional image data sets. In our discussion, we emphasise the comparison with contrastive divergence and score matching as the dominant methodologies in EBM training.

### 5.1 Density Estimation

We first demonstrate the effectiveness of energy discrepancy on several two-dimensional distributions. Figure 3 displays the estimated unnormalised densities as well as samples that were synthesised with Langevin dynamics for energy discrepancy (ED), score matching (SM) and contrastive divergence (CD). More experimental results and details are given in Appendix D.4. Our results confirm the aforementioned nearsightedness of score matching which does not learn the uniform weight distribution of the Gaussian mixture. For CD, it can be seen that CD consistently produces flattened energy landscapes which can be attributed to the short-run MCMC (Nijkamp et al., 2020a, see) not having converged. Consequently, the synthe-

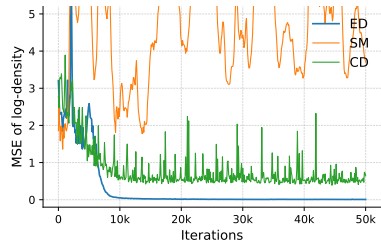

Figure 4: Density estimation accuracy in the 25-Gaussians dataset.

sised samples of energies learned with CD can lie outside of the data-support. In contrast, ED is able to model multi-modal distributions faithfully and learns sharp edges in the data support as in the chessboard data set. We quantify our results in Figure 4 which shows the mean squared error of the estimated log-density of 25-Gaussians over the number of training iterations. The partition function is estimated using importance sampling $\log Z \approx \text{logsumexp}(-E(\mathbf{x}_i) - \log p_{\text{data}}(\mathbf{x}_i)) - \log N$, where $\mathbf{x}_i$ is sampled from the data distribution $p_{\text{data}}(\mathbf{x})$ and $N = 5,000$. It shows that ED outperforms SM and CD with faster convergence, lower mean square error, and better stability.

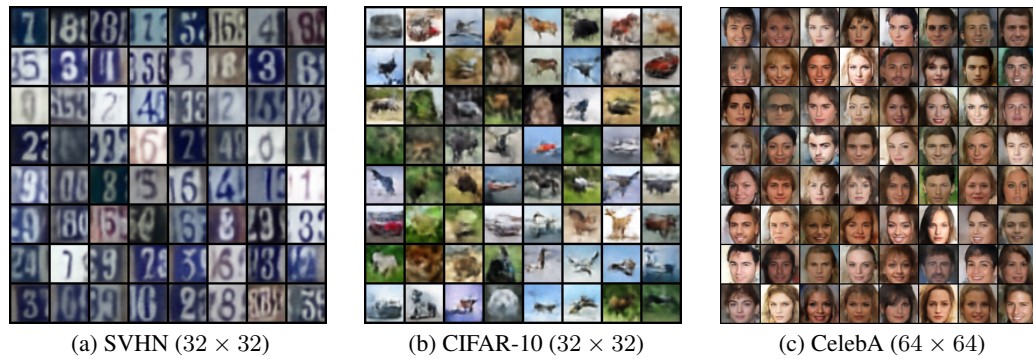

| (a) SVHN ($32 \times 32$) | (b) CIFAR-10 ($32 \times 32$) | (c) CelebA ($64 \times 64$) |

Figure 5: Generated examples by the ED-LEBM trained on SVHN, CIFAR-10, and CelebA.

Table 1: Comparison of MSE(↓) and FID(↓) on the SVHN, CIFAR-10, and CelebA datasets.

| | SVHN | | CIFAR-10 | | CelebA | |
|---|---|---|---|---|---|---|
| | MSE | FID | MSE | FID | MSE | FID |
| VAE (Kingma & Welling, 2014) | 0.019 | 46.78 | 0.057 | 106.37 | 0.021 | 65.75 |
| 2s-VAE (Dai & Wipf, 2018) | 0.019 | 42.81 | 0.056 | 72.90 | 0.021 | 44.40 |
| RAE (Ghosh et al., 2019) | 0.014 | 40.02 | 0.027 | 74.16 | 0.018 | 40.95 |
| SRI (Nijkamp et al., 2020b) | 0.018 | 44.86 | 0.020 | - | - | 61.03 |
| SRI (L=5) (Nijkamp et al., 2020b) | 0.011 | 35.32 | - | - | 0.015 | 47.95 |
| CD-LEBM (Pang et al., 2020) | 0.008 | 29.44 | **0.020** | **70.15** | 0.013 | 37.87 |
| SM-LEBM | 0.010 | 34.44 | 0.026 | 77.82 | 0.014 | 41.21 |
| ED-LEBM (ours) | **0.006** | **28.10** | 0.023 | 73.58 | **0.009** | **36.73** |

## 5.2 Image Modelling

In this experiment, our methods are evaluated by training a latent EBM on three image datasets: SVHN (Netzer et al., 2011), CIFAR-10 (Krizhevsky et al., 2009), and CelebA (Liu et al., 2015). The effectiveness of energy discrepancy is diagnosed through image generation, image reconstruction from their latent representation, and the faithfulness of the learned latent representation. The model architectures, training details, and the choices of hyper-parameters can be found in Appendix C.3.

**Image Generation and Reconstruction.** We benchmark latent EBM priors trained with energy discrepancy (ED-LEBM) and score matching (SM-LEBM) against various baselines for latent variable models which are included in Table 1. Note that the original work on latent EBMs (Pang et al., 2020) uses contrastive divergence (CD-LEBM) (see appendix C for details).

For a well-learned model, the latent EBM should produce realistic samples and faithful reconstructions. The reconstruction error is measured via the mean square error while the image generation is measured with the FID (Heusel et al., 2017) which are both reported in Table 1. We observe that ED can improve the contrastive divergence benchmark on SVHN and CelebA while the results on CIFAR-10 could not be improved. However, we emphasise that ED only requires $M$ (here, $M = 16$) evaluations of the energy function per data point which is significantly less than CD and SM that both require the calculation of a high-dimensional spatial gradient. Besides the quantitative metrics, we present qualitative results of the generated samples in Figure 5. It can be seen that our model generates diverse high-quality images. The qualitative results of the reconstruction are shown in Figure 6, for which we use the test set of CelebA $64 \times 64$. The right column shows the original image $\mathbf{x}$ to be reconstructed. The left column shows the reconstruction based on a latent variable initialised from the base distribution $\mathbf{z}_0 \sim p_0(\mathbf{z})$, and the middle column displays the image reconstructed from $\mathbf{z}_k \sim p_{\phi,\theta}(\mathbf{z}|\mathbf{x})$ which is sampled via Langevin dynamics. One can see that our model can successfully reconstruct the test images, verifying the validity of the latent prior learned with energy discrepancy. In addition, we showcase the scalability of our approach by

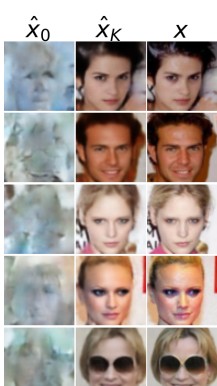

$\hat{x}_0$ $\hat{x}_K$ $x$

Figure 6: Image reconstruction results on CelebA $64 \times 64$.

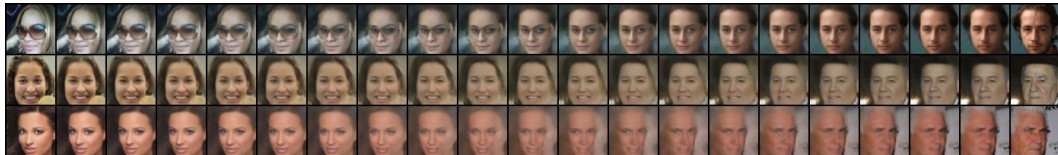

Figure 7: Image interpolation results on CelebA $64 \times 64$. The first and last columns show the observed images, while the middle columns display the interpolation results using the inferred latent vectors.

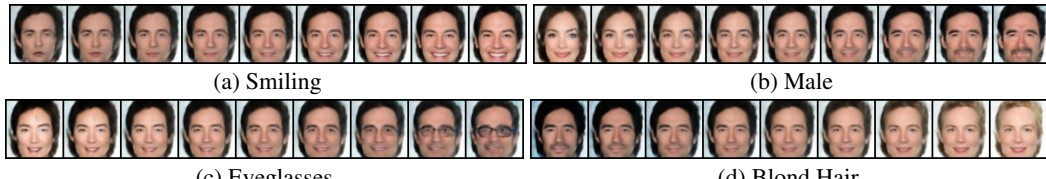

| (a) Smiling | (b) Male |

| (c) Eyeglasses | (d) Blond Hair |

Figure 8: Attribute manipulation by interpolating the latent variable along an attribute vector. The middle image corresponds to the original image, and each subfigure represents a different attribute.

Table 2: Comparison of AUPRC(↑) for unsupervised anomaly detection on MNIST dataset.

| Heldout Digit | 1 | 4 | 5 | 7 | 9 |
|---|---|---|---|---|---|
| VAE (Kingma & Welling, 2014) | 0.063 | 0.337 | 0.325 | 0.148 | 0.104 |
| ABP (Han et al., 2017) | $0.095 \pm 0.03$ | $0.138 \pm 0.04$ | $0.147 \pm 0.03$ | $0.138 \pm 0.02$ | $0.102 \pm 0.03$ |
| MEG (Kumar et al., 2019) | $0.281 \pm 0.04$ | $0.401 \pm 0.06$ | $0.402 \pm 0.06$ | $0.290 \pm 0.04$ | $0.342 \pm 0.03$ |
| BiGAN-$\sigma$ (Zenati et al., 2018) | $0.287 \pm 0.02$ | $0.443 \pm 0.03$ | $0.514 \pm 0.03$ | $0.347 \pm 0.02$ | $0.307 \pm 0.03$ |
| CD-LEBM (Pang et al., 2020) | $0.336 \pm 0.01$ | $0.630 \pm 0.02$ | $0.619 \pm 0.01$ | $0.463 \pm 0.01$ | $0.413 \pm 0.01$ |
| SM-LEBM | $0.285 \pm 0.01$ | $0.663 \pm 0.01$ | $0.610 \pm 0.01$ | $0.471 \pm 0.01$ | $0.422 \pm 0.01$ |
| ED-LEBM (ours) | $\mathbf{0.342 \pm 0.01}$ | $\mathbf{0.740 \pm 0.01}$ | $\mathbf{0.708 \pm 0.01}$ | $\mathbf{0.501 \pm 0.02}$ | $\mathbf{0.444 \pm 0.01}$ |

applying it successfully to high-resolution images (CelebA $128 \times 128$). More results can be found in Appendix D.5.

**Image Interpolation and Manipulation.** We use a latent-variable to model the effective low-dimensional structure in the data set. To probe how well the latent space describes the geometry and meaning of the data-manifold we analyse the structure of the latent space through interpolation and attribute manipulation. For the interpolation between two samples we linearly interpolate between their latent representations which were sampled from the posterior distribution. The results in Figure 7 demonstrate that the latent space has learned the data-manifold well and almost all intermediate samples appear as realistic faces. Further results are given in Figures 19 and 20. In addition, we can utilize the labels in the CelebA dataset to modify the attributes of an image through the manipulation technique proposed in (Kingma & Dhariwal, 2018). Specifically, each image in the dataset is associated with a binary label that indicates the presence or absence of attributes such as smiling, male, eyeglasses. For each manipulated attribute, we compute the average latent vectors $\mathbf{z}_{\mathrm{pos}}$ of images with and $\mathbf{z}_{\mathrm{neg}}$ of images without the attribute in the training set. Then, we use the difference $\mathbf{z}_{\mathrm{pos}} - \mathbf{z}_{\mathrm{neg}}$ as the direction for manipulating the attribute of an image. The results presented in Figures 8 and 22 confirm the meaningfulness of the latent space learned by energy discrepancy.

## 5.3 Anomaly Detection

In a well-learned energy-based model the likelihood should be higher for in-distribution examples and lower for out-of-distribution examples. Based on this principle, we conduct anomaly detection experiments to compare our method with other baselines. Specifically, given a test sample $\mathbf{x}$, we first sample $\mathbf{z}$ from the posterior $p_\theta(\mathbf{z}|\mathbf{x})$ by Langevin dynamics, and then compute the unnormalised log-density $p_\theta(\mathbf{x}, \mathbf{z})$ as the decision function. Following the protocol in (Zenati et al., 2018), we designate each digit class in the MNIST dataset as an anomaly and leave the rest as training data. The area under the precision-recall curve (AUPRC) is used to evaluate different methods. As shown in Table 2, our model consistently outperforms the baseline methods, demonstrating the advantages of training latent EBMs with energy discrepancy.

# 6 Related Work

**Training Energy-based models.** While energy-based models have been around for some time (Hinton, 2002), the training of energy-based models remains challenging. For a summary on existing methods for the training of energy-based models see Song & Kingma (2021) and LeCun et al. (2006). Contrastive divergence is still the most used option for training energy-based models. Recent extensions have improved the scalability of the basic algorithm, making it possible to train EBMs on high-dimensional data (Ngiam et al., 2011; Dai et al., 2015; Xie et al., 2016; Nijkamp et al., 2019; Du & Mordatch, 2019). Despite these advances, it has been noted that contrastive divergence is not the gradient of any fixed loss-function (Sutskever & Tieleman, 2010) and can yield energy functions that do not adequately reflect the data distribution (Nijkamp et al., 2020a). This has motivated improvements to the standard methodology (Du et al., 2021) by approximating overlooked entropy terms or by improving the convergence of MCMC sampling with an underlying diffusion model (Gao et al., 2021).

Score-based methods (Hyvärinen & Dayan, 2005; Vincent, 2011; Song et al., 2020) are typically implemented by directly learning the score function instead of the energy function. Song & Ermon (2019); Song et al. (2021b) point out the importance to introduce multiple noise levels at which the score is learned. Li et al. (2019) adopt the strategy of learning an energy-based model using multi-scale denoising score matching.

We learn a latent EBM prior (Pang et al., 2020) on high-dimensional data to address situations where the data lives on a lower dimensional submanifold. This methodology has been improved with a multi-stage approach (Xiao & Han, 2022) with score-independent noise contrastive estimation (Gutmann & Hyvärinen, 2010).

**Related loss functions.** Energy discrepancies can be derived from Kullback Leibler-contraction divergences (Lyu, 2011) which we also discuss briefly in Appendix A.6. Diffusion recovery likelihood (Gao et al., 2021) technically optimises the same loss as us but the EBM is trained with contrastive divergence. Luo et al. (2023) make similar observations to us while estimating the KL-contraction by leveraging the time evolution of the energy function along a diffusion process. Energy discrepancy also shares similarities with other contrastive loss functions in machine learning. The $w$-stabilised energy-discrepancy loss with $w, M = 1$ is equivalent to conditional noise contrastive estimation (Ceylan & Gutmann, 2018) with Gaussian noise. For $M > 1$, the stabilised ED loss shares great structural similarity to certain contrastive learning losses such as InfoNCE (van den Oord et al., 2018). We hope that such observations can lead to interpretations of the $w$-stabilisation.

**Theoretical connections between likelihood and score-based functionals.** The connection between likelihood methods and score-based methods generalises the de Bruijn identity (Stam, 1959; Lyu, 2009) which explains that score matching appears frequently as the limit of EBM training methods. Similar connections have been mentioned and exploited by Song et al. (2021a) to choose a weighting scheme for a combination of score matching losses.

**Generative modelling for manifold-valued data.** The manifold hypothesis was, for example, described in Bengio et al. (2013). Prior work to us shows how score-based diffusion models detect this manifold (Pidstrigach, 2022) and give reasons why CD is more robust to this issue than other contrastive methods (Yair & Michaeli, 2021). Arbel et al. (2021) learn an energy-based model as a tilt of a GAN-based prior that models the data-manifold. We believe that a combination of above results with energy discrepancy could enable training EBMs with energy discrepancy on data space.

# 7 Discussion and Outlook

We demonstrate that energy discrepancy provides a new tool for the fast training of energy-based models without MCMC or scores. We show for Euclidean data that ED interpolates between score matching and maximum likelihood estimation, thus alleviating problems of nearsightedness of score matching without additional annealing strategies as in score-based diffusion models. Based on our theoretical analysis, we show that training EBMs using energy discrepancy yields more accurate estimates of the energy function for two-dimensional data than explicit score matching and contrastive divergence. In this task, it is robust to the hyperparameters used, making energy discrepancy a useful tool for energy-based modelling with little tuning required. We further establish that energy discrepancy achieves comparable results to contrastive divergence at a lower computational cost

when learning a lower-dimensional energy-based prior for high-dimensional data. This shows that energy discrepancy is scalable to high dimensions.

**Limitations:** Energy-based models make the assumption that the data-distribution has a positive density which is violated for most high-dimensional data sets due to the manifold hypothesis. Compared to contrastive divergence or diffusion-based strategies, energy discrepancy is especially sensitive to such singularities in the data set. Currently, this limits energy discrepancy to settings in which a lower-dimensional representation with non-singular support can be learned or constructed efficiently and accurately.

**Outlook:** This work can be extended in various ways. Firstly, we want to explore other choices for the conditional distribution $q$. In particular, the effect of different types of noise such as Laplace noise or anisotropic Gaussian noise are open questions. Furthermore, energy discrepancy is a well-suited objective function for discrete data for appropriately chosen perturbations. We present a preliminary study of energy discrepancy on discrete spaces in the workshop paper Schröder et al. (2023). Secondly, we believe that improvements to our methodology can be made by learning the energy function of image-data on pixel space directly, as in this domain specific architectures such as U-nets (Ronneberger et al., 2015; Song & Ermon, 2019) can be used in the modelling. However, this requires further work in learning the data-manifold during training so that early saturation can be prevented. Finally, the partial differential equations arising in Euclidean energy discrepancies promise exciting insights into the connections between energy-based learning, stochastic differential equations, and optimal control which we want to investigate further.

# Acknowledgements

TS and ABD are grateful to G.A. Pavliotis for insightful discussions throughout the project. TS was supported by the EPSRC-DTP scholarship partially funded by the Department of Mathematics, Imperial College London. ZO was supported by the Lee Family Scholarship. JNL was supported by the Feuer International Scholarship in Artificial Intelligence. ABD was supported by Wave 1 of The UKRI Strategic Priorities Fund under the EPSRC Grant EP/T001569/1 and EPSRC Grant EP/W006022/1, particularly the "Ecosystems of Digital Twins" theme within those grants and The Alan Turing Institute. We thank the anonymous reviewer for their comments.

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

# Appendix for "Energy Discrepancies: A Score-Independent Loss for Energy-Based Models"

## Contents

## A  Proofs and Derivations

In this section we are going to prove Theorem 1 and Theorem 2. Furthermore, we are going to generalise Theorem 2 to arbitrary diffusion processes and discuss the connections of energy discrepancy to other training losses for energy-based models.

### A.1  Proof of the Non-Parametric Estimation Theorem 1

In this subsection we give a formal proof for the uniqueness of minima of $\mathrm{ED}_q(p_{\mathrm{data}}, U)$ as a functional in the energy function $U$. We first reiterate the theorem as stated in the paper:

**Theorem 1.** *Let $p_{\mathrm{data}}$ be a positive probability density on $(\mathcal{X}, \mathrm{d}\mathbf{x})$, and let $q(\cdot|\mathbf{x})$ be a conditional probability density. Under mild assumptions on $q$ and the set of admissible energy functions $U$, energy discrepancy $\mathrm{ED}_q$ is functionally convex in $U$ and has, up to additive constants, a unique global minimiser $U^* = \arg\min \mathrm{ED}_q(p_{\mathrm{data}}, U)$ with $\exp(-U^*) \propto p_{\mathrm{data}}$.*

For this theorem we need to specify additional assumptions on the conditional distribution $q$ and on the optimisation domain to guarantee uniqueness. Firstly, we require the energy-based distribution to be normalisable which implies that $\exp(-U) \in L^1(\mathcal{X}, \mathrm{d}\mathbf{x})$. For the existence and uniqueness of minimisers we have to constrain the space of energy functions since $\mathrm{ED}_q(p_{\mathrm{data}}, U) = \mathrm{ED}_q(p_{\mathrm{data}}, U + c)$

for any constant $c \in \mathbb{R}$. Hence, we restrict the optimisation domain to functions $U$ that satisfy $\min_{\mathbf{x} \in \mathcal{X}} U(\mathbf{x}) = 0$. To specify the condition on $q$ we define an equivalence relation on $\mathcal{X}$:

**Definition 2** ($q$-equivalence). *For a pair of samples $\mathbf{x}_1, \mathbf{x}_2 \in \mathcal{X}$ we say that $\mathbf{x}_1$ and $\mathbf{x}_2$ are $q$-neighbours if there exists a $\mathbf{y} \in \mathcal{Y}$ such that $\mathbf{x}_1, \mathbf{x}_2 \in \mathrm{supp}(q(\mathbf{y}|\cdot))$. We denote the set of $q$-neighbours of $\mathbf{x}$ as $\mathcal{N}_q(\mathbf{x})$. We then define the $q$-equivalence relation as:*

$$\mathbf{x}_1 \sim_q \mathbf{x}_2 \Leftrightarrow \text{ there exist } (\mathbf{z}_{1:R}) \text{ with } \mathbf{z}_1 = \mathbf{x}_1, \quad \mathbf{z}_R = \mathbf{x}_2, \text{ and } \quad \mathbf{z}_{r+1} \in \mathcal{N}_q(\mathbf{z}_r)$$

It is easy to see that $q$-equivalence defines an equivalence relation, i.e. it is symmetric, reflexive, and transitive. We summarise the assumptions for the non-parametric approximation theorem as follows:

**Assumption 1.** *We define the optimisation domain*

$$\mathcal{G} := \left\{ U : \mathcal{X} \mapsto \mathbb{R} \text{ such that } \exp(-U) \in L^1(\mathcal{X}, \mathrm{d}\mathbf{x}), \; U \in L^1(p_{\text{data}}), \text{ and } \min_{\mathbf{x} \in \mathcal{X}} U(x) = 0 \right\}$$

*We then make the following assumptions on $q$ and $\mathcal{G}$:*

*1. All elements of $\mathcal{X}$ are $q$-equivalent, i.e. for every $\mathbf{x} \in \mathcal{X}$ it holds that $[\mathbf{x}]_q = \mathcal{X}$.*

*2. There exists a $U^* \in \mathcal{G}$ such that $\exp(-U^*) \propto p_{\text{data}}$.*

Under Assumption 1, $\mathrm{ED}_q(p_{\text{data}}, U)$ has a unique global minimiser $U^* = -\log p_{\text{data}} + c$ in $\mathcal{G}$. We prove this by computing the first and second variation of $\mathrm{ED}_q$. Note that $\mathcal{G}$ may not be a vector space in general since in some cases $0 \notin \mathcal{G}$. Furthermore, we did not specify a norm on $\mathcal{G}$. We will omit this technical issue in this discussion. We start from the following lemmata and complete the proof of Theorem 1 in Corollary 1.

**Lemma 1.** *Let $h \in \mathcal{G}$ be arbitrary. The first variation of $\mathrm{ED}_q$ is given by*

$$\frac{\mathrm{d}}{\mathrm{d}\epsilon} \mathrm{ED}_q(p_{\text{data}}, U + \epsilon h) \Big|_{\epsilon=0} = \mathbb{E}_{p_{\text{data}}(\mathbf{x})}[h(\mathbf{x})] - \mathbb{E}_{p_{\text{data}}(\mathbf{x})} \mathbb{E}_{q(\mathbf{y}|\mathbf{x})} \mathbb{E}_{p_U(\mathbf{z}|\mathbf{y})}[h(\mathbf{z})] \tag{7}$$

*where $p_U(\mathbf{z}|\mathbf{y}) = \frac{q(\mathbf{y}|\mathbf{z}) \exp(-U(\mathbf{z}))}{\int q(\mathbf{y}|\mathbf{z}') \exp(-U(\mathbf{z}')) \mathrm{d}\mathbf{z}'}$.*

*Proof.* We define the short-hand notation $U_\epsilon := U + \epsilon h$. The energy discrepancy at $U_\varepsilon$ reads

$$\mathrm{ED}_q(p_{\text{data}}, U_\epsilon) = \mathbb{E}_{p_{\text{data}}(\mathbf{x})}[U_\epsilon(\mathbf{x})] + \mathbb{E}_{p_{\text{data}}(\mathbf{x})} \mathbb{E}_{q(\mathbf{y}|\mathbf{x})} \left[ \log \int q(\mathbf{y}|\mathbf{z}) \exp(-U_\epsilon(\mathbf{z})) \mathrm{d}\mathbf{z} \right].$$

For the first functional derivative, we only need to calculate

$$\frac{\mathrm{d}}{\mathrm{d}\epsilon} \log \int q(\mathbf{y}|\mathbf{z}) \exp(-U_\epsilon(\mathbf{z})) \mathrm{d}\mathbf{z} = \int \frac{-q(\mathbf{y}|\mathbf{z}) h(\mathbf{z}) \exp(-U_\epsilon(\mathbf{z}))}{\int q(\mathbf{y}|\mathbf{z}') \exp(-U_\epsilon(\mathbf{z}')) \mathrm{d}\mathbf{z}'} \mathrm{d}\mathbf{z} = -\mathbb{E}_{p_{U_\epsilon}(\mathbf{z}|\mathbf{y})}[h(\mathbf{z})]. \tag{8}$$

Plugging this expression into $\mathrm{ED}_q(p_{\text{data}}, U_\epsilon)$ and setting $\epsilon = 0$ yields the first variation of $\mathrm{ED}_q$. $\square$

**Lemma 2.** *The second variation of $\mathrm{ED}_q$ is given by*

$$\frac{\mathrm{d}^2}{\mathrm{d}\epsilon^2} \mathrm{ED}_q(p_{\text{data}}, U + \epsilon h) \Big|_{\epsilon=0} = \mathbb{E}_{p_{\text{data}}(\mathbf{x})} \mathbb{E}_{q(\mathbf{y}|\mathbf{x})} \mathrm{Var}_{p_U(\mathbf{z}|\mathbf{y})}[h(\mathbf{z})].$$

*Proof.* For the second order term, we have based on equation 8 and the quotient rule for derivatives:

$$\begin{aligned}
\frac{\mathrm{d}^2}{\mathrm{d}\epsilon^2} & \log \int q(\mathbf{y}|\mathbf{z}) \exp(-U_\epsilon(\mathbf{z})) \mathrm{d}\mathbf{z} \\
&= \frac{\int q(\mathbf{y}|\mathbf{z}) \exp(U_\epsilon(\mathbf{z})) h^2(\mathbf{z}) \mathrm{d}\mathbf{z} \int q(\mathbf{y}|\mathbf{z}') \exp(-U_\epsilon(\mathbf{z}')) \mathrm{d}\mathbf{z}'}{\left( \int q(\mathbf{y}|\mathbf{z}') \exp(-U_\epsilon(\mathbf{z}')) \mathrm{d}\mathbf{z}' \right)^2} \\
&\quad - \frac{\int q(\mathbf{y}|\mathbf{z}) \exp(U_\epsilon(\mathbf{z})) h(\mathbf{z}) \mathrm{d}\mathbf{z} \int q(\mathbf{y}|\mathbf{z}') \exp(-U_\epsilon(\mathbf{z}')) h(\mathbf{z}') \mathrm{d}\mathbf{z}'}{\left( \int q(\mathbf{y}|\mathbf{z}') \exp(-U_\epsilon(\mathbf{z}')) \mathrm{d}\mathbf{z}' \right)^2} \\
&= \mathbb{E}_{p_{U_\epsilon}(\mathbf{z}|\mathbf{y})}[h^2(\mathbf{z})] - \mathbb{E}_{p_{U_\epsilon}(\mathbf{z}|\mathbf{y})}[h(\mathbf{z})]^2 = \mathrm{Var}_{p_{U_\epsilon}(\mathbf{z}|\mathbf{y})}[h(\mathbf{z})].
\end{aligned}$$

We obtain the desired result by interchanging the outer expectations with the derivatives in $\epsilon$. $\square$

**Corollary 1.** *Let* $c = \min_{\mathbf{x} \in \mathcal{X}}(-\log p_{\text{data}}(\mathbf{x}))$. *For* $U^* = -\log(p_{\text{data}}) - c \in \mathcal{G}$ *it holds under Assumption 1 that*

$$\frac{\mathrm{d}}{\mathrm{d}\epsilon} \text{ED}_q(p_{\text{data}}, U^* + \epsilon h) \bigg|_{\epsilon=0} = 0$$

$$\frac{\mathrm{d}^2}{\mathrm{d}\epsilon^2} \text{ED}_q(p_{\text{data}}, U^* + \epsilon h) \bigg|_{\epsilon=0} > 0 \qquad \text{for all} \qquad h \in \mathcal{G}.$$

*Furthermore,* $U^*$ *is the unique global minimiser of* $\text{ED}_q(p_{\text{data}}, \cdot)$ *in* $\mathcal{G}$.

*Proof.* By definition, the variance is non-negative, i.e. for every $h \in \mathcal{G}$:

$$\frac{\mathrm{d}^2}{\mathrm{d}\epsilon^2} \text{ED}_q(p_{\text{data}}, U + \epsilon h) \bigg|_{\epsilon=0} = \text{Var}_{p_U(\mathbf{z}|\mathbf{y})}[h(\mathbf{z})] \geq 0.$$

Consequently, the energy discrepancy is convex and an extremal point of $\text{ED}_q(p_{\text{data}}, \cdot)$ is a global minimiser. We are left to show that the minimiser is obtained at $U^*$ and unique. First of all, we have:

$$\mathbb{E}_{p_{U^*}(\mathbf{z}|\mathbf{y})}[h(\mathbf{z})] = \int \frac{q(\mathbf{y}|\mathbf{z})\exp(-U^*(\mathbf{z}))}{\int q(\mathbf{y}|\mathbf{z}')\exp(-U^*(\mathbf{z}'))\mathrm{d}\mathbf{z}'} h(\mathbf{z})\mathrm{d}\mathbf{z} = \int \frac{q(\mathbf{y}|\mathbf{z})p_{\text{data}}(\mathbf{z})}{\int q(\mathbf{y}|\mathbf{z}')p_{\text{data}}(\mathbf{z}')\mathrm{d}\mathbf{z}'} h(\mathbf{z})\mathrm{d}\mathbf{z}.$$

By applying the outer expectations we obtain

$$\mathbb{E}_{p_{\text{data}}(\mathbf{x})}\mathbb{E}_{q(\mathbf{y}|\mathbf{x})}\mathbb{E}_{p_{U^*}(\mathbf{z}|\mathbf{y})}[h(\mathbf{z})] = \int\int p_{\text{data}}(\mathbf{x})q(\mathbf{y}|\mathbf{x})\mathrm{d}\mathbf{x} \int \frac{q(\mathbf{y}|\mathbf{z})p_{\text{data}}(\mathbf{z})}{\int q(\mathbf{y}|\mathbf{z}')p_{\text{data}}(\mathbf{z}')\mathrm{d}\mathbf{z}'} h(\mathbf{z}) \, \mathrm{d}\mathbf{y} \, \mathrm{d}\mathbf{z}$$

$$= \int\int q(\mathbf{y}|\mathbf{x})p_{\text{data}}(\mathbf{z})h(\mathbf{z}) \, \mathrm{d}\mathbf{y} \, \mathrm{d}\mathbf{z} = \mathbb{E}_{p_{\text{data}}(\mathbf{z})}[h(\mathbf{z})],$$

where we used that the marginal distributions $\int p_{\text{data}}(\mathbf{x})q(\mathbf{y}|\mathbf{x})\mathrm{d}\mathbf{x}$ cancel out and the conditional probability density integrates to one. This implies

$$\frac{\mathrm{d}}{\mathrm{d}\epsilon} \text{ED}_q(p_{\text{data}}, U^* + \epsilon h) \bigg|_{\epsilon=0} = \mathbb{E}_{p_{\text{data}}(\mathbf{z})}[h(\mathbf{z})] - \mathbb{E}_{p_{\text{data}}(\mathbf{z})}[h(\mathbf{z})] = 0.$$

for all $h \in \mathcal{G}$. We now show that the second variation is also positive definite, i.e.

$$\frac{\mathrm{d}^2}{\mathrm{d}\epsilon^2} \text{ED}_q(p_{\text{data}}, U^* + \epsilon h) \bigg|_{\epsilon=0} = \mathbb{E}_{p_{\text{data}}(\mathbf{x})}\mathbb{E}_{q(\mathbf{y}|\mathbf{x})}\text{Var}_{p_{\text{data}}(\mathbf{z}|\mathbf{y})}[h(\mathbf{z})] > 0.$$

where $p_{\text{data}}(\mathbf{z}|\mathbf{y}) = \frac{q(\mathbf{y}|\mathbf{z})p_{\text{data}}(\mathbf{z})}{\int q(\mathbf{y}|\mathbf{z}')p_{\text{data}}(\mathbf{z}')\mathrm{d}\mathbf{z}'}$. Assume that there exists a test function $h \in \mathcal{G}$ such that $\mathbb{E}_{p_{\text{data}}(\mathbf{x})}\mathbb{E}_{q(\mathbf{y}|\mathbf{x})}\text{Var}_{p_{\text{data}}(\mathbf{z}|\mathbf{y})}[h(\mathbf{z})] = 0$. Since this is the expectation of a non-negative random variable, it follows for all $\mathbf{y} \in \mathcal{Y}$ that $\text{Var}_{p_{\text{data}}(\mathbf{z}|\mathbf{y})}[h(\mathbf{z})] = 0$. Consequently, $h(\mathbf{z})$ has to be constant $p_{\text{data}}(\cdot|\mathbf{y})$-almost surely and there exists a measurable function $g : \mathcal{Y} \to \mathbb{R}$ such that $h(\mathbf{z}) = g(\mathbf{y})$ for all $\mathbf{z} \in \text{supp}(p(\cdot|\mathbf{y}))$. Next, assume that $\mathbf{z}_1$ and $\mathbf{z}_2$ are $q$-neighbours. By definition, there exists a $\mathbf{y} \in \mathcal{Y}$ such that $\mathbf{z}_1, \mathbf{z}_2 \in \text{supp}(q(\mathbf{y}|\cdot))$. Since the data density is positive, the support of $q(\mathbf{y}|\cdot)$ equals the support of $p_{\text{data}}(\cdot|\mathbf{y})$. Consequently, it holds that $h(\mathbf{z}_1) = g(\mathbf{y}) = h(\mathbf{z}_2)$. It now follows from the definition of $q$-equivalence that $h$ has to be constant almost surely on all $q$-equivalence classes $[\mathbf{x}]_q$, since any two elements in an equivalence class can be connected by a chain of $q$-neighbours. By Assumption 1, $[\mathbf{x}]_q = \mathcal{X}$, and thus $h$ is constant on $\mathcal{X}$. However, there are no non-trivial constant test function in $\mathcal{G}$. Hence, the second variation has to be positive definite for all $h \in \mathcal{G}$. Consequently, $U^*$ is the unique global minimiser of $\text{ED}_q$ which completes the statement in Theorem 1. $\square$

### A.2 Equivalence of Energy Discrepancy for Brownian Motion and Ornstein-Uhlenbeck Processes

In this subsection we show that an energy discrepancy based on an Ornstein-Uhlenbeck process is equivalent to the energy discrepancy based on a time-changed Brownian motion.

**Proposition 1.** *Let* $q_t$ *be the transition density for the Ornstein-Uhlenbeck process* $\mathrm{d}\mathbf{x}_t = \alpha \mathbf{x}_t \mathrm{d}t + \sqrt{\beta}\mathrm{d}\mathbf{w}_t$ *with standard Brownian motion* $\mathbf{w}_t$, *and let* $\gamma_t(\mathbf{y}|\mathbf{x}) \propto \exp(-\|\mathbf{y} - \mathbf{x}\|^2/2t)$ *be the Gaussian transition density of Brownian motion. Then,*

$$\text{ED}_{q_t}(p_{\text{data}}, U) = \text{ED}_{\gamma_{\sigma^2_{-\alpha}(t)}}(p_{\text{data}}, U) - \alpha t$$

*where* $\sigma_\alpha(t) = \sqrt{\frac{\beta}{2\alpha}(e^{2\alpha t} - 1)}$ *and* $\sigma_0(t) = \sqrt{\beta t}$.

*Proof.* At time $t$, the Ornstein-Uhlenbeck process has distribution

$$\mathbf{x}_t \stackrel{d}{=} e^{\alpha t}\mathbf{x}_0 + \sigma_\alpha(t)\boldsymbol{\xi}, \quad \boldsymbol{\xi} \sim \mathcal{N}(0, \mathbf{I}) \tag{9}$$

with $\sigma_\alpha(t) = \sqrt{\frac{\beta}{2\alpha}(e^{2\alpha t} - 1)}$ and $\sigma_0(t) = \sqrt{\beta t}$. The Ornstein-Uhlenbeck process is variance exploding for $\alpha \geq 0$ and variance preserving for $\alpha < 0$. Based on (9). the transition density of $\mathbf{x}_t$ is given as

$$q_t(\mathbf{y}|\mathbf{x}) = \frac{1}{\sqrt{2\pi\sigma_\alpha(t)}^d} \exp\left(-\frac{\|\mathbf{y} - e^{\alpha t}\mathbf{x}\|^2}{2\sigma_\alpha^2(t)}\right)$$

Hence, we obtain via the change of variables $\boldsymbol{\xi}' := (\mathbf{y} - e^{\alpha t}\mathbf{x})/\sigma_\alpha(t) \sim \mathcal{N}(0, \mathbf{I})$ for the contrastive potential

$$U_t(\mathbf{y}) = -\log \int q_t(\mathbf{y}|\mathbf{x}) \exp(-U(\mathbf{x}))d\mathbf{x}$$

$$= -\log \int \gamma_1(\boldsymbol{\xi}') \exp\left(-U\left(e^{-\alpha t}(\mathbf{y} - \sigma_\alpha(t)\boldsymbol{\xi}')\right)\right) d\boldsymbol{\xi}' - \alpha t.$$

We now evaluate the contrastive potential at the forward process $\mathbf{y} = \mathbf{x}_t$ which yields

$$U_t(\mathbf{x}_t) = -\log \int \gamma_1(\boldsymbol{\xi}') \exp\left(-U\left(e^{-\alpha t}(e^{\alpha t}\mathbf{x}_0 + \sigma_\alpha(t)\boldsymbol{\xi} - \sigma_\alpha(t)\boldsymbol{\xi}')\right)\right) d\boldsymbol{\xi}' - \alpha t$$

$$= -\log \int \gamma_1(\boldsymbol{\xi}') \exp\left(-U\left(\mathbf{x}_0 + \sigma_{-\alpha}(t)\boldsymbol{\xi} - \sigma_{-\alpha}(t)\boldsymbol{\xi}'\right)\right) d\boldsymbol{\xi}' - \alpha t$$

$$= -\log \int \gamma_{\sigma_{-\alpha}^2(t)}\left(\mathbf{w}_{\sigma_{-\alpha}^2(t)} - \mathbf{x}\right) \exp(-U(\mathbf{x})d\mathbf{x} - \alpha t$$

where we used that $e^{-\alpha t}\sigma_\alpha(t) = \sigma_{-\alpha}(t)$ in the second equality and the change of variables $\mathbf{x} = \mathbf{w}_{\sigma_{-\alpha}^2(t)} - \boldsymbol{\xi}'$ in the third equality. Hence, the energy discrepancy for the Ornstein-Uhlenbeck process is equivalent to the energy discrepancy for Brownian motion with time parameter

$$\sigma_{-\alpha}(t) = \sqrt{\frac{\beta}{2\alpha}(1 - e^{-2\alpha t})}$$

$\square$

Notice that for the *variance-exploding* process with $\alpha > 0$ the contrasting particles have a finite horizon since $\sigma_{-\alpha}(t) \xrightarrow{t\to\infty} \sqrt{\frac{\beta}{2\alpha}} < \infty$. Hence, the maximum-likelihood limit in Theorem 2 is only achieved for the variance preserving process with $\alpha < 0$ and for the critical case of Brownian motion with $\alpha = 0$.

### A.3 Interpolation between Score-Matching and Maximum-Likelihood Estimation

We first prove the result as stated in the Gaussian case. We then show how the result can be generalised to arbitrary diffusions by using Ito calculus.

**Gaussian case** Denote the Gaussian density as

$$\gamma_t(\mathbf{y} - \mathbf{x}) := \frac{1}{\sqrt{2\pi t}^d} \exp\left(-\frac{\|\mathbf{y} - \mathbf{x}\|^2}{2t}\right).$$

and define the convolved distributions $p_t := \gamma_t * p_{\text{data}}$ and $\exp(-U_t) := \gamma_t * \exp(-U)$.

**Proposition 2.** *The energy discrepancy is the multi noise-scale score-matching loss*

$$\text{ED}_{\gamma_t}(p_{\text{data}}, U) = \int_0^t \mathbb{E}_{p_s(\mathbf{x}_s)}\left[-\Delta U_s(\mathbf{x}_s) + \frac{1}{2}\|\nabla U_s(\mathbf{x}_s)\|^2\right] ds$$

*Proof.* It is known that $\gamma_t$ is the solution of the heat equation:

$$\partial_t \gamma_t(\mathbf{y} - \mathbf{x}) = \frac{1}{2}\Delta_y \gamma_t(\mathbf{y} - \mathbf{x}).$$

Consequently, both, $p_t$ and $\exp(-U_t)$ satisfy the heat-equation because the integral commutes with the differential operators. Based on the heat-equation we can derive the following non-linear partial differential equation for the contrastive potential $U_t$:

$$\begin{aligned}
\partial_t e^{-U_t(\mathbf{y})} &= \int \partial_t \gamma_t(\mathbf{y} - \mathbf{x}) e^{-U(\mathbf{x})} \mathrm{d}\mathbf{x} \\
&= \frac{1}{2} \int \Delta_{\mathbf{y}} \gamma_t(\mathbf{y} - \mathbf{x}) e^{-U(\mathbf{x})} \mathrm{d}\mathbf{x} \\
&= \frac{1}{2} \Delta_{\mathbf{y}} e^{-U_t(\mathbf{y})} \\
&= -\frac{1}{2} \nabla_{\mathbf{y}} \cdot \left( \left( \nabla_{\mathbf{y}} U_t(\mathbf{y}) \right) e^{-U_t(\mathbf{y})} \right) \\
&= \left( \frac{1}{2} \|\nabla_{\mathbf{y}} U_t(\mathbf{y})\|^2 - \frac{1}{2} \Delta_{\mathbf{y}} U_t(\mathbf{y}) \right) e^{-U_t(\mathbf{y})}
\end{aligned}$$

Since $\partial_t e^{-U_t} = -e^{-U_t} \partial_t U_t$, we get after cancellation of the exponentials:

$$\partial_t U_t(\mathbf{y}) = \frac{1}{2} \Delta_y U_t(\mathbf{y}) - \frac{1}{2} \|\nabla_{\mathbf{y}} U_t(\mathbf{y})\|^2$$

The integral notation of the contrastive term in energy discrepancy takes the form

$$\mathbb{E}_{p_{\text{data}}(\mathbf{x})} \mathbb{E}_{\gamma_t(\mathbf{y} - \mathbf{x})} [U_t(\mathbf{y})] = \int U_t(\mathbf{y}) p_t(\mathbf{y}) \mathrm{d}\mathbf{y}.$$

We now take a derivative of the energy discrepancy and find

$$\begin{aligned}
\partial_t \mathrm{ED}_{\gamma_t}(p_{\text{data}}, U) &= -\partial_t \int U_t(\mathbf{y}) p_t(\mathbf{y}) \mathrm{d}\mathbf{y} \\
&= -\int \left( \partial_t U_t(\mathbf{y}) \right) p_t(\mathbf{y}) \mathrm{d}\mathbf{y} - \int U_t(\mathbf{y}) \partial_t p_t(\mathbf{y}) \mathrm{d}\mathbf{y} \\
&= -\int \left( \partial_t U_t(\mathbf{y}) \right) p_t(\mathbf{y}) \mathrm{d}\mathbf{y} - \int U_t(\mathbf{y}) \frac{1}{2} \Delta_{\mathbf{y}} p_t(\mathbf{y}) \mathrm{d}\mathbf{y} \\
&= -\int \left( \partial_t U_t(\mathbf{y}) \right) p_t(\mathbf{y}) \mathrm{d}\mathbf{y} - \int \frac{1}{2} \left( \Delta_{\mathbf{y}} U_t(\mathbf{y}) \right) p_t(\mathbf{y}) \mathrm{d}\mathbf{y}
\end{aligned}$$

where we used integration by parts twice in the final equation to shift the differential operator from $p_t$ to $U_t$. Now, plugging in the differential equation for $U_t$ we find

$$\begin{aligned}
\partial_t \mathrm{ED}_{\gamma_t}(p_{\text{data}}, U) &= \int \left( -\frac{1}{2} \Delta_{\mathbf{y}} U_t(\mathbf{y}) + \frac{1}{2} \|\nabla_{\mathbf{y}} U_t(\mathbf{y})\|^2 - \frac{1}{2} \Delta_{\mathbf{y}} U_t(\mathbf{y}) \right) p_t(\mathbf{y}) \mathrm{d}\mathbf{y} \\
&= \mathbb{E}_{p_{\text{data}}(\mathbf{x})} \mathbb{E}_{\gamma_t(\mathbf{y} - \mathbf{x})} \left[ -\Delta_{\mathbf{y}} U_t(\mathbf{y}) + \frac{1}{2} \|\nabla_{\mathbf{y}} U_t(\mathbf{y})\|^2 \right]
\end{aligned}$$

Finally, we obtain energy discrepancy by integrating above expression:

$$\begin{aligned}
\mathrm{ED}_{\gamma_t}(p_{\text{data}}, U) &= \int_0^t \partial_s \mathrm{ED}_{\gamma_s}(p_{\text{data}}, U) \mathrm{d}s \\
&= \int_0^t \mathbb{E}_{p_{\text{data}}(\mathbf{x})} \mathbb{E}_{\gamma_s(\mathbf{y} - \mathbf{x})} \left[ -\Delta_{\mathbf{y}} U_s(\mathbf{y}) + \frac{1}{2} \|\nabla_{\mathbf{y}} U_s(\mathbf{y})\|^2 \right] \mathrm{d}s
\end{aligned}$$

This gives the desired integral representation in Proposition 2. $\qquad\square$

**Proposition 3.** *Let $\gamma_t$ be the Gaussian transition density and $p_{\text{ebm}} \propto \exp(-U)$ the energy-based distribution. The energy discrepancy converges to a cross entropy loss at a linear rate in time*

$$\left| \mathrm{ED}_{\gamma_t}(p_{\text{data}}, U) + \mathbb{E}_{p_{\text{data}}(\mathbf{x})} \left[ \log p_{\text{ebm}}(\mathbf{x}) \right] - c(t) \right| \leq \frac{1}{2t} \mathbb{W}_2^2(p_{\text{data}}, p_{\text{ebm}})$$

*where $c(t)$ is a renormalising constant independent of $U$.*

For the proof we employ the following lemma of Yihong Wu which was given in Raginsky & Sason (2013).

**Lemma 3.** *Let $\gamma_t$ be the Gaussian transition density of a standard Brownian motion. Let $\mu, \nu$ be probability distributions and denote $\mu_t := \gamma_t * \mu$ and $\nu_t := \gamma_t * \nu$. The following information-transport inequality holds:*

$$\mathrm{KL}(\mu_t \parallel \nu_t) \leq \frac{1}{2t} \mathbb{W}_2^2(\mu, \nu)$$

*Proof.* Let $\pi$ be a probability density of $(\mathbf{x}, \mathbf{x}')$ with marginal distributions $\mu(\mathbf{x})$ and $\nu(\mathbf{x}')$ (also called a coupling in optimal transport). We have by the decomposition of Kullback-Leibler divergences

$$\int \mathrm{KL}(\gamma_t(\cdot - \mathbf{x}) \parallel \gamma_t(\cdot - \mathbf{x}'))\pi(\mathbf{x}, \mathbf{x}')\mathrm{d}\mathbf{x}\mathrm{d}\mathbf{x}' - \mathrm{KL}(\mu_t \parallel \nu_t)$$

$$= \int \mathrm{KL}(\frac{\gamma_t(\mathbf{y} - \mathbf{x})\pi(\mathbf{x}, \mathbf{x}')}{\mu_t(\mathbf{y})} \parallel \frac{\gamma_t(\mathbf{y} - \mathbf{x})\pi(\mathbf{x}, \mathbf{x}')}{\nu_t(\mathbf{y})})\mu_t(\mathbf{y})\mathrm{d}\mathbf{y} \geq 0$$

Hence, we find by rearranging the inequality

$$\mathrm{KL}(\mu_t \parallel \nu_t) \leq \int \mathrm{KL}(\gamma_t(\cdot - \mathbf{x}) \parallel \gamma_t(\cdot - \mathbf{x}'))\pi(\mathbf{x}, \mathbf{x}')\mathrm{d}\mathbf{x}\mathrm{d}\mathbf{x}'$$

The right hand side is the Kullback-Leibler divergence between Gaussians, so

$$\mathrm{KL}(\gamma_t(\cdot - \mathbf{x}) \parallel \gamma_t(\cdot - \mathbf{x}')) = \frac{1}{2t}\|\mathbf{x} - \mathbf{x}'\|^2$$

Since the coupling was arbitrary, we can minimise over all couplings $\pi$ of $\mu$ and $\nu$ which results in the Wasserstein-distance

$$\mathrm{KL}(\mu_t \parallel \nu_t) \leq \min_{\pi \in \Pi(\mu,\nu)} \int \mathrm{KL}(\gamma_t(\cdot - \mathbf{x}) \parallel \gamma_t(\cdot - \mathbf{x}'))\pi(\mathbf{x}, \mathbf{x}')\mathrm{d}\mathbf{x}\mathrm{d}\mathbf{x}' = \frac{1}{2t}\mathbb{W}_2^2(\mu, \nu)$$

where $\Pi(\mu, \nu)$ denotes the set of all joint distributions with marginals $\mu$ and $\nu$. $\qquad\square$

The proof of Proposition 3 then follows:

*Proof.* Let $\mu = p_{\mathrm{data}}$ and $\nu = p_{\mathrm{ebm}}$, denote the convolved distributions as $p_{t,\mathrm{data}}$ and $p_{t,\mathrm{ebm}}$. Notice that for any $\mathbf{x}, \mathbf{y} \in \mathbb{R}^d$

$$\log \frac{p_{t,\mathrm{ebm}}(\mathbf{y})}{p_{\mathrm{ebm}}(\mathbf{x})} = U(\mathbf{x}) - U_t(\mathbf{y})$$

since both models have the same normalising constant. We then have for arbitrary $\mathbf{x}$

$$\mathrm{KL}(p_{t,\mathrm{data}} \parallel p_{t,\mathrm{ebm}}) = \int (\log p_{t,\mathrm{data}}(\mathbf{y}) - \log p_{t,\mathrm{ebm}}(\mathbf{y}))\, p_{t,\mathrm{data}}(\mathbf{y})\mathrm{d}\mathbf{y}$$

$$= \int \left(\log p_{t,\mathrm{data}}(\mathbf{y}) - \log \frac{p_{t,\mathrm{ebm}}(\mathbf{y})}{p_{\mathrm{ebm}}(\mathbf{x})} - \log p_{\mathrm{ebm}}(\mathbf{x})\right) p_{t,\mathrm{data}}(\mathbf{y})\mathrm{d}\mathbf{y}$$

$$= \int (\log p_{t,\mathrm{data}}(\mathbf{y}) + U_t(\mathbf{y}) - U(\mathbf{x}))\, p_{t,\mathrm{data}}(\mathbf{y})\mathrm{d}\mathbf{y} - \log p_{\mathrm{ebm}}(\mathbf{x})$$

$$= c(t) + \int U_t(\mathbf{y})p_{t,\mathrm{data}}(\mathbf{y})\mathrm{d}\mathbf{y} - U(\mathbf{x}) - \log p_{\mathrm{ebm}}(\mathbf{x})$$

with $U$ independent entropy term $c(t) := \mathbb{E}_{p_{t,\mathrm{data}}(\mathbf{y})}[\log p_{t,\mathrm{data}}(\mathbf{y})]$. Since $\mathbf{x}$ was chosen arbitrarily, we can integrate with respect to $p_{\mathrm{data}}(\mathbf{x})$ and find

$$0 \leq \mathrm{KL}(p_{t,\mathrm{data}} \parallel p_{t,\mathrm{ebm}})$$

$$= c(t) + \int U_t(\mathbf{y})p_{t,\mathrm{data}}(\mathbf{y})\mathrm{d}\mathbf{y} - \int U(\mathbf{x})p_{\mathrm{data}}(\mathbf{x}) - \int \log p_{\mathrm{ebm}}(\mathbf{x})p_{\mathrm{data}}(\mathbf{x})$$

$$= c(t) - \mathrm{ED}_{\gamma_t}(p_{\mathrm{data}}, U) - \mathbb{E}_{p_{\mathrm{data}}(\mathbf{x})}[\log p_{\mathrm{ebm}}(\mathbf{x})] \leq \frac{1}{2t}\mathbb{W}_2^2(p_{\mathrm{data}}, p_{\mathrm{ebm}})$$

$\qquad\square$

## A.4 Representing Energy Discrepancy as Multi-Scale SM for General Diffusion Processes

We now prove the connection between energy discrepancy and multi-noise scale score matching in a general context. For all following results we will assume that $\mathbf{x}_t$ is some stochastic diffusion process which satisfies the SDE $\mathrm{d}\mathbf{x}_t = a(\mathbf{x}_t)\mathrm{d}t + b(\mathbf{x}_t)\mathrm{d}\mathbf{w}_t$ and assume that $\mathbf{x}_0 \sim p_{\mathrm{data}}$. Let $q_t$ denote the associated transition probability density. To make the exposition cleaner we write $U_t := U_{q_t}$.

The main idea will be the following observation:

**Proposition 4.** *The diffusion-based energy discrepancy is given by the expectation of the Ito integral*

$$\mathrm{ED}_{q_t}(p_{\mathrm{data}}, U) = -\mathbb{E}\left[\int_0^t \mathrm{d}U_s(\mathbf{x}_s)\right]$$

*Proof.* The stochastic integral with respect to the differential $\mathrm{d}U_s(\mathbf{x}_s)$ is defined to satisfy the following generalisation of the fundamental theorem of calculus:

$$U_t(\mathbf{x}_t) = U_0(\mathbf{x}_0) + \int_0^t \mathrm{d}U_s(\mathbf{x}_s)$$

We obtain the desired result by taking expectations on both sides in $(\mathbf{x}_s)_{0 \leq s \leq t}$. $\qquad\square$

Notice that the law of the random variable $\mathbf{x}_s$ is fixed by the initial distribution of the diffusion $\mathbf{x}_0 \sim p_{\mathrm{data}}$. These distributions are implied when taking the expectation. We will now explore this connection further. For this we make some basic assumptions which allow us to connect stochastic differential equations with partial differential equations.

**Assumption 2.** *Consider the stochastic differential equation $\mathrm{d}\mathbf{x}_t = a(\mathbf{x}_t)\mathrm{d}t + b(\mathbf{x}_t)\mathrm{d}\mathbf{w}_t$ for drift $a : \mathbb{R}^d \to \mathbb{R}^d$ and $b : \mathbb{R}^d \to \mathbb{R}^k$. Further, define the diffusion matrix $\Sigma(\mathbf{x}) = b(\mathbf{x})b(\mathbf{x})^T \in \mathbb{R}^{d \times d}$. We make the following assumptions:*

1. *There exists a $\mu > 0$ such that for all $\boldsymbol{\xi}, \mathbf{x} \in \mathbb{R}^d$ $\langle \boldsymbol{\xi}, \Sigma(\mathbf{x})\boldsymbol{\xi} \rangle \geq \mu\|\boldsymbol{\xi}\|^2$*

2. *$\Sigma$ and $a$ are bounded and uniformly Lipschitz-continuous in $\mathbf{x}$ on every compact subset of $\mathbb{R}^d$*

3. *$\Sigma$ is uniformly Hölder-continuous in $\mathbf{x}$*

**Theorem 4** (Fokker-Planck equation). *Under Assumption 2, $\mathbf{x}_t$ has a transition density function given by*

$$\mathbb{P}(\mathbf{x}_t \in A | \mathbf{x}_0 = \mathbf{x}) = \int_A q_t(\mathbf{y}|\mathbf{x})\mathrm{d}\mathbf{y}\,.$$

*Furthermore, $q_t$ satisfies the Fokker-Planck partial differential equation*

$$\partial_t q_t(\mathbf{y}|\mathbf{x}) = \sum_{i=1}^d \partial_{\mathbf{y}_i}\left(-a_i(\mathbf{y})q_t(\mathbf{y}|\mathbf{x}) + \frac{1}{2}\sum_{j=1}^d \partial_{\mathbf{y}_j}\left(\Sigma_{ij}(\mathbf{y})q_t(\mathbf{y}|\mathbf{x})\right)\right) \tag{10}$$

$$q_0(\mathbf{y}|\mathbf{x}) = \delta(\mathbf{y} - \mathbf{x})$$

For a reference, see (Friedman, 2012, Theorem 5.4)

The Fokker-Planck equation yields the following important differential equation for the contrastive potential $U_t$:

**Proposition 5.** *Consider the stochastic differential equation $\mathrm{d}\mathbf{x}_t = a(\mathbf{x}_t)\mathrm{d}t + b(\mathbf{x}_t)\mathrm{d}\mathbf{w}_t$ for drift $a : \mathbb{R}^d \to \mathbb{R}^d$ and $b : \mathbb{R}^d \to \mathbb{R}^k$, and diffusion matrix $\Sigma(\mathbf{x}) = b(\mathbf{x})b(\mathbf{x})^T \in \mathbb{R}^{d \times d}$ that satisfies assumptions 2. Let $q_t$ be the associated transition density and define the contrastive potential $U_t(\mathbf{y}) := -\log \int q_t(\mathbf{y}|\mathbf{x})\exp(-U(\mathbf{x}))\mathrm{d}\mathbf{x}$. Furthermore, we define the scalar field*

$$c(a, \Sigma)(\mathbf{y}) := \sum_{i=1}^d \left(\partial_{\mathbf{y}_i}a_i(\mathbf{y}) - \frac{1}{2}\sum_{j=1}^d \partial_{\mathbf{y}_i}\partial_{\mathbf{y}_j}\Sigma_{ij}\right)$$

*and the linear operator*

$$\mathcal{L}(a, \Sigma) := \sum_{i=1}^{d} -a_i \frac{\partial}{\partial \mathbf{y}_i} + \frac{1}{2} \sum_{i,j=1}^{d} \left( 2\partial_{\mathbf{y}_j} \Sigma_{ij} \frac{\partial}{\partial \mathbf{y}_i} + \Sigma_{ij} \frac{\partial^2}{\partial \mathbf{y}_i \partial \mathbf{y}_j} \right).$$

*Then, the contrastive potential satisfies the non-linear partial differential equation*

$$\partial_t U_t(\mathbf{y}) = \mathcal{L}(a, \Sigma) U_t(\mathbf{y}) + \frac{1}{2} \|b^T(\mathbf{y}) \nabla U_t(\mathbf{y})\|^2 + c(a, \Sigma)(\mathbf{y})$$

*Proof.* We commute the linear operator of the Fokker-Planck equation to see that $e^{-U_t}$ satisfies the Fokker-Planck equation in Theorem 4, too, i.e.

$$\partial_t e^{-U_t(\mathbf{y})} = \sum_{i=1}^{d} \partial_{\mathbf{y}_i} \left( -a_i(\mathbf{y}) e^{-U_t(\mathbf{y})} + \frac{1}{2} \sum_{j=1}^{d} \partial_{\mathbf{y}_j} \left( \Sigma_{ij}(\mathbf{y}) e^{-U_t(\mathbf{y})} \right) \right)$$

$$e^{-U_0(\mathbf{y})} = e^{-U(\mathbf{x})}.$$

We now expand the term corresponding to the drift term:

$$\sum_{i=1}^{d} \partial_{\mathbf{y}_i} \left( -a_i(\mathbf{y}) e^{-U_t(\mathbf{y})} \right) = \sum_{i=1}^{d} \left( -\partial_{\mathbf{y}_i} a_i(\mathbf{y}) + a_i(\mathbf{y}) \partial_{\mathbf{y}_i} U_t(\mathbf{y}) \right) e^{-U_t(\mathbf{y})}$$

Similarly, we treat the diffusion term:

$$\frac{1}{2} \sum_{i,j=1}^{d} \partial_{\mathbf{y}_i} \partial_{\mathbf{y}_j} \left( \Sigma_{ij}(\mathbf{y}) e^{-U_t(\mathbf{y})} \right) = \frac{1}{2} \sum_{i,j=1}^{d} \Big( \partial_{\mathbf{y}_i} \partial_{\mathbf{y}_j} \Sigma_{ij}(\mathbf{y}) + \Sigma_{ij}(\mathbf{y}) \partial_{\mathbf{y}_j} U_t(\mathbf{y}) \partial_{\mathbf{y}_i} U_t(\mathbf{y})$$

$$- 2\partial_{\mathbf{y}_j} \Sigma_{ij}(\mathbf{y}) \partial_{\mathbf{y}_i} U_t(\mathbf{y}) - \Sigma_{ij}(\mathbf{y}) \partial_{\mathbf{y}_j} \partial_{\mathbf{y}_i} U_t(\mathbf{y}) \Big) e^{-U_t(\mathbf{y})}$$

Finally, the time derivative simply becomes $\partial_t e^{-U_t(\mathbf{y})} = -\partial_t U_t(\mathbf{y}) e^{-U_t(\mathbf{y})}$. We can now collect all terms independent of $U$ and identify

$$c(a, \Sigma)(\mathbf{y}) = \sum_{i=1}^{d} \left( \partial_{\mathbf{y}_i} a_i(\mathbf{y}) - \frac{1}{2} \sum_{j=1}^{d} \partial_{\mathbf{y}_i} \partial_{\mathbf{y}_j} \Sigma_{ij}(\mathbf{y}) \right)$$

as well as the linear operator term

$$\mathcal{L}(a, \Sigma) := \sum_{i=1}^{d} -a_i \frac{\partial}{\partial \mathbf{y}_i} + \frac{1}{2} \sum_{i,j=1}^{d} \left( 2\partial_{\mathbf{y}_j} \Sigma_{ij} \frac{\partial}{\partial \mathbf{y}_i} + \Sigma_{ij} \frac{\partial^2}{\partial \mathbf{y}_i \partial \mathbf{y}_j} \right)$$

Finally, we have

$$\sum_{i,j=1}^{d} \Sigma_{ij}(\mathbf{y}) \partial_{\mathbf{y}_j} U_t(\mathbf{y}) \partial_{\mathbf{y}_i} U_t(\mathbf{y}) = \sum_{i,j=1}^{d} \sum_{l=1}^{k} b_{il}(\mathbf{y}) b_{jl}(\mathbf{y}) \partial_{\mathbf{y}_j} U_t(\mathbf{y}) \partial_{\mathbf{y}_i}$$

$$= \sum_{l=1}^{k} \sum_{i=1}^{d} (b_{l,i}^T(\mathbf{y}) \partial_{\mathbf{y}_i} U_t(\mathbf{y}))^2 = \|b^T(\mathbf{y}) \nabla U_t(\mathbf{y})\|^2$$

This gives us the partial differential equation

$$-\partial_t U_t(\mathbf{y}) e^{-U_t(\mathbf{y})} = \left( -\mathcal{L}(a, \Sigma) U_t(\mathbf{y}) + \frac{1}{2} \|b^T(\mathbf{y}) \nabla U_t(\mathbf{y})\|^2 - c(a, \Sigma)(\mathbf{y}) \right) e^{-U_t(\mathbf{y})}$$

Cancelling all exponentials from both sides of the equation yields the desired result. □

**Theorem 5.** *The energy discrepancy takes the form of a generalised multi-noise scale score matching loss:*

$$\mathrm{ED}_{q_t}(p_{\mathrm{data}}, U) = \mathbb{E} \left[ \int_0^t -\sum_{i,j=1}^{d} \partial_{\mathbf{x}_j} \left( \Sigma_{ij}(\mathbf{x}_s) \partial_{\mathbf{x}_i} U(\mathbf{x}_s) \right) + \frac{1}{2} \|b^T(\mathbf{x}_s) \nabla U_s(\mathbf{x}_s)\|^2 \mathrm{d}s \right] + \mathrm{const.}$$

*Proof.* For this proof we return to the stochastic process $U_s(\mathbf{x}_s)$ from Proposition 4. By Ito's formula, $U_s(\mathbf{x}_s)$ satisfies the stochastic differential equation

$$\mathrm{d}U_s(\mathbf{x}_s) = \left(\partial_s U_s(\mathbf{x}_s) + \sum_{i=1}^{d} a_i(\mathbf{x}_s)\partial_{\mathbf{x}_i} U_s(\mathbf{x}_s) + \frac{1}{2}\sum_{i,j=1}^{d} \Sigma_{ij}(\mathbf{y})\partial_{\mathbf{x}_i}\partial_{\mathbf{x}_j} U(\mathbf{x}_s)\right)\mathrm{d}s$$
$$+ \sum_{i=1}^{d}\sum_{l=1}^{k} \partial_{\mathbf{x}_i} U_s(\mathbf{x}_s) b_{i,l}(\mathbf{x}_s) d\mathbf{w}_s^l$$

Under the additional integrability condition that $\mathbb{E}\int_0^t \|b^T(\mathbf{x}_s)\nabla U_s(\mathbf{x}_s)\|^2 \mathrm{d}s < \infty$, the stochastic integral with respect to Brownian motion $d\mathbf{w}_s$ has expectation zero. Furthermore, we can replace $\partial_s U_s(\mathbf{x}_s)$ with our previously obtained non-linear partial differential equation

$$\partial_s U_s(\mathbf{x}_s) = \mathcal{L}(a, \Sigma) U_s(\mathbf{x}_s) + \frac{1}{2}\|b^T(\mathbf{x}_s)\nabla U_s(\mathbf{x}_s)\|^2 + c(a, \Sigma)(\mathbf{x}_s)\,.$$

Due to opposing signs, the drift $a$ cancels, i.e.

$$\mathcal{L}(a, \Sigma) U_s(\mathbf{x}_s) + \sum_{i=1}^{d} a_i(\mathbf{x}_s)\partial_{\mathbf{x}_i} U_s(\mathbf{x}_s) + \frac{1}{2}\sum_{i,j=1}^{d} \Sigma_{ij}(\mathbf{y})\partial_{\mathbf{x}_i}\partial_{\mathbf{x}_j} U(\mathbf{x}_s)$$
$$= \sum_{i,j=1}^{d} \left(\partial_{\mathbf{y}_j}\Sigma_{ij}(\mathbf{y})\frac{\partial}{\partial \mathbf{y}_i} + \Sigma_{ij}(\mathbf{y})\frac{\partial^2}{\partial \mathbf{x}_i \partial \mathbf{x}_j}\right) U_s(\mathbf{x}_s)$$
$$= \sum_{i,j=1}^{d} \partial_{\mathbf{x}_j}\left(\Sigma_{ij}\partial_{\mathbf{x}_i} U(\mathbf{x}_s)\right)$$

Consequently, we obtain the final energy discrepancy expression using Proposition 4

$$\mathrm{ED}_{q_t}(p_{\mathrm{data}}, U) = -\mathbb{E}\left[\int_0^t dU_s(\mathbf{x}_s)\right]$$
$$= -\mathbb{E}\left[\int_0^t \sum_{i,j=1}^{d} \partial_{\mathbf{x}_j}\left(\Sigma_{ij}(\mathbf{x}_s)\partial_{\mathbf{x}_i} U(\mathbf{x}_s)\right) - \frac{1}{2}\|b^T(\mathbf{x}_s)\nabla U_s(\mathbf{x}_s)\|^2 \mathrm{d}s\right] + \mathrm{const.}$$

with $U$-independent constant $\int_0^t c(a, \Sigma)(\mathbf{x}_s)\mathrm{d}s$. This completes the proof. $\square$

As a corollary we obtain the proof of the first statement in Theorem 2: Assume that $q_t$ is defined through the stochastic differential equation $\mathrm{d}\mathbf{x}_t = a(\mathbf{x}_t)\mathrm{d}t + \mathrm{d}\mathbf{w}_t$. In this case, $\Sigma = \mathbf{I}$ and $\sum_{i,j=1}^{d} \partial_{\mathbf{x}_j}\left(\Sigma_{ij}(\mathbf{x}_s)\partial_{\mathbf{x}_i} U(\mathbf{x}_s)\right) = \Delta U(\mathbf{x}_s)$. Consequently, we obtain from Theorem 5 the score matching representation of $\mathrm{ED}_{q_t}$ in Theorem 2. In the special case that $\Sigma = bb^T$ is independent of $\mathbf{x}$ we obtain an integrated sliced score-matching loss (Song et al., 2020).

## A.5 Connections of Energy Discrepancy with Contrastive Divergence

The contrastive divergence update can be derived from an energy discrepancy when, for $E_\theta$ fixed, $q$ satisfies the detailed balance relation

$$q(\mathbf{y}|\mathbf{x})\exp(-E_\theta(\mathbf{x})) = q(\mathbf{x}|\mathbf{y})\exp(-E_\theta(\mathbf{y}))\,.$$

To see this, we calculate the contrastive potential induced by $q$: We have

$$-\log\int q(\mathbf{y}|\mathbf{x})\exp(-E_\theta(\mathbf{x}))\mathrm{d}\mathbf{x} = -\log\int q(\mathbf{x}|\mathbf{y})\exp(-E_\theta(\mathbf{y}))\mathrm{d}\mathbf{x} = E_\theta(\mathbf{y})\,.$$

Consequently, the energy discrepancy induced by $q$ is given by

$$\mathrm{ED}_q(p_{\mathrm{data}}, E_\theta) = \mathbb{E}_{p_{\mathrm{data}}(\mathbf{x})}[E_\theta(\mathbf{x})] - \mathbb{E}_{p_{\mathrm{data}}(\mathbf{x})}\mathbb{E}_{q(\mathbf{y}|\mathbf{x})}[E_\theta(\mathbf{y})]\,.$$

Updating $\theta$ based on a sample approximation of this loss leads to the contrastive divergence update

$$\Delta\theta \propto \frac{1}{N}\sum_{i=1}^{N}\nabla_\theta E_\theta(\mathbf{x}^i) - \frac{1}{N}\sum_{i=1}^{N}\nabla_\theta E_\theta(\mathbf{y}^i) \quad \mathbf{y}^i \sim q(\cdot|\mathbf{x}^i)$$

Three things are important to notice:

1. Implicitly, the distribution $q$ depends on $E_\theta$ and needs to adjusted in each step of the algorithm

2. For fixed $q$, $\mathrm{ED}_q(p_{\mathrm{data}}, E_\theta)$ satisfies Theorem 1. This means that each step of contrastive divergence optimises a loss with minimiser $E_\theta^* = -\log p_{\mathrm{data}} + c$. However, $q$ needs to be adjusted in each step as otherwise the contrastive potential is not given by the energy function $E_\theta$ itself.

3. This result highlights the importance to use Metropolis-Hastings adjusted Langevin-samplers to implement CD to ensure that the implied $q$ distribution satisfies the detailed balance relation. This matches the observations found by Yair & Michaeli (2021).

### A.6 Derivation of Energy Discrepancy from KL Contractions

A Kullback-Leibler contraction is the divergence function $\mathrm{KL}(p_{\mathrm{data}} \parallel p_{\mathrm{ebm}}) - \mathrm{KL}(Qp_{\mathrm{data}} \parallel Qp_{\mathrm{ebm}})$ (Lyu, 2011) for the convolution operator $Qp(\mathbf{y}) = \int q(\mathbf{y}|\mathbf{x})p(\mathbf{x})\mathrm{d}\mathbf{x}$. The linearity of the convolution operator retains the normalisation of the measure, i.e. for the energy-based distribution $p_{\mathrm{ebm}}$ we have

$$Qp_{\mathrm{ebm}} = \frac{1}{Z_U} \int q(\mathbf{y}|\mathbf{x}) \exp(-U(\mathbf{x})) \quad \text{with} \quad Z_U = \int \exp(-U(\mathbf{x}))\mathrm{d}\mathbf{x}.$$

The KL divergences then become with $U_q := -\log Q \exp(-U(\mathbf{x}))$

$$\mathrm{KL}(p_{\mathrm{data}} \parallel p_{\mathrm{ebm}}) = \mathbb{E}_{p_{\mathrm{data}}(\mathbf{x})}[\log p_{\mathrm{data}}(\mathbf{x})] + \mathbb{E}_{p_{\mathrm{data}}(\mathbf{x})}[U(\mathbf{x})] + \log Z_U$$

$$\mathrm{KL}(Qp_{\mathrm{data}} \parallel Qp_{\mathrm{ebm}}) = \mathbb{E}_{Qp_{\mathrm{data}}(\mathbf{y})}[\log Qp_{\mathrm{data}}(\mathbf{y})] + \mathbb{E}_{Qp_{\mathrm{data}}(\mathbf{y})}[U_q(\mathbf{y})] + \log Z_U$$

Since the normalisation cancels when subtracting the two terms we find

$$\mathrm{KL}(p_{\mathrm{data}} \parallel p_{\mathrm{ebm}}) - \mathrm{KL}(Qp_{\mathrm{data}} \parallel Qp_{\mathrm{ebm}}) = \mathrm{ED}_q(p_{\mathrm{data}}, U) + c$$

where $c$ is a constant that contains the $U$-independent entropies of $p_{\mathrm{data}}$ and $Qp_{\mathrm{data}}$.

## B Aspects of Training Energy-Based Models with Energy Discrepancy

In this section, we discuss the $w$-stabilisation in depth. Furthermore, we give a proof for Theorem 3 and show energy discrepancy can be approximated for other diffusion processes via the Feynman-Kac formula.

### B.1 Conceptual Understanding of the $w$-Stabilisation

The $w$-stabilisation is a useful stabilisation for all types of perturbation $q$. For example, we use the same stabilisation in the discrete setting (see Appendix D.3). To provide a better intuition for the stabilisation, we will investigate it more closely for the Gaussian perturbation.

The critical step for using energy discrepancy in practice is a stable approximation of the contrastive potential. For the Gaussian-based energy discrepancy, we can write the contrastive potential as $U_t(\mathbf{y}) = -\log \mathbb{E}[\exp(-U(\mathbf{y} + \sqrt{t}\boldsymbol{\xi}'))]$ with $\boldsymbol{\xi}' \sim \mathcal{N}(0, \mathbf{I})$ and $\mathbf{y} \in \mathbb{R}^d$. A naive approximation of the expectation with a Monte-Carlo estimator, however, is biased because of Jensen's inequality, i.e. for $\boldsymbol{\xi}'$, $\boldsymbol{\xi}'^j \sim \mathcal{N}(0, \mathbf{I})$ we have

$$U_t(\mathbf{y}) = -\log \mathbb{E}\left[\exp(-U(\mathbf{y} + \sqrt{t}\boldsymbol{\xi}'))\right] < -\mathbb{E}\left[\log \frac{1}{M}\sum_{j=1}^{M} \exp(-U(\mathbf{y} + \sqrt{t}\boldsymbol{\xi}'^j))\right].$$

Our first observation is that the appearing bias can be quantified to leading order. For this, define $v_t(\mathbf{y}) := \mathbb{E}\left[\exp(-U(\mathbf{y} + \sqrt{t}\boldsymbol{\xi}'))\right]$ and $\widehat{v}_t(\mathbf{y}) := \frac{1}{M}\sum_{j=1}^{M} \exp(-U(\mathbf{y} + \sqrt{t}\boldsymbol{\xi}'^j))$. We use the Taylor-expansion of $\log(1 + u) \approx u - 1/2u^2 + \text{h.o.t.}$ which gives

$$\mathbb{E}\log\widehat{v}_t(\mathbf{y}) - \log v_t(\mathbf{y}) = \mathbb{E}\left[\log\left(1 + \frac{\widehat{v}_t(\mathbf{y}) - v_t(\mathbf{y})}{v_t(\mathbf{y})}\right)\right] \tag{11}$$

$$\approx -\frac{1}{2}\mathbb{E}\left[\frac{(\widehat{v}_t(\mathbf{y}) - v_t(\mathbf{y}))^2}{v_t(\mathbf{y})^2}\right] + \text{h.o.t.}$$

$$= -\frac{1}{2Mv_t(\mathbf{y})^2}\mathrm{Var}\left[\exp(-U(\mathbf{y} + \sqrt{t}\boldsymbol{\xi}'))\right] + \text{h.o.t.}$$

The linear term in the Taylor-expansion does not contribute because $\mathbb{E}[\widehat{v}_t(\mathbf{y})] = v_t(\mathbf{y})$. In the final equation we used that $\mathrm{Var}(\widehat{v}_t(\mathbf{y})) = \mathrm{Var}[\exp(-U(\mathbf{y} + \sqrt{t}\boldsymbol{\xi}'))]/M$ because all $\boldsymbol{\xi}'^j$ are independent. The Taylor expansion shows that the dominating contribution to the bias is the variance of the approximated convolution integral.

Our second observation is that this occurring bias can become infinite for malformed energy functions. For this reason, the optimiser may start to increase the bias instead of minimising our target loss. To illustrate how a high-variance estimator of the contrastive potential can be divergent, consider the energy function

$$U(\mathbf{x}) = \begin{cases} 0 & \text{for } \mathbf{x} \leq 0 \\ b\mathbf{x} & \text{for } \mathbf{x} > 0 \end{cases}.$$

The energy function does not strictly adhere to our conditions that the energy based model should be normalisable. Our argument still holds when $\exp(-U)$ is changed to be normalisable. In theory, the contrastive potential at $0$ is upper bounded because

$$U_t(0) \leq -\lim_{b \to \infty} \log \mathbb{E}\left[\exp(-U(\sqrt{t}\boldsymbol{\xi}'))\right] = -\log \mathbb{P}(\boldsymbol{\xi}' \leq 0) = -\log(1/2)$$

because $\exp(-U(\mathbf{x}))$ converges to an indicator function on $\{\mathbf{x} \leq 0\}$ as $b \to \infty$. The Monte Carlo estimator of the contrastive potential, on the other hand, has upper bound

$$\widehat{U}_t(0) = -\log \frac{1}{M} \sum_{j=1}^{M} \exp(-U(\sqrt{t}\boldsymbol{\xi}'^j)) \leq \min[U(\sqrt{t}\boldsymbol{\xi}'^1), \ldots, U(\sqrt{t}\boldsymbol{\xi}'^M)] + \log(M)$$

which can be seen by applying standard inequalities for the logsumexp function[4]. Hence, as long as there exists a $j$ such that $\boldsymbol{\xi}^j \leq 0$, the estimated contrastive potential does not diverge. If, however, $\boldsymbol{\xi}'^j > 0$ for every $j = 1, \ldots, M$, then

$$\widehat{U}_t(0) \geq \min[U(\sqrt{t}\boldsymbol{\xi}'^1), \ldots, (\sqrt{t}\boldsymbol{\xi}'^M)] = b\sqrt{t}\min[\boldsymbol{\xi}'^1, \ldots, \boldsymbol{\xi}'^M] \xrightarrow{b \to \infty} \infty.$$

Consequently, the approximate contrastive potential may attain diverging values at discontinuities in the energy function. Indeed, this phenomenon is observed for $w = 0$ in Figure 2. Here, the learned energy becomes discontinuous at the edge of the support and the energy discrepancy loss diverges during training. In low dimensions, this problem can be alleviated by using variance reduction techniques such as antithetic variables or by using large enough values of $M$ during the training. The stabilising effect of $M$ is observed in our ablation studies in Figure 24. In high-dimensional settings, however, such variance reduction techniques are infeasible.

The idea of the $w$-stabilisation is that the value of the energy at non-perturbed data points $U(\mathbf{x}_0)$ is guaranteed to stay controlled since it is minimised in the optimisation of ED. Hence, the diverging contrasting potential can be controlled by including $U(\mathbf{x}_0)$ in the summation in the logsumexp operation which acts as a soft-min over all contrasting energy contributions. Indeed, this augmentation provides a deterministic upper bound to the approximated contrastive potential:

$$\widehat{U}_{t,w}(\mathbf{x}_t) = -\log\left(\frac{w}{M}\exp(-U(\mathbf{x}_0)) + \frac{1}{M}\sum_{j=1}^{M}\exp(-U(\mathbf{x}_t + \sqrt{t}\boldsymbol{\xi}'^j))\right)$$

$$\leq \min[U(\mathbf{x}_t + \sqrt{t}\boldsymbol{\xi}'^1), \ldots, U(\mathbf{x}_t + \sqrt{t}\boldsymbol{\xi}'^M), U(\mathbf{x}) - \log(w)] + \log(M)$$

Additionally, the $w$-stabilisation introduces a negative bias to the approximated contrastive potential. Hence, if tuned correctly, it counteracts the bias introduced by the Jensen-gap of the logarithm.

To gain additional intuition on the effect of $w$, notice that by the same bounds as before,

$$U(\mathbf{x}_0) - \widehat{U}_{t,w}(\mathbf{x}_t) \leq \min\left[U(\mathbf{x}_0) - U(\mathbf{x}_t + \sqrt{t}\boldsymbol{\xi}'^1), \ldots, U(\mathbf{x}_0) - U(\mathbf{x}_t + \sqrt{t}\boldsymbol{\xi}'^M), \log(w)\right]$$

for every data point $\mathbf{x}$. This tells us that, roughly speaking, a perturbed data point with $U(\mathbf{x}_0) - U(\mathbf{x}_t + \sqrt{t}\boldsymbol{\xi}') > \log(w)$ should have a small contribution to the loss and the optimisation converges if the data distribution is learned or when the bound is violated at all perturbed data points. Thus, $\log(w)$ describes a weak notion of a margin between positive and negative energy contributions. Consequently, large values for $w \in (0, 1)$ tend to lead to flatter learned energies, while smaller values lead to steeper learned energies. This intuition is confirmed by Figures 2 and 24.

---

[4]It holds that $\min(u_1, u_2, \ldots, u_M) - \log(M) \leq -\mathrm{LSE}(-u_1, -u_2, \ldots, -u_M) \leq \min(u_1, u_2, \ldots, u_M)$

**Asymptotic consistency of sample approximation of ED**  We give a proof for Theorem 3 which states that our approximation of energy discrepancy is justified. To make the exposition easier to understand, we first show how the energy discrepancy is transformed into a conditional expectation. Recall the probabilistic representation of the contrastive potential Section 4. Using $E_\theta(\mathbf{x}) = \log(\exp(E_\theta(\mathbf{x})))$ we obtain the following rewritten form of energy discrepancy:

$$\mathrm{ED}_{\gamma_t}(p_{\mathrm{data}}, E_\theta) = \mathbb{E}_{p_{\mathrm{data}}(\mathbf{x})}[E_\theta(\mathbf{x})] + \mathbb{E}_{p_{\mathrm{data}}(\mathbf{x})}\mathbb{E}_{\gamma_t(\mathbf{y}-\mathbf{x})}\left[\log \mathbb{E}_{\gamma_1(\xi')}\left[\exp(-E_\theta(\mathbf{y}+\sqrt{t}\xi')|\mathbf{y}]\right]\right]$$

$$= \mathbb{E}_{p_{\mathrm{data}}(\mathbf{x})}[E_\theta(\mathbf{x})] + \mathbb{E}_{p_{\mathrm{data}}(\mathbf{x})}\mathbb{E}_{\gamma_1(\xi)}\left[\log \mathbb{E}_{\gamma_1(\xi')}\left[\exp(-E_\theta(\mathbf{x}+\sqrt{t}\xi+\sqrt{t}\xi')|\mathbf{x},\xi]\right]\right]$$

$$= \mathbb{E}_{p_{\mathrm{data}}(\mathbf{x})}\mathbb{E}_{\gamma_1(\xi)}\left[\log(\exp(E_\theta(\mathbf{x}))) + \log \mathbb{E}_{\gamma_1(\xi')}\left[\exp(-E_\theta(\mathbf{x}+\sqrt{t}\xi+\sqrt{t}\xi')|\mathbf{x},\xi]\right]\right]$$

$$= \mathbb{E}_{p_{\mathrm{data}}(\mathbf{x})}\mathbb{E}_{\gamma_1(\xi)}\left[\log\left(\mathbb{E}_{\gamma_1(\xi')}\left[\exp(E_\theta(\mathbf{x})-E_\theta(\mathbf{x}+\sqrt{t}\xi+\sqrt{t}\xi')|\mathbf{x},\xi]\right)\right)\right]$$

The conditioning means that the expectation is not taken with respect to $\mathbf{y}$ or $\mathbf{x}$ and $\xi$ in the inner expectation. The conditioning is important to understand how the law of large numbers is to be applied. We now come to the proof that our approximation is consistent with the definition of energy discrepancy:

**Theorem 3.** *Assume that $\mathbf{x} \mapsto \exp(-E_\theta(\mathbf{x}))$ is uniformly bounded. Then, for every $\varepsilon > 0$ there exist $N$ and $M(N)$ such that $\left|\mathcal{L}_{t,M(N),w}(\theta) - \mathrm{ED}_{\gamma_t}(p_{\mathrm{data}}, E_\theta)\right| < \varepsilon$ almost surely.*

*Proof.* First, consider independent random variables $\mathbf{x} \sim p_{\mathrm{data}}$, $\xi \sim \mathcal{N}(0, \mathbf{I})$, and $\xi'^j \overset{i.i.d}{\sim} \mathcal{N}(0, \mathbf{I})$. Using the triangle inequality, we can upper bound the difference $|\mathrm{ED}_{\gamma_t}(p_{\mathrm{data}}, E_\theta) - \mathcal{L}_{t,M,w}(\theta)|$ by upper bounding the following two terms, individually:

$$\left|\mathrm{ED}_{\gamma_t}(p_{\mathrm{data}}, E_\theta) - \frac{1}{N}\sum_{i=1}^{N}\log \mathbb{E}\left[\exp(E_\theta(\mathbf{x}^i)-E_\theta(\mathbf{x}^i+\sqrt{t}\xi^i+\sqrt{t}\xi'^j))\,\Big|\,\mathbf{x}^i, \xi^i\right]\right|$$

$$+ \left|\frac{1}{N}\sum_{i=1}^{N}\log \mathbb{E}\left[\exp(E_\theta(\mathbf{x}^i)-E_\theta(\mathbf{x}^i+\sqrt{t}\xi^i+\sqrt{t}\xi'^j))\,\Big|\,\mathbf{x}^i, \xi^i\right] - \mathcal{L}_{t,M,w}(\theta)\right|$$

The first term can be bounded by a sequence $\varepsilon_N \xrightarrow{a.s.} 0$ due to the normal strong law of large numbers. The second term can be estimated by applying the following conditional version of the strong law of large numbers (Majerek et al., 2005, Theorem 4.2):

$$\frac{1}{M}\sum_{j=1}^{M}\exp\left(E_\theta(\mathbf{x})-E_\theta(\mathbf{x}+\sqrt{t}\xi+\sqrt{t}\xi'^j)\right) \xrightarrow{a.s.} \mathbb{E}\left[\exp(E_\theta(\mathbf{x})-E_\theta(\mathbf{x}+\sqrt{t}\xi+\sqrt{t}\xi'))\,\Big|\,\mathbf{x}, \xi\right]$$

Next, we have that the deterministic sequence $w/M \to 0$. Thus, adding the regularistion $w/M$ does not change the limit in $M$. Furthermore, since the logarithm is continuous, the limit also holds after applying the logarithm. Finally, the estimate translates to the sum by another application of the triangle inequality. We define

$$\Delta e_\theta(\mathbf{x}, \xi, \xi') := \exp(E_\theta(\mathbf{x})-E_\theta(\mathbf{x}+\sqrt{t}\xi+\sqrt{t}\xi'))$$

For each $i = 1, 2, \ldots, N$ there exists a sequence $\varepsilon_{i,M} \xrightarrow{a.s.} 0$ such that

$$\left|\frac{1}{N}\sum_{i=1}^{N}\log \mathbb{E}\left[\Delta e_\theta(\mathbf{x}^i, \xi^i, \xi')\,\Big|\,\mathbf{x}^i, \xi^i\right] - \mathcal{L}_{t,M,w}(\theta)\right|$$

$$\leq \frac{1}{N}\sum_{i=1}^{N}\left|\log \mathbb{E}\left[\Delta e_\theta(\mathbf{x}^i, \xi^i, \xi')\,\Big|\,\mathbf{x}^i, \xi^i\right] - \log \frac{1}{M}\sum_{j=1}^{M}\Delta e_\theta(\mathbf{x}^i, \xi^i, \xi'^j)\right|$$

$$< \frac{1}{N}\sum_{i=1}^{N}\varepsilon_{i,M} \leq \max(\varepsilon_{1,M}, \ldots, \varepsilon_{N,M}).$$

Hence, for each $\varepsilon > 0$ there exists an $N \in \mathbb{N}$ and an $M(N) \in \mathbb{N}$ such that $|\mathrm{ED}_{\gamma_t}(p_{\mathrm{data}}, E_\theta) - \mathcal{L}_{t,M(N),w}(\theta)| < \varepsilon$ almost surely. $\qquad\square$

## B.2 Approximation of Energy Discrepancy based on general Ito Diffusions

Energy discrepancies are useful objectives for energy-based modelling when the contrastive potential can be approximated easily and stably. In most cases this requires us to write the contrastive potential as an expectation which can be computed using Monte Carlo methods. We show how such a probabilistic representation can be achieved for a much larger class of stochastic processes via application of the Feynman-Kac formula. We first highlight the difficulty. Consider the integral $\int f(\mathbf{y})q_t(\mathbf{y}|\mathbf{x})p_{\text{data}}(\mathbf{x})\mathrm{d}\mathbf{x}\mathrm{d}\mathbf{y}$. Since the expectation is taken in $\mathbf{y}$, the integral can be represented as an expectation of the forward process associated with $q_t$, i.e.

$$\int f(\mathbf{y})q_t(\mathbf{y}|\mathbf{x})p_{\text{data}}(\mathbf{x})\mathrm{d}\mathbf{x}\mathrm{d}\mathbf{y} = \mathbb{E}_{p_{\text{data}}(\mathbf{x}_0)}[f(\mathbf{x}_t)] \approx \frac{1}{N}\sum_{i=1}^{N} f(\mathbf{x}_t^i)$$

where $\mathbf{x}_t^i$ are simulated processes initialised at $\mathbf{x}_0^i = \mathbf{x}^i \sim p_{\text{data}}$. Next, consider the integral

$$v(t, \mathbf{y}) := \int q_t(\mathbf{y}|\mathbf{x})g(\mathbf{x})\mathrm{d}\mathbf{x}\,.$$

This integral is more difficult to approximate because the function $g$ is evaluated at the starting point of the diffusion $\mathbf{x}_t$ but weighted by it's transition probability density. To compute such integrals without sampling from $g$ we use the Feynman-Kac formula, see e.g. Øksendal (2003):

**Theorem 6** (Feynman-Kac). *Let $g \in \mathcal{C}_0^2(\mathbb{R}^d)$ and $c \in \mathcal{C}(\mathbb{R}^d)$. Assume that $v \in \mathcal{C}^{1,2}(\mathbb{R}_{\geq 0}, \mathbb{R}^d)$ is bounded on $K \times \mathbb{R}^d$ with $K$ compact and satisfies*

$$\begin{cases} \partial_t v(t, \mathbf{y}) = \mathcal{A}v(t, \mathbf{y}) + c(\mathbf{y})v(t, \mathbf{y}) & \text{for all } t > 0,\ \mathbf{y} \in \mathbb{R}^d \\ v(0, \mathbf{y}) = g(\mathbf{y}) & \text{for all } \mathbf{y} \in \mathbb{R}^d \end{cases}. \tag{12}$$

*Then, $v$ has the probabilistic representation*

$$v(t, \mathbf{y}) = \mathbb{E}_{\mathbf{y}}\left[\exp\left(\int_0^t c(\mathbf{y}_s)\mathrm{d}s\right)g(\mathbf{y}_t)\right]$$

*where $(\mathbf{y}_t)_{t\geq 0}$ is a diffusion process with infinitesimal generator $\mathcal{A}$.*

We will establish that $v(t, \mathbf{y})$ satisfies a partial differential equation of the above form which yields a probabilistic representation of the contrastive potential. We know that $v(t, \mathbf{y})$ satisfies the Fokker-Planck equation (10). By applying the product rule to each term in the Fokker-Planck equation we find

$$\partial_t v(t, \mathbf{y}) = \left(\overbrace{\sum_{i=1}^{d}\left(-a_i(\mathbf{y}) + \sum_{j=1}^{d}\partial_{\mathbf{y}_j}\Sigma_{ij}(\mathbf{y})\right)\partial_{\mathbf{y}_i}}^{\alpha(\mathbf{y})} + \frac{1}{2}\sum_{i,j}^{d}\Sigma_{ij}(\mathbf{y})\partial_{\mathbf{y}_i}\partial_{\mathbf{y}_j}\right)v(t, \mathbf{y})$$

$$+ \underbrace{\left(\frac{1}{2}\sum_{i,j=1}^{d}\partial_{\mathbf{y}_i}\partial_{\mathbf{y}_j}\Sigma_{ij}(\mathbf{y}) - \sum_{i=1}^{d}\partial_{\mathbf{y}_i}a_i(\mathbf{y})\right)}_{c(\mathbf{y})}v(t, \mathbf{y})$$

$$v(0, \mathbf{y}) = g(\mathbf{y})\,.$$

By comparing with Theorem 6, we identify the infinitesimal generator

$$\mathcal{A} = \sum_{i=1}^{d}\alpha_i(\mathbf{y})\frac{\partial}{\partial_{\mathbf{y}_i}} + \frac{1}{2}\sum_{i=1}^{d}\sum_{j=1}^{d}\Sigma_{ij}(\mathbf{y})\frac{\partial^2}{\partial_{\mathbf{y}_i}\partial_{\mathbf{y}_j}}$$

Hence we associate the forward diffusion process $\mathrm{d}\mathbf{x}_t = a(\mathbf{x}_t)\mathrm{d}t + b(\mathbf{x}_t)\mathrm{d}\mathbf{w}_t$ with it's backwards process with infinitesimal generator $\mathcal{A}$

$$\mathrm{d}\mathbf{y}_t = \alpha(\mathbf{y}_t)\mathrm{d}t + b(\mathbf{y}_t)\mathrm{d}\mathbf{w}_t'$$

with $\Sigma(\mathbf{y}) = b(\mathbf{y})b^T(\mathbf{y})$. This yields the probabilistic representation of $v(t, \mathbf{y})$ in terms of the backward process $\mathbf{y}_t$:

$$v(t, \mathbf{y}) = \int q_t(\mathbf{y}|\mathbf{x})g(\mathbf{x})\mathrm{d}\mathbf{x} = \mathbb{E}_{\mathbf{y}}\left[\exp\left(\int_0^t c(\mathbf{y}_s)\mathrm{d}s\right)g(\mathbf{y}_t)\right]$$

Hence, we also obtain a probabilistic representation for the contrastive potential by choosing $g(\mathbf{x}) := \exp(-U(\mathbf{x}))$. This finally gives

$$U_t(\mathbf{y}) = -\log\mathbb{E}_{\mathbf{y}}\left[\exp\left(\int_0^t c(\mathbf{y}_s)\mathrm{d}s - U(\mathbf{y}_t)\right)\right] .$$

Unlike the contrasting term in contrastive divergence, this expression can indeed be calculated by simulating stochastic processes that are entirely independent of $U$. For this we simulate from the forward process starting at $\mathbf{x}$ which yields $\tilde{\mathbf{x}}_t$, where the tilde denotes that this simulation may not be exact. We then simulate $M$ copies of the reverse process and keep all values at intermediate steps, i.e. $(\tilde{\mathbf{y}}_{t_0}^j = \tilde{\mathbf{x}}_t, \tilde{\mathbf{y}}_{t_1}^j, \ldots, \tilde{\mathbf{y}}_{t_K=t}^j)$ for $j = 1, \ldots, M$. Finally we evaluate the contrastive potential as

$$U_t(\mathbf{x}_t) \approx -\log\frac{1}{M}\sum_{j=1}^M \exp\left(\left(\sum_{k=1}^K c(\tilde{\mathbf{y}}_{t_k}^j)(t_k - t_{k-1})\right) - U(\tilde{\mathbf{y}}_t^j)\right)$$

The simulation method for the stochastic process and for the integration $\int_0^t c(\mathbf{y}_s)\mathrm{d}s$ may be altered in this approximation. At this stage, it is unclear what practical implications the weighting term $\int_0^t c(\mathbf{y}_s)\mathrm{d}s$ has. Notice that the process $\mathbf{y}_t$ is initialised at the final simulated position of the forward process $\tilde{\mathbf{x}}_t$. Furthermore, the bias correction with the $w$-stabilisation or an alternative method should still be relevant for stable training of energy-based models.

## C   Latent Space Energy-Based Prior Models

In this section, we first briefly review the latent space energy-based prior models (LEBMs) and its variants: CD-LEBM, SM-LEBM, and ED-LEBM. We then proceed to the experimental details.

### C.1   A Brief Review of LEBMs

Latent space energy-based prior models (Pang et al., 2020) seek to model latent variable models $p_{\phi,\theta}(\mathbf{x}) = \int p_\phi(\mathbf{x}|\mathbf{z})p_\theta(\mathbf{z})\mathrm{d}\mathbf{z}$ with an EBM prior $p_\theta(\mathbf{z}) = \frac{\exp(-E_\theta(\mathbf{z}))p_0(\mathbf{z})}{Z_\theta}$, where $p_0(\mathbf{z})$ is a base distribution which we choose as standard Gaussian (Pang et al., 2020). LEBMs often perform better than latent variable models with a fixed Gaussian prior like VAEs since the EBM prior is more informative and expressive (Pang et al., 2020, Appendix C). However, training LEBMs is more expensive compared to latent variable models with fixed Gaussian prior because of the cost of training for energy-based models. This motivates to explore various training strategies for EBMs such as contrastive divergence, score matching, and the proposed energy discrepancy, where we find that energy discrepancy is the most efficient in terms of computational complexity.

The parameter update for the LEBM can be derived from maximum-likelihood estimation of $p_{\phi,\theta}(\mathbf{x})$. Using the identity $\mathbb{E}_{p_{\phi,\theta}(\mathbf{z}|\mathbf{x})}[\nabla_{\phi,\theta}\log p_{\phi,\theta}(\mathbf{z}|\mathbf{x})] = 0$, the gradient of the log-likelihood of a data point $\mathbf{x}$ is given by

$$\nabla_{\phi,\theta}\log p_{\phi,\theta}(\mathbf{x}) = \mathbb{E}_{p_{\phi,\theta}(\mathbf{z}|\mathbf{x})}[\nabla_{\phi,\theta}\log p_{\phi,\theta}(\mathbf{z},\mathbf{x})] = \mathbb{E}_{p_{\phi,\theta}(\mathbf{z}|\mathbf{x})}[\nabla_\phi\log p_\phi(\mathbf{x}|\mathbf{z}) + \nabla_\theta\log p_\theta(\mathbf{z})].$$

The posterior $p_{\phi,\theta}(\mathbf{z}|\mathbf{x})$ prescribes the latent representation of the data point $\mathbf{x}$. Consequently, in each parameter update, samples are generated from the posterior distribution $p_{\phi,\theta}(\mathbf{z}|\mathbf{x})$ via running Langevin dynamics and are treated as data on latent space. The generator is then updated via

$$\nabla_\phi\log p_{\phi,\theta}(\mathbf{x}) = \mathbb{E}_{p_{\phi,\theta}(\mathbf{z}|\mathbf{x})}[\nabla_\phi\log p_\phi(\mathbf{x}|\mathbf{z})], .$$

Similarly, the maximum-likelihood update for the EBM parameters $\theta$ is given by $\nabla_\theta\log p_{\phi,\theta}(\mathbf{x}) = \mathbb{E}_{p_{\phi,\theta}(\mathbf{z}|\mathbf{x})}[\nabla_\theta\log p_\theta(\mathbf{z})]$. As with any EBM, this gradient can not be used, directly, since this would require a tractable normalisation constant $Z_\theta$. To make this update tractable, we replace the gradient of the log-likelihood with contrastive divergence, score matching, and energy discrepancy as outlined below.

| **Algorithm 1** CD-LEBM | **Algorithm 2** SM-LEBM | **Algorithm 3** ED-LEBM |
|---|---|---|
| 1: sample from posterior and prior
   $\mathbf{z}_+ \sim p(\mathbf{z}\|\mathbf{x})$; $\mathbf{z}_- \sim p_\theta(\mathbf{z})$ | 1: sample from posterior
   $\mathbf{z} \sim p(\mathbf{z}\|\mathbf{x})$ | 1: sample from posterior
   $\mathbf{z} \sim p(\mathbf{z}\|\mathbf{x})$ |
| 2: evaluate the energy difference
   $d_\theta \leftarrow E_\theta(\mathbf{z}_+) - E_\theta(\mathbf{z}_-)$ | 2: evaluate the score difference
   $\boldsymbol{d}_\theta \leftarrow \nabla_\mathbf{z} \log p_\theta(\mathbf{z}) - \nabla_\mathbf{z} \log p(\mathbf{z}\|\mathbf{x})$ | 2: evaluate the energy difference
   $d_\theta \leftarrow \frac{1}{M}\sum_{j=1}^{M} e^{\tilde{E}_\theta(\mathbf{z}) - \tilde{E}_\theta(\mathbf{z} + \sqrt{t}\xi + \sqrt{t}\xi'^j)}$ |
| 3: Update parameter $\theta$ using (13)
   $\theta \leftarrow \theta - \eta_\theta \nabla_\theta d_\theta$ | 3: Update parameter $\theta$ using (14)
   $\theta \leftarrow \theta - \eta_\theta \nabla_\theta \frac{1}{2}\|\boldsymbol{d}_\theta\|_2^2$ | 3: Update parameter $\theta$ using (15)
   $\theta \leftarrow \theta - \eta_\theta \nabla_\theta \log(w/M + d_\theta)$ |

Figure 9: The training procedure for the EBM prior. We use one training sample only to illustrate.

---

**Algorithm 4** Learning latent space energy-based prior models

1: **repeat**
2:    Sample training data points $\{\mathbf{x}^i\}_{i=1}^N \sim p_{\mathrm{data}}(\mathbf{x})$
3:    For each $\mathbf{x}^i$, sample the corresponding latent variable $\mathbf{z}^i \sim p_{\phi,\theta}(\mathbf{z}|\mathbf{x}^i)$ via
      $\mathbf{z}_{k+1}^i = \mathbf{z}_k^i + \frac{\epsilon}{2}\nabla_\mathbf{z} \log p_{\phi,\theta}(\mathbf{z}|\mathbf{x}^i) + \sqrt{\epsilon}\boldsymbol{\omega}_k, \quad \boldsymbol{\omega}_k \sim \mathcal{N}(0, \mathbf{I}), \mathbf{z}_0^i \sim p_0(\mathbf{z})$
4:    Update parameter $\phi$ by maximizing log-likelihood
      $\phi \leftarrow \phi + \eta_\phi \nabla_\phi \frac{1}{N}\sum_{i=1}^N \log p_\phi(\mathbf{z}^i|\mathbf{x}^i)$
5:    Update parameter $\alpha$ by running Algorithms 1, 2, or 3
      $\theta \leftarrow \theta - \eta_\theta \nabla_\theta \mathcal{L}_{\text{CD, SM, or ED}}(\theta)$
6: **until** convergence of parameters $(\phi, \theta)$

---

**CD-LEBM (Pang et al., 2020).** The contrastive divergence update is obtained as per usual by expressing the gradient of the log likelihood in terms of the energy function

$$\nabla_\theta \log p_{\phi,\theta}(\mathbf{x}) = \mathbb{E}_{p_{\phi,\theta}(\mathbf{z}|\mathbf{x})}[\nabla_\theta \log p_\theta(\mathbf{z})] = \mathbb{E}_{p_\theta(\mathbf{z})}[\nabla_\theta E_\theta(\mathbf{z})] - \mathbb{E}_{p_{\phi,\theta}(\mathbf{z}|\mathbf{x})}[\nabla_\theta E_\theta(\mathbf{z})].$$

Therefore, the EBM prior can be learned by minimising

$$\mathcal{L}_{\mathrm{CD}}(\theta) := \frac{1}{N}\sum_{i=1}^N E_\theta(\mathbf{z}_+^i) - E_\theta(\mathbf{z}_-^i), \quad \mathbf{z}_+^i \sim p_{\phi,\theta}(\mathbf{z}|\mathbf{x}^i), \mathbf{z}_-^i \sim p_\theta(\mathbf{z}). \tag{13}$$

Note that optimizing CD-LEBM is computationally expensive, as training the EBM prior requires simulating Langevin dynamics to sample $\mathbf{z}$ from $p_{\phi,\theta}(\mathbf{z}|\mathbf{x})$ to generate positive samples and $p_\theta(\mathbf{z})$ to generate negative samples.

**SM-LEBM.** The second solution is to minimize the Fisher divergence between the posterior and prior, which has the following form

$$\frac{1}{2}\mathbb{E}_{p_{\phi,\theta}(\mathbf{z}|\mathbf{x})}[\|\nabla_\mathbf{z} \log p_\theta(\mathbf{z}) - \nabla_\mathbf{z} \log p_{\phi,\theta}(\mathbf{z}|\mathbf{x})\|_2^2].$$

This is equivalent to score matching (Hyvärinen & Dayan, 2005) when $p_{\phi,\theta}(\mathbf{z}|\mathbf{x})$ is treated as *parameter independent* data distribution. We refer to this approach as score-matching LEBM, in which the EBM prior is learned by minimising

$$\mathcal{L}_{\mathrm{SM}}(\theta) := \frac{1}{N}\sum_{i=1}^N \frac{1}{2}\|\nabla_\mathbf{z} \log p_\theta(\mathbf{z}^i) - \nabla_\mathbf{z} \log p(\mathbf{z}^i|\mathbf{x})\|_2^2, \quad \mathbf{z}^i \sim p_{\phi,\theta}(\mathbf{z}|\mathbf{x}^i). \tag{14}$$

where the parameters of $p_{\phi,\theta}(\mathbf{z}|\mathbf{x})$ are suppressed in the update. Note that score matching generally requires computing the Hessian of the log density as in but in score-matching LEBM, we have $\nabla_\mathbf{z} \log p(\mathbf{z}|\mathbf{x}) = \nabla_\mathbf{z} \log p(\mathbf{x}|\mathbf{z}) + \nabla_\mathbf{z} \log p(\mathbf{z})$.

**ED-LEBM.** Finally, the EBM prior can be learned by minimising the energy discrepancy between the posterior and the EBM prior with $\tilde{E}_\theta(\mathbf{z}) := E_\theta(\mathbf{z}) - \log p_0(\mathbf{z})$, which can be estimated as follows

$$\mathcal{L}_{\mathrm{ED}}(\theta) := \frac{1}{N}\sum_{i=1}^N \log\left(\frac{w}{M} + \frac{1}{M}\sum_{j=1}^M \exp(\tilde{E}_\theta(\mathbf{z}^i) - \tilde{E}_\theta(\mathbf{z}^i + \sqrt{t}\boldsymbol{\xi}^i + \sqrt{t}\boldsymbol{\xi}'^{i,j}))\right) \tag{15}$$

with $\mathbf{z}^i \sim p_{\phi,\theta}(\mathbf{z}|\mathbf{x}^i)$. Note that energy discrepancy does not require simulating MCMC sampling on the EBM prior and calculating the score of the log density, which is computationally friendly for large-scale training. It is critical to include the base distribution $p_0(\mathbf{z})$ in the energy function $\tilde{E}_\theta$. We summarize the training process of the EBM prior using CD-, SM-, and ED-LEBM in Algorithms 1, 2, and 3, with the training procedure of LEBM given in Algorithm 4.

Table 3: Model architectures of LEBMs on various datasets.

(a) Generator for SVHN $32 \times 32$, ngf $= 64$

| Layers | In-Out Size | Stride |
|---|---|---|
| Input: $\mathbf{x}$ | 1x1x100 | - |
| 4x4 convT(ngf x 8), LReLU | 4x4x(ngf x 8) | 1 |
| 4x4 convT(ngf x 4), LReLU | 8x8x(ngf x 4) | 2 |
| 4x4 convT(ngf x 2), LReLU | 16x16x(ngf x 2) | 2 |
| 4x4 convT(3), Tanh | 32x32x3 | 2 |

(b) Generator for CIFAR-10 $32 \times 32$, ngf $= 128$

| Layers | In-Out Size | Stride |
|---|---|---|
| Input: $\mathbf{x}$ | 1x1x128 | - |
| 8x8 convT(ngf x 8), LReLU | 8x8x(ngf x 8) | 1 |
| 4x4 convT(ngf x 4), LReLU | 16x16x(ngf x 4) | 2 |
| 4x4 convT(ngf x 2), LReLU | 32x32x(ngf x 2) | 2 |
| 3x3 convT(3), Tanh | 32x32x3 | 1 |

(c) Generator for CelebA $64 \times 64$, ngf $= 128$

| Layers | In-Out Size | Stride |
|---|---|---|
| Input: $\mathbf{x}$ | 1x1x100 | - |
| 4x4 convT(ngf x 8), LReLU | 4x4x(ngf x 8) | 1 |
| 4x4 convT(ngf x 4), LReLU | 8x8x(ngf x 4) | 2 |
| 4x4 convT(ngf x 2), LReLU | 16x16x(ngf x 2) | 2 |
| 4x4 convT(ngf x 1), LReLU | 32x32x(ngf x 1) | 2 |
| 4x4 convT(3), Tanh | 64x64x3 | 2 |

(d) Generator for MNIST $28 \times 28$, ngf $= 16$

| Layers | In-Out Size | Stride |
|---|---|---|
| Input: $\mathbf{x}$ | 16 | - |
| 4x4 convT(ngf x 8), LReLU | 4x4x(ngf x 8) | 1 |
| 3x3 convT(ngf x 4), LReLU | 7x7x(ngf x 4) | 2 |
| 4x4 convT(ngf x 2), LReLU | 14x14x(ngf x 2) | 2 |
| 4x4 convT(1), Tanh | 28x28x1 | 2 |

(d) EBM prior

| Layers | In-Out Size |
|---|---|
| Input: $\mathbf{z}$ | 16/100/128 |
| Linear, LReLU | 200 |
| Linear, LReLU | 200 |
| Linear | 1 |

## C.2 Langevin Sampling, Reconstruction, and Generation

To sample from the EBM prior $p_\theta(\mathbf{z})$ and posterior $p_{\phi,\theta}(\mathbf{z}|\mathbf{x})$ we employ a standard unadjusted Langevin sampling routine, i.e. we repeat for $k = 0, 1, \ldots, K$

$$\mathbf{z}_{k+1}^i = \mathbf{z}_k^i + \frac{\epsilon}{2}\nabla_\mathbf{z} \log p(\mathbf{z}) + \sqrt{\epsilon}\boldsymbol{\omega}_k, \quad \boldsymbol{\omega}_k \sim \mathcal{N}(0, \mathbf{I})$$

where $\mathbf{z}_0 \sim p_0(\mathbf{z})$ and the distribution $p(\mathbf{z})$ is replaced by the prior or posterior densities, respectively.

The generator is modelled as the Gaussian $p_\phi(\mathbf{x}|\mathbf{z}) = \mathcal{N}(\mu_\phi(z), \sigma^2\mathbf{I})$. In reconstruction of $\mathbf{x}$, we sample from the posterior $\mathbf{z_x} \sim p_{\phi,\theta}(\mathbf{z}|\mathbf{x})$ and compute the reconstruction as $\hat{\mathbf{x}} = \mu_\phi(\mathbf{z_x})$. In data generation, we sample from the EBM prior $\mathbf{z}_{\text{gen}} \sim p_\theta(\mathbf{z})$ and compute the generated synthetic data point as $\mathbf{x}_{\text{gen}} = \mu_\phi(\mathbf{z}_{\text{gen}})$.

## C.3 Experimental Details of LEBMs

**Datasets.** We use the following datasets in image modelling: SVHN (Netzer et al., 2011), CIFAR-10 (Krizhevsky et al., 2009), and CelebA (Liu et al., 2015). SVHN is of resolution $32 \times 32$, and contains $73,257$ training images and $26,032$ test images. CIFAR-10 consists of $50,000$ training images and $10,000$ test images with a resolution of $32 \times 32$. For CelebA, which contains $162,770$ training images and $19,962$ test images, we follow the pre-processing step in (Pang et al., 2020), taking $40,000$ examples of CelebA as training data and resizing it to $64 \times 64$. In anomaly detection, we follow the setting in (Zenati et al., 2018) and the dataset can be found in their published code[5].

**Model Architectures.** We adopt the same network architecture used in CD-LEBM (Pang et al., 2020), with the details depicted in Table 3, where convT($n$) indicates a transposed convolutional operation with $n$ output channels. We use Leaky ReLU as activation functions and the slope is set to be $0.2$ and $0.1$ in the generator and EBM prior, respectively.

**Details of Training and Inference.** Here, we provide a detailed description of the hyperparameters setup for ED-LEBM. Following (Pang et al., 2020), we utilise Xavier normal (Glorot & Bengio, 2010)

---

[5]https://github.com/houssamzenati/Efficient-GAN-Anomaly-Detection

to initialise the parameters. For the posterior sampling during training, we use the Langevin sampler with step size of $0.1$ and run it for $20$ steps for SVHN and CelebA, and $40$ steps on CIFAR-10. We set $t = 0.25, M = 16, w = 1$ throughout the experiments. The proposed models are trained for $200$ epochs using the Adam optimizer (Kingma & Ba, 2015) with a fixed learning $0.0001$ for the generator and $0.00005$ for the EBM prior. We choose the largest batch size from $\{128, 256, 512\}$ such that it can be trained on a single NVIDIA-GeForce-RTX-2080-Ti GPU. In test time, we observed that slightly increasing the number of Langevin sampler steps can improve reconstruction performance. Therefore, we choose $100$ steps with a step size of $0.1$ for posterior sampling. Based on the insights gained from the MCMC diagnostic presented in Figure 21, we choose $500$ steps with a step size of $0.2$ to ensure convergence of the Langevin dynamics when sampling from the EBM prior.

**Evaluation Metrics.** In image modelling, we use FID and MSE to quantitatively evaluate the quality of the generated samples and reconstructed images. On all datasets the FID is computed based on $50,000$ samples and the MSE is computed on the test set. Following (Zenati et al., 2018; Pang et al., 2020), we report the performance using AUPRC in anomaly detection and results are averaged over last 10 epochs to account for variance.

# D Additional Experimental Results

In this section we present additional experimental results to highlight the effect of various parameters of our approach.

## D.1 Experimental Setup for Figure 1 (Healing the nearsightedness of score-matching)

A major problem of score-based methods is their nearsightedness, which refers to their inability to capture global properties of a distribution with disjoint supports such as the mixture weights of two well-separated modes (Zhang et al., 2022). In sight of Theorem 2, energy discrepancy should alleviate this problem as it implicitly compares the scores of both distributions at multiple noise-scales. Following Zhang et al. (2022), we investigate this by computing energy discrepancy as a function of the mixture weight $\rho$ for the mixture of two Gaussians $g_1 := \mathcal{N}(-5, 1)$ and $g_2 := \mathcal{N}(5, 1)$, *i.e.,*

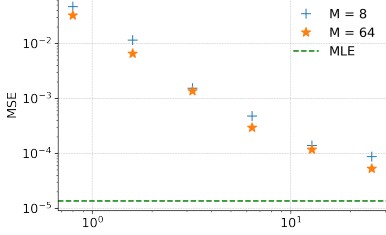

Figure 10: Study of the influence of $t$ and $M$ on estimating mixing weights.

$$p_\rho(x) = \rho g_1(x) + (1 - \rho)g_2(x).$$

where the true data has the mixture weight $\rho = 0.2$. We compare energy discrepancy $\mathcal{L}_{t,M=32,w=1}(\rho) \approx \mathrm{ED}(p_{\rho=0.2}, \log p_\rho)$ with the objective of maximum likelihood estimation $\mathrm{MLE}(\rho) := \mathbb{E}_{p_{\rho=0.2}(x)}[-\log p_\rho(x)]$ and the score matching objective which here is given by the Fisher divergence $\mathrm{SM}(\rho) := \frac{1}{2}\mathbb{E}_{p_{\rho=0.2}(x)}[\|\nabla_x \log p_{\rho=0.2}(x) - \nabla_x p_\rho(x)\|_2^2]$. The losses as functions of $\rho$ are shown in Figure 1. We find that energy discrepancy is convex as a function of the mixture weight and approximates the negative log-likelihood as $t$ increases. Consequently, energy discrepancy can capture the mixture weight well for sufficiently large values of $t$. SM, on the other hand, is a constant function and is blind to the value of the mixture weight.

To further investigate the impact of $t$ and $M$ on the efficiency of energy discrepancy, we minimise the energy discrepancy loss $\mathcal{L}_{t,M=32,w=1}(\rho)$ as a function of the scalar parameter $\rho$ for various choices of $M$ and $t$. We compute the mean-square error of 50 independent estimated mixture weights for choice of $t$ and $M$. As shown in Figure 10, the estimation performance approaches that of the maximum likelihood estimator as $t$ increases, which verifies the statement in Theorem 2. Moreover, if the number of samples $M$ used to estimate the contrastive potential is increased, the estimation performance can be further increased towards the mean-square error of the maximum-likelihood estimator.

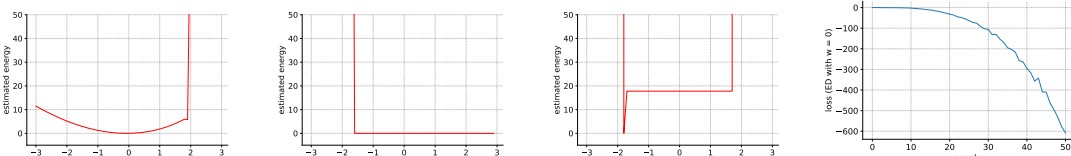

Figure 11: Potential outcomes for the estimated energy and loss history when ED does not converge with $w = 0$.

## D.2 Experimental Setup for Figure 2 (Understanding the $w$-stablisation)

To probe our interpretation of the $w$-stablisation, we train a neural-network to learn the energy function using $4,096$ data points of a one-dimensional standard Gaussian $p_{\text{data}}(x) \propto \exp(-x^2/2)$. The neural network uses an input layer, a hidden linear layer of width two $\mathbb{R}^2 \to \mathbb{R}^2$, and a scalar output layer $\mathbb{R}^2 \to \mathbb{R}$ with a Sigmoid Linear Unit activation between the layers. This neural network has sufficient capacity to model the Gaussian data as well as degenerate energy functions that illustrate potential pitfalls of energy discrepancy for $w = 0$. The energy discrepancy is set up with hyperparameters $M = 4$, $t = 1$, and $w \in \{0, 0.05, 0.25, 2.\}$ and is trained for 50 epochs with Adam. Our results are shown in Figure 2 which confirms the relevance of the $w$-stablisation to obtain a stable optimisation of energy discrepancy. We remark here that the degenerate case $w = 0$ is not strictly reproducible. Different types of lacking smoothness of the energy-function at the edge of the support lead to diverging loss values. We chose a result that illustrates the best the theoretical exposition of the $w$-stablisation in Appendix B.1 and refer to Figure 11 to reflect other malformed estimated energies as well as an example of a diverging loss history.

## D.3 Energy Discrepancy on the Discrete Space $\{0, 1\}^d$

Energy discrepancies are, in principle, well-defined on discrete spaces. To illustrate this point, we describe the energy-discrepancy loss for $\{0, 1\}^d$ valued data such as images with binary pixel values, in which case the discrete energy-discrepancy is straight forward to implement. We will replace the Gaussian transition density with a Bernoulli distribution. For $\varepsilon \in (0, 1)$, let $\boldsymbol{\xi} \sim \text{Bernoulli}(\varepsilon)^d$. Then the transition $\mathbf{y} = \mathbf{x} + \boldsymbol{\xi} \mod(2)$ is symmetric and induces a symmetric transition density $q(\mathbf{y} - \mathbf{x})$. Because of the symmetry, the energy discrepancy can be implemented in the same way as in the continuous case, i.e.

$$\mathcal{L}(\theta) := \frac{1}{N} \sum_{i=1}^{N} \left( w + \sum_{j=1}^{M} \exp\left(E_\theta(\mathbf{x}_i) - E_\theta(\mathbf{x}^i \oplus \boldsymbol{\xi}^i \oplus \boldsymbol{\xi}'^{i,j})\right) \right)$$

with $\boldsymbol{\xi}^i, \boldsymbol{\xi}'^{i,j} \sim \text{Bernoulli}$ and where $\oplus$ is the addition modulo 2. To test the effectiveness of energy discrepancy with Bernoulli perturbation, we test energy discrepancy on various discrete settings. In Figure 12, we estimate the symmetric connectivity matrix of Ising models following the exposition in Grathwohl et al. (2021). We then learn the densities of two-dimensional synthetic data sets that were mapped to the space $\{0, 1\}^{32}$ via grey codes in Figure 13, following Dai et al. (2020). Finally, we estimate the densities of image data with binary pixel values for the static and dynamic MNIST dataset, the omniglot dataset, and the caltech silhouettes dataset. As shown in Figure 14, energy discrepancy performs competitively to contrastive divergence. We propose other potential discrete perturbations in Schröder et al. (2023).



Figure 12: Results of learning Ising models. Left: ground truth; Right: learned pattern.

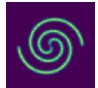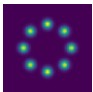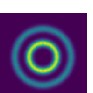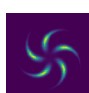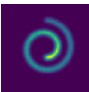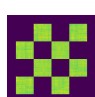

Figure 13: Visualisation of the energy function learned on 2D plan. Here we train a discrete EBM on 32-dim grey codes which are one-to-one mapping from $\mathbb{R}^2$ to $\{0, 1\}^{32}$.

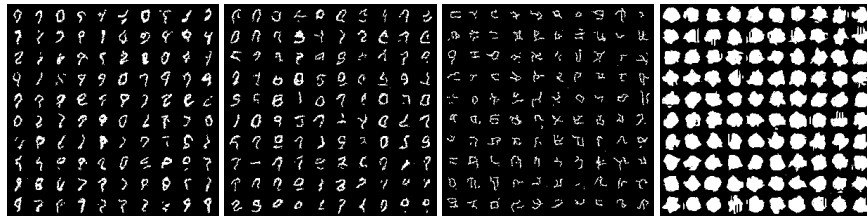

Figure 14: Generated samples on discrete image modelling. Here we train a discrete EBM using the Bernoulli transition. Left to right: Static MNIST, Dynamic MNIST, Omniglot, Caltech Silhouettes.

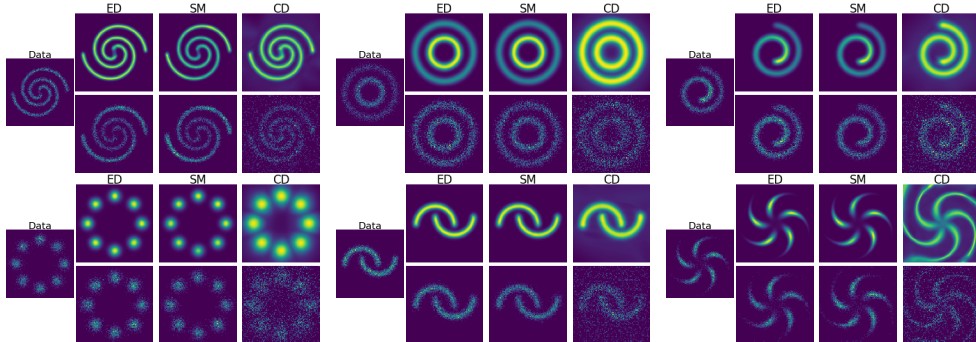

Figure 15: Additional results on density estimation.

### D.4 Additional Density Estimation Results

Here, we provide additional details and results on the density estimation experiments.

**Details of Training and Inference.** Our choice for the energy-net for density estimation is a 4-layer feed-forward neural network with $128$ hidden units and softplus activation function. In the context of energy discrepancy, we select $t = 1$, $M = 4$, and $w = 1$ as hyperparameters. For the contrastive divergence approach, we utilise CD-1, in which the gradient of the log-likelihood in Equation (1) is estimated by employing a Langevin sampler with a single step and a step size of $0.1$. For score matching, we train EBMs using the explicit score matching in (2), where the Laplacian of the score is explicitly computed. We train the model using the Adam optimizer with a learning rate of $0.001$ and iterations of $50,000$. After training, synthesised samples are drawn by simulating Langevin dynamics with $100$ steps and a step size of $0.1$.

**Additional Experimental Results.** The additional results depicted in Figure 15 demonstrate the strong performance of energy discrepancy on various toy datasets, consistently yielding accurate energy landscapes. In contrast, contrastive divergence consistently produces flattened energy landscapes. Despite the success of score matching in these toy examples, score matching struggles to effectively learn distributions with disjoint support which can be seen in the results in Figure 3.

**Comparison with Denoising Score Matching** We further compare energy discrepancy with denoising score matching (DSM) (Vincent, 2011). Specifically, we set $w = 1, M = 4$ and experiment with various $t$. As shown in Figure 16, DSM fails to work when the noise scale is too large or too small. This is because DSM is a biased estimator which is optimised for $p_{\theta^*}(\mathbf{y}) = \int \gamma_t(\mathbf{y} - \mathbf{x}) p_{\text{data}}(\mathbf{x}) d\mathbf{x}$. In contrast, energy discrepancy is more robust to the choice of $t$ since energy discrepancy considers all noise scales up to $t$ simulta-

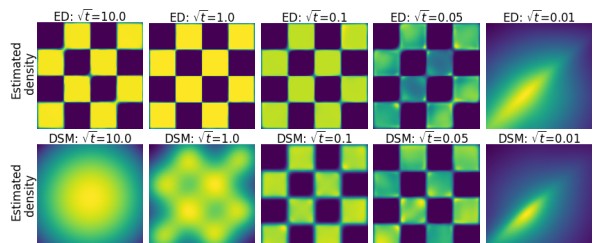

Figure 16: Comparing energy discrepancy (ED) with denoising score matching (DSM) with different noise scales.

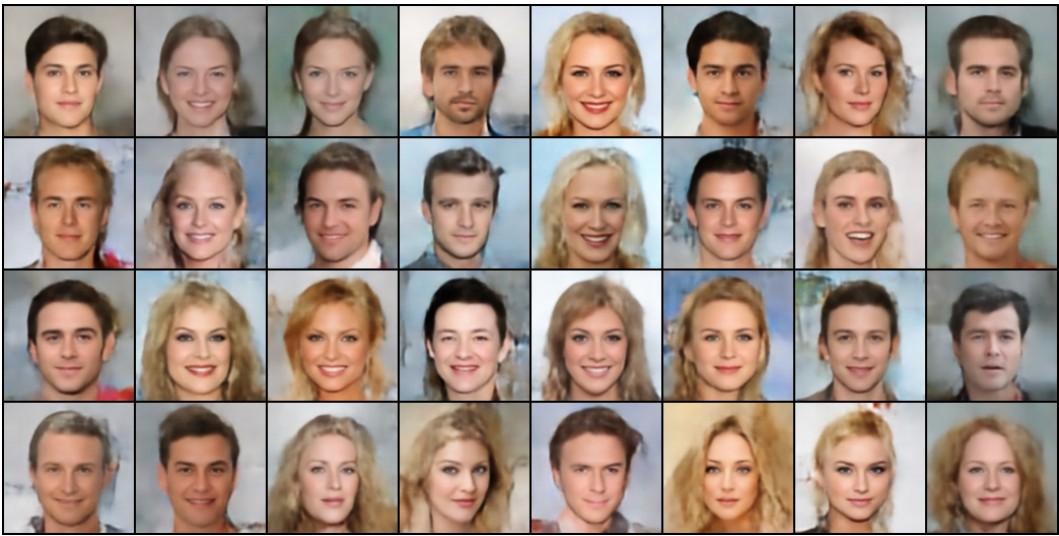

Figure 17: Generated images on CelebA $128 \times 128$.

neously and has an unique optimum $p_{\theta^*}(\mathbf{x}) = p_{\text{data}}(\mathbf{x})$. However, in the case that $t$ is large and $M$ is small, estimation with energy discrepancy deteriorates due to high variance of the estimated loss function. This provides an explanation for the superior performance of energy discrepancy at $\sqrt{t} = 1$ compared to $\sqrt{t} = 10$. Further ablation studies are presented in Figure 23.

### D.5 Additional Image Modelling Results

**Additional Image Generation and Reconstruction Results.** Figures 17 and 18 show additional examples of image generation on CelebA $128 \times 128$ and image reconstruction on CelebA $64 \times 64$. The images are computed through the sampling process outlined in Appendix C.2.

**Additional Image Interpolation and Manipulation Results.** Figures 19, 20 and 22 show additional results of image interpolation and manipulation on CelebA $64 \times 64$. Note that there are two types of interpolations: posterior interpolation and prior interpolation. For posterior interpolation, we consider two real images $\mathbf{x}_1$ and $\mathbf{x}_2$ from the dataset and perform linear interpolation among their corresponding latent variables $\mathbf{z}_1 \sim p_{\phi,\theta}(\mathbf{z}|\mathbf{x}_1)$ and $\mathbf{z}_2 \sim p_{\phi,\theta}(\mathbf{z}|\mathbf{x}_2)$. For prior interpolation, we apply linear interpolation between $\mathbf{z}_1 \sim p_\theta(\mathbf{z})$ and $\mathbf{z}_2 \sim p_\theta(\mathbf{z})$.

**Long-run MCMC Diagnostics.** Figure 21 depicts several convergence diagnostics for long-run MCMC on the EBM prior, where we simulate Langevin dynamics with a large number of steps $(2,000)$. Firstly, the energy profiles converge at approximately 250 steps, as demonstrated in Figure 21a, and the quality of the synthesized samples improves as the number of steps increases. Secondly, we compute the Gelman-Rubin statistic $\hat{R}$ (Gelman & Rubin, 1992) using 64 chains. The histograms of $\hat{R}$ over $5,000 \times 64$ chains are shown in Figure 21b, with a mean of $1.08 < 1.20$, indicating that the Langevin dynamics have approximately converged. Thirdly, we present auto-correlation results in Figure 21c using $5,000$ chains, where the mean is depicted as a line and the standard deviation as bands. The auto-correlation decreases to zero within 200 steps, which is consistent with the Gelman-Rubin statistic that assesses convergence across multiple chains.

### D.6 Qualitative Results on the Effect of $t$, $M$, and $w$

The hyperparameters $t, M, w$ play important roles in energy discrepancy. Here, we provide some qualitative results to understand their effects. According to Theorem 2, $t$ controls the nearsightedness of energy discrepancy. For small $t$, energy discrepancy behaves like score matching $\frac{1}{t}\text{ED}_{\gamma_t}(p_{\text{data}}, U) = \frac{1}{t}\int_0^t \text{SM}(p_s, U_s)\mathrm{d}s \approx \text{SM}(p_{\text{data}}, U)$ and is expected to be unable to resolve local mixture weights. This assertion can be confirmed by qualitative results depicted in Figure 23, which show that when $t = 0.0025$, energy discrepancy fails to identify the weights of components in the 25-Gaussians and pinwheel datasets. For large $t$, energy discrepancy inherits favourable properties

of the maximum likelihood estimator. While large values of $t$ consequently mitigate problems of nearsightedness, it is worth noting that energy discrepancy may encounter issues with high variance when $t$ become excessively large. In such situations, it is necessary to consider increasing the value of $M$ to reduce the variance.

We also investigate the effect of $w$ in Figure 24. As pointed out by the analysis in Appendix B.1, $w$ serves as a stabilises training of energy based models with energy discrepancy. Based on our experimental observations, when $w = 0$ and $M$ is small (*e.g.,* $M \leq 128$ in the 25-Gaussians dataset and $M \leq 32$ in the pinwheel dataset), energy discrepancy exhibits rapid divergence within 100 optimisation steps and fails to converge in the end. If, however, $w$ is increased, e.g. to 1, energy discrepancy shows stable convergence even with $M = 1$. This property is highly appealing as it significantly reduces the computational complexity. Additionally, we find in Figure 2 that larger $w$ tends to result in a flatter estimated energy landscapes which aligns with our intuition gained in Appendix B.1.

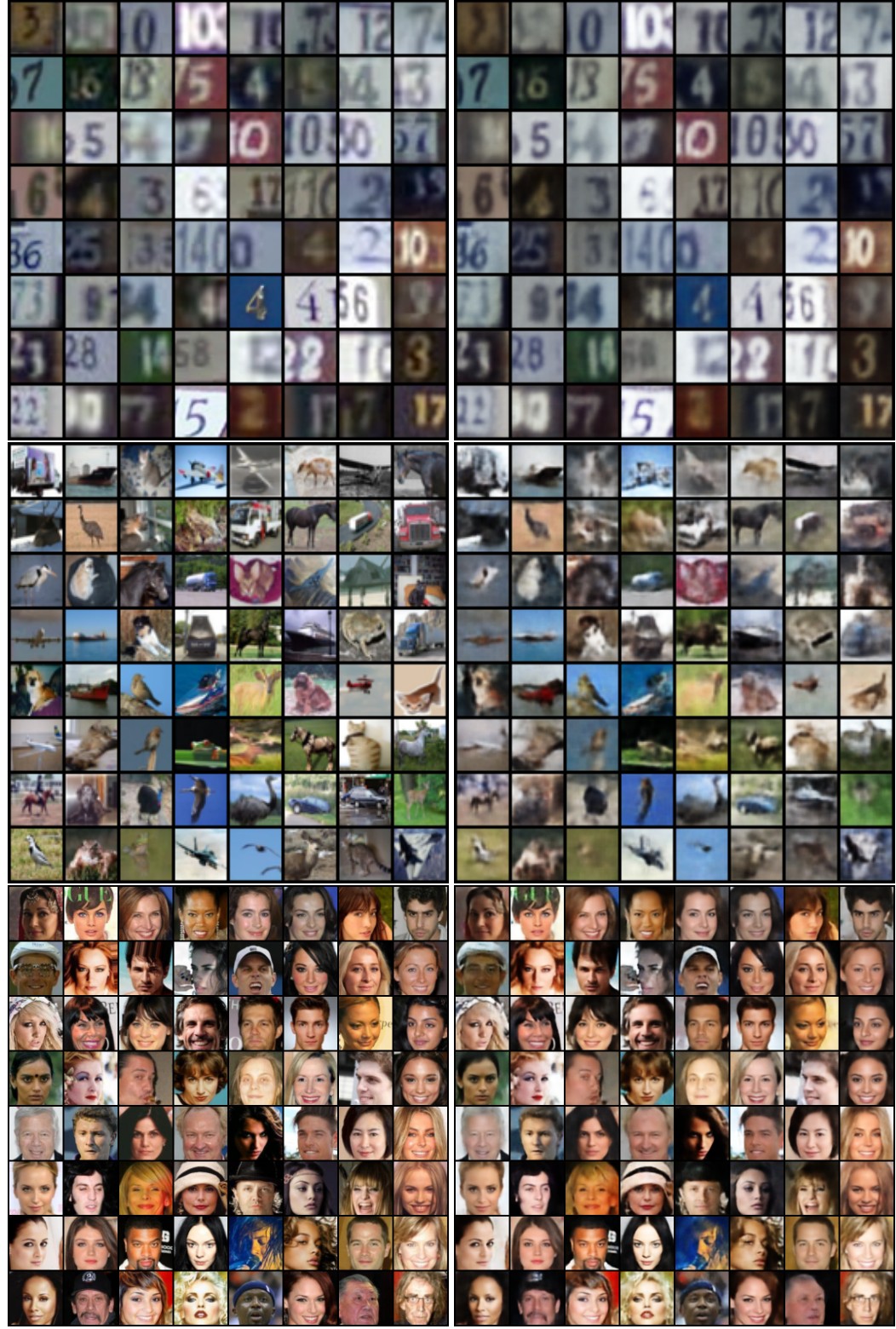

Figure 18: Qualitative results of reconstruction on test images. Left: real image from the dataset. Right: reconstructed images by sampling from the posterior.

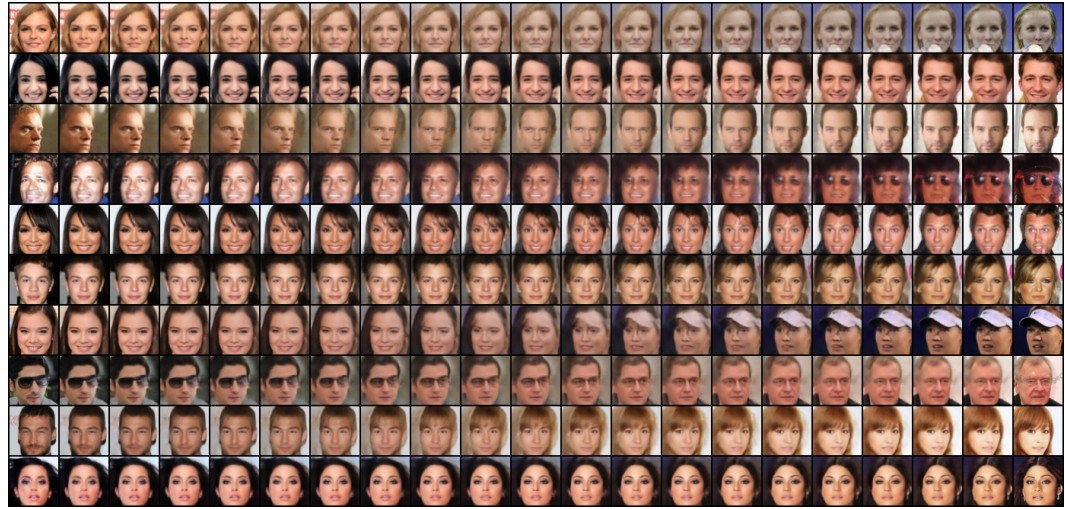

Figure 19: Linear interpolation results in posterior latent space between real images.

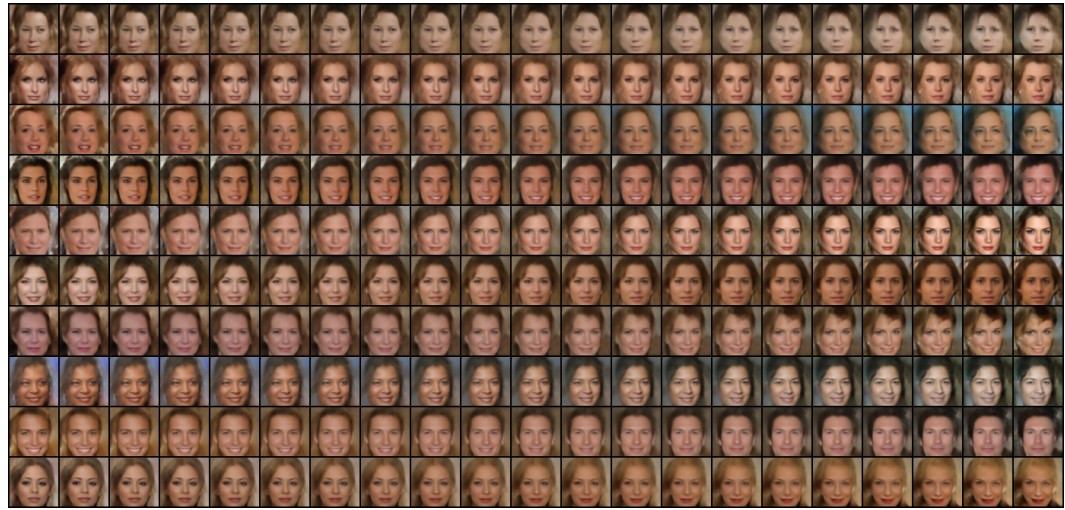

Figure 20: Linear interpolation results in prior latent space between generated images.

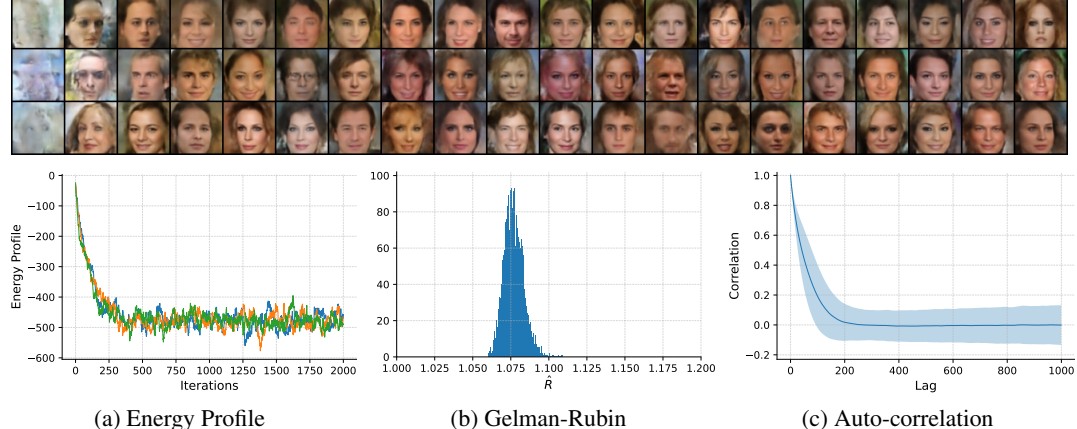

(a) Energy Profile

(b) Gelman-Rubin

(c) Auto-correlation

Figure 21: Diagnostics for the mixing of MCMC chains with $2,000$ steps on CelebA $64 \times 64$. *Top*: Trajectory in the data space. *Bottom*: (a) Energy profile over time; (b) Histograms of Gelman-Rubin statistic of multiple chains; (c) Auto-correlation of a single chain over time lags.

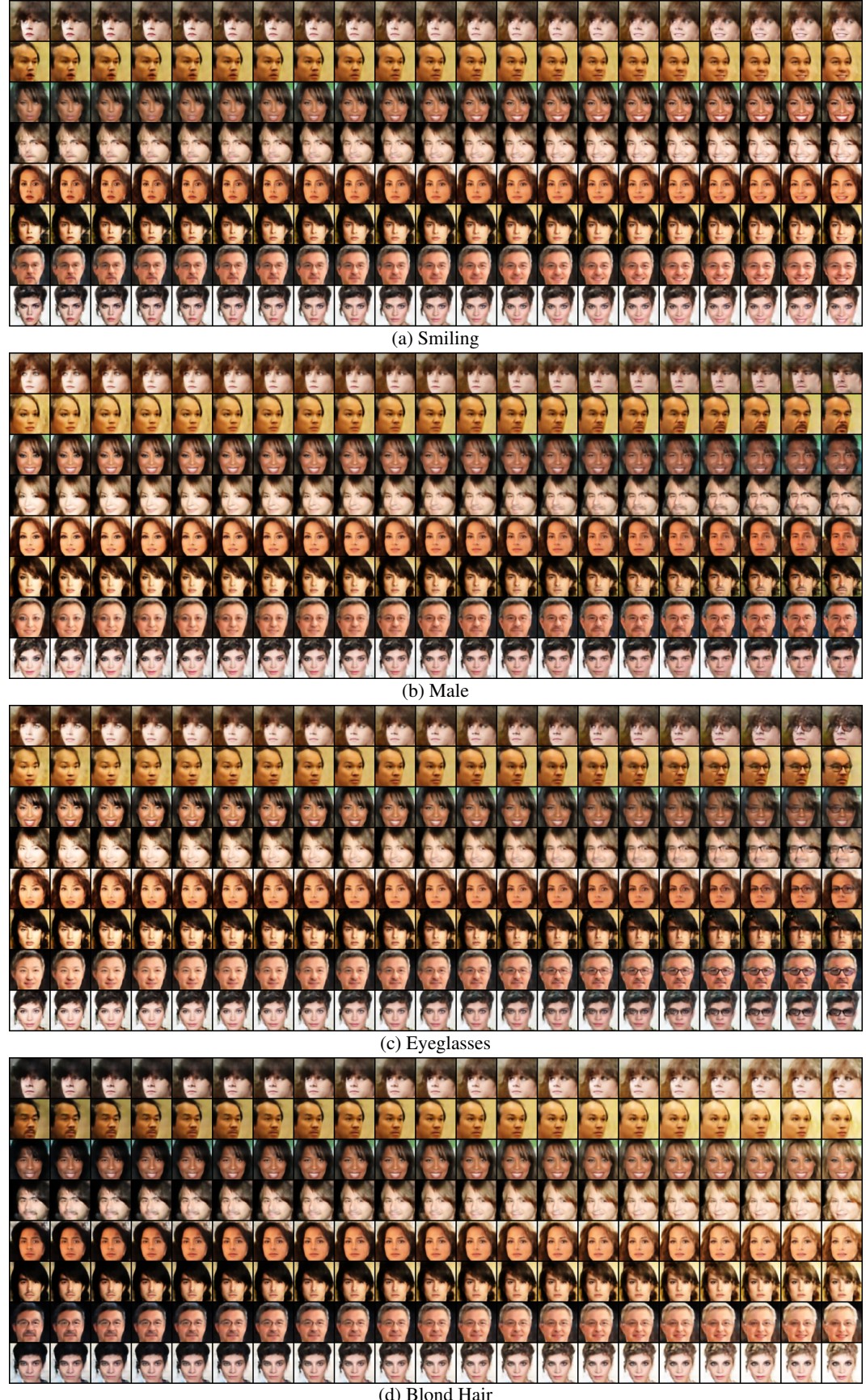

(a) Smiling

(b) Male

(c) Eyeglasses

(d) Blond Hair

Figure 22: Attribute manipulation results on CelebA $64 \times 64$. Each row is made by interpolating the latent variable along an attribute vector, with the middle image being the original image.

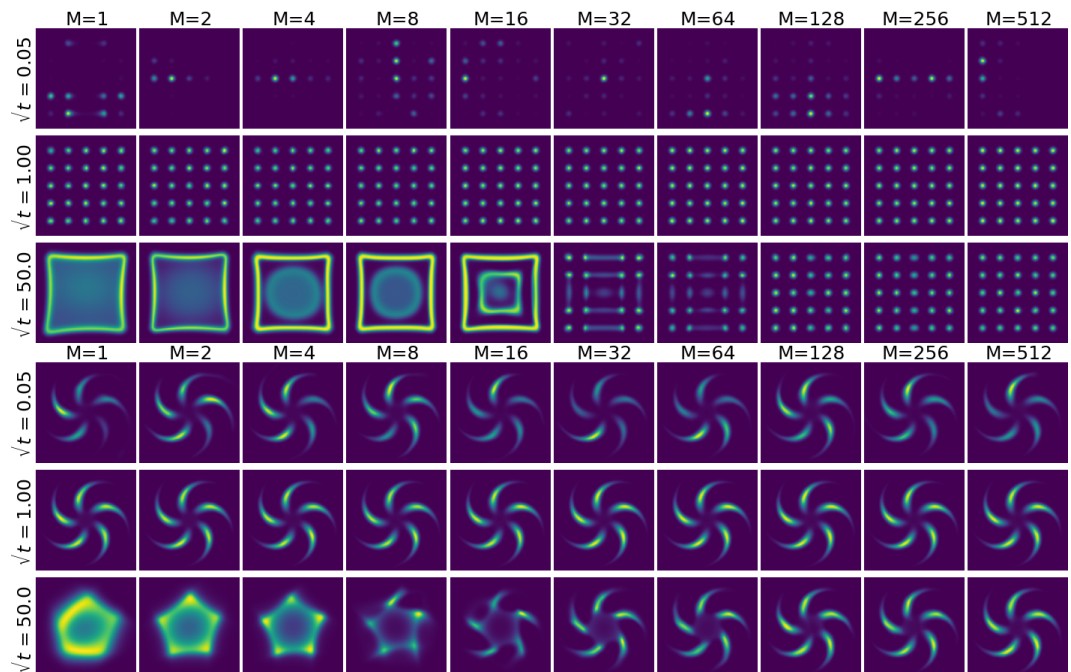

Figure 23: Density estimation on 25-Gausssians and pinwheel with different $t$, $M$ and $w = 1$.

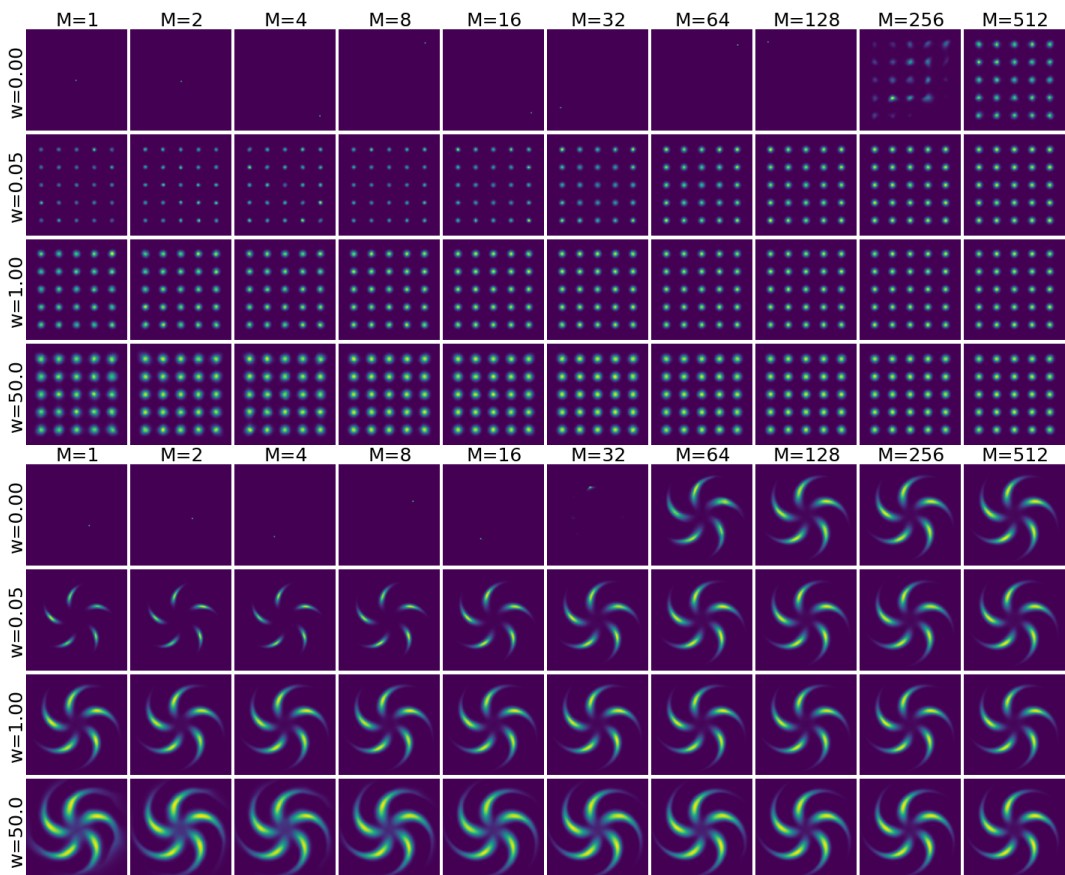

Figure 24: Density estimation 25-Gausssians and pinwheel with different $w$, $M$ and $t = 1$.

