# OpenReview forum: "Energy Discrepancies: A Score-Independent Loss for Energy-Based Models"
_NeurIPS.cc/2023/Conference — NeurIPS 2023 poster_

### Official Review · Reviewer_37sQ · 2023-07-04

**Soundness:** 4 excellent
**Presentation:** 3 good
**Contribution:** 4 excellent
**Rating:** 7
**Confidence:** 3

**Summary:**

Energy discrepancy is presented as a new loss for the training of EBM.
ED interpolates between the losses of score matching and maximum-likelihood estimation.
Efficacy of ED on a latent variable energy-based model is demonstrated to tackle manifold hypothesis as an important challenge in the adoption of likelihood-based training.
Extensive numerical experiments are done to show ED's superiority over contrast divergence and score matching.

**Strengths:**

1. Theoretical derivation is rigorous.
2. Numerical experiments are solid.
3. Paper is in-general well written.
4. The proposed algorithm is easy to implement.


**Weaknesses:**

No obvious weakness to me.

**Questions:**

How important is the w-stabilization term for the numerical experiments?  If w is small(close to 0), are numerical experiments results still hold?

**Limitations:**

No negative social impact is seen.

---

> ### Author Rebuttal · Authors · 2023-08-08
>
> Thank you very much for the positive and helpful comments. Here is our answer regarding the w-stabilisation:
>
> > How important is the w-stabilization term for the numerical experiments? If w is small(close to 0), are numerical experiment results still hold?
> >
>
> **ANSWER:** We found that w-stabilisation is important to make Energy Discrepancy scalable. Initially, Energy Discrepancy was developed without this stabilisation and worked well on two-dimensional data when $M= 256$ contrastive samples were used. With the w-stabilisation, however, we were able to reduce the number of contrastive samples to $M=1$ in two-dimensional settings and learn high-dimensional distributions as in our image modelling experiments using $M=16$. A study on the effect of $w$ and $M$ can be found in Appendix D.5. in Figure 21.
>
> In general, we found that our experimental results are robust to the choice of $w$ and the parameter requires no fine tuning. The choice becomes less consequential the larger $M$ is chosen. Large values for $w$ encourage smoother energy landscapes. Small values of $w$ may underestimate the variance of the distribution if $M$ is chosen too small, but typically stabilise the training sufficiently to produce good results. Since $w$ adds no computational complexity and corrects the approximation bias of the contrastive potential, we found that it is reasonable to choose $w=1$, which appears to work well in all settings we investigated.

---

> > ### Comment · Reviewer_37sQ · 2023-08-18
> >
> > Thanks for the reply. My concern is well-addressed. I will keep my original given score

---

> > > ### Author Response · Authors · 2023-08-19
> > >
> > > We are glad that we could address your concerns. Thank you for reviewing our work!

---

### Official Review · Reviewer_M9yD · 2023-07-05

**Soundness:** 4 excellent
**Presentation:** 3 good
**Contribution:** 4 excellent
**Rating:** 7
**Confidence:** 3

**Summary:**

This paper has proposed a new loss funciton, i.e., Energy Discrepancy to train energy-based models without computaiton of MCMC. The proposed loss function could be directly derived from the energy function without relying on MCMC samples.

**Strengths:**

The proposed energy discrepancy could be directly computed from the energy function and alleviate the problem of nearsightedness of score matching and approximates maximum-likelihood estimation. The experiments show that the energy discrepancy could achieve better performance than score matching and contrastive divergence in image generation.

**Weaknesses:**

I think this is very good paper that proposes a novel optimization method in enery-based models. I only have several questions about the experiment part. Pleare refer to the question section.

**Questions:**

1. Model in [1] utlizes MCMC-based maximum likelihood algorithm to train a normalizing flow model for image generation. Could the proposed energy discrepancy used in this model?
2. According to Table 1, the performance of CD-LEBM and the proposed ED-LEBM are quite closed. The gap is much smaller than the experiments in density estimation.
3. Are there any comparison with GAN-based models?
4. For comparison with score-based methods, [2] shows a much better performance in FID on Cifar10. Could the author give some discussion about these results as the author has claimed that the proposed energy discrepancy has advantages of score matching?

Reference:

[1] Xie, Jianwen, et al. "A Tale of Two Latent Flows: Learning Latent Space Normalizing Flow with Short-run Langevin Flow for Approximate Inference." arXiv preprint arXiv:2301.09300 (2023).

[2] Song, Yang, et al. "Score-based generative modeling through stochastic differential equations." arXiv preprint arXiv:2011.13456 (2020).

**Limitations:**

Please refer to the question section.

---

> ### Author Rebuttal · Authors · 2023-08-09
>
> Thanks for your insightful comments and helpful suggestions! Our answers are listed below.
>
> > Model in [1] utlizes MCMC-based maximum likelihood algorithm to train a normalizing flow model for image generation. Could the proposed energy discrepancy used in this model?
> >
>
> **ANSWER:** Thank you for the interesting reference. The Model in [1] works analogously to the latent EBM [3] used for image data in our work, with the EBM prior being replaced with a normalising flow prior. Since the normalising flow has a tractable likelihood, the maximum likelihood training is possible without training approaches for EBMs like CD, SM, and Energy Discrepancy. However, [1] requires MCMC sampling from the posterior $p(\mathbf z \vert \mathbf x)$ to generate latent representations of data, and this step is necessary in our work as well.
>
> It is an exciting avenue for future research to use a normalising flow as the base distribution of an EBM [4], which would enable training of EBMs with Energy Discrepancy without the need of a latent variable model or MCMC.
>
> > According to Table 1, the performance of CD-LEBM and the proposed ED-LEBM are quite closed. The gap is much smaller than the experiments in density estimation.
> >
>
> **ANSWER:** This is true and we can currently only make assumptions why this is the case. Most likely, the reason lies in the latent variable model used in both methods:
>
> In the density estimation experiments, we observed that CD learns energy landscapes that are biased towards being too smooth. In the image modelling experiments, the latent representations obtained by sampling from $p(\mathbf z\vert \mathbf x)$ may be fairly noisy which corresponds to smooth energy landscapes that can be learned equally well by CD and ED. Flat energy landscapes are also suggested by the interpolation results which show that recognisable images can be produced from midpoints between two latent representations.
>
> Additionally, it is noteworthy that the results are compared in different metrics. In our density estimation results we are comparing the MSE between the estimated density and ground truth density. For the image modeling results, we are comparing the FID of images, which not only measures the quality of the learned EBM prior but also that of the decoder. We do not have access to the true prior to assess the accuracy of the learned EBM and to what extent the decoder compensates for poorly learned energies.
>
> We would like to point out that Energy Discrepancy requires significantly fewer computations per iteration to produce comparable results. Furthermore, we hope that we can improve our experimental results in the future by finding alternatives to the latent EBM model.
>
> > Are there any comparison with GAN-based models?
> >
>
> **ANSWER:** Neither VAEs nor EBMs can currently outperform GAN-based models if it comes to image generation. The appeal of EBMs lies in the fact that they don’t just generate data but also encode data into a probability distribution. An exciting research direction, however, are Generalised EBMs [5] which combine EBMs with GANs and outperform both methods. Exploring new perturbation strategies to train Generalised EBMs with Energy Discrepancy holds promise for future work.
>
> > For comparison with score-based methods, [2] shows a much better performance in FID on Cifar10. Could the author give some discussion about these results as the author has claimed that the proposed energy discrepancy has advantages of score matching?
> >
>
> **ANSWER:** The results in [2] are possible because the score-based method in [2] trains a diffusion model, i.e. the work uses an annealing scheme to learn the scores of data at various noise scales. This achieves two things: Firstly, the annealing scheme makes score matching aware of other modes in the distribution and the scores in low-density areas are estimated accurately. Secondly, the approach learns a sampler as part of the model, which enables a high sample quality. However, score-based diffusion models do not learn an energy-based model or a density. Energy-based models, on the other hand, learn the data generating density but are hard to sample from, resulting in lower FID scores compared to diffusion models.
>
> Training energy-based models with score matching is challenging because the estimated scores are only accurate in the vicinity of modes of the data distribution. [6] elaborates these difficulties of score matching and how they influenced the development of score-based diffusion models ([2, 6]). Energy Discrepancy alleviates some of these difficulties by implicitly incorporating data information at different noise scales since the Gaussian Energy Discrepancy is equivalent to a multi-noise-scale score matching loss.
>
> [1] Xie, Jianwen, et al. "A Tale of Two Latent Flows: Learning Latent Space Normalizing Flow with Short-run Langevin Flow for Approximate Inference." arXiv preprint arXiv:2301.09300 (2023).
>
> [2] Song, Yang, et al. "Score-based generative modeling through stochastic differential equations." arXiv preprint arXiv:2011.13456 (2020)
>
> [3] Pang, Bo et al. "Learning Latent Space Energy-Based Prior Model." NeurIPS, 2020
>
> [4] Nijkamp, Erik, et al. "MCMC Should Mix: Learning Energy-Based Model with Neural Transport Latent Space MCMC" ICLR, 2022.
>
> [5] Arbel, Michael, Liang Zhou, and Arthur Gretton: Generalized energy based models, ICLR 2021.
>
> [6] Song, Yang and Ermon, Stefano: Generative Modeling by Estimating Gradients of the Data Distribution, NeurIPS 2019

---

> > ### Comment · Reviewer_M9yD · 2023-08-19
> >
> > Thanks for the authors' reply. My question is well-answered and I will keep my rating.

---

### Official Review · Reviewer_NEwe · 2023-07-07

**Soundness:** 3 good
**Presentation:** 3 good
**Contribution:** 3 good
**Rating:** 7
**Confidence:** 2

**Summary:**

This paper proposes a new loss function for training energy-based models, called energy discrepancy (ED). ED does not rely on score functions and MCMC samples. Instead, it is defined as the difference between the energy function of data and some conditional samples. They prove that optimizing this objective function yields the appropriate energy function. Then, they build connections between ED, score matching and contrastive divergence. Finally, this paper focuses on training the latent energy-based prior model, which is a VAE with EBM prior.

For experiments, the authors first showcase the density estimation on several 2D pdfs. ED outperforms SM and CD on pdf estimation and sampling. Then, they train latent EBMs on SVHN, CIFAR-10 and CelebA. ED also outperforms SM and CD on image reconstruction/generating and out-of-distribution detection.


**Strengths:**

1. Training EBMs is an important problem in generative modeling. This paper proposes a new training criteria, which does not rely on possibly ill-conditioned score functions and time-consuming MCMC samples.

2. The authors also build connections between the proposed method and score-matching estimates or MLEs.


**Weaknesses:**

1. The authors do not train EBMs directly on the data space. Instead, they train a VAE with a EBM prior. Since CD or SM can lead to competitive EBMs, I am wondering if there is any empirical result on training EBMs directly on the data space using the proposed loss?

2. Since the proposed loss function is similar to the conditional NCE, I think one promising baseline could be training the same model using CNCE loss.

3. The authors said ED provides fast training (in Line 338). Is there any comparison about the running time?


**Questions:**

1. In the Equation between Line 187 and 188, is there $E_\theta(x^i)$ missing behind $w / M$?

---

> ### Author Rebuttal · Authors · 2023-08-09
>
> Thank you very much for your constructive feedback! Here are our responses to the mentioned weaknesses and questions:
>
> **Weaknesses:**
>
> > The authors do not train EBMs directly on the data space. Instead, they train a VAE with a EBM prior. Since CD or SM can lead to competitive EBMs, I am wondering if there is any empirical result on training EBMs directly on the data space using the proposed loss?
> >
>
> **ANSWER:** In our density estimation experiments, the energy-based models are trained on data space, and Energy Discrepancy outperforms CD or SM in this domain. Furthermore, we have new empirical results for discrete EBMs trained directly on pixel space for various binary image data sets (See Figure 3 in the attached PDF).
>
> The VAE approach is used in our image modelling experiments, only. The so-called manifold hypothesis suggests that most image data lives on a lower-dimensional subset of the higher dimensional pixel space. As a consequence of this, the energy function outside of the manifold is undefined, i.e. for a noisy data point, we have $p_\mathrm{data}(\tilde{\mathbf x}^i)=0$ which implies for the learned energy that $E_\theta(\tilde{\mathbf x}^i) \approx -\log p_\mathrm{data}(\tilde{\mathbf x}^i) \to \infty$. For this reason, we use a latent variable model to construct representations of the data that are supported in the whole latent space. The probability of a noisy latent representation is still positive $p_z(\tilde{\mathbf{z}}^i)>0$ and we can learn a well-defined energy.
>
> Contrastive divergence is not as sensitive to this problem because it only pulls up low-energy states, and the value of the energy function stays bounded. However, this comes at the cost of biased energy landscapes as shown in Figure 3, which pose difficulties in likelihood-based tasks like anomaly detection as shown in Table 2. Furthermore, sampling from the learned energies remains challenging.
>
> Score-based generative models normally require the use of multiple noise scales [1, 2] to alleviate issues with its nearsightedness and scores being undefined outside the data manifold. Training EBMs with score matching directly on data space without annealing faces challenges similar to the ones outlined for Energy Discrepancy with the additional difficulty of SMs nearsightedness.
>
> > Since the proposed loss function is similar to the conditional NCE, I think one promising baseline could be training the same model using CNCE loss.
> >
>
> **ANSWER:** This is a great suggestion. In fact, in the special case of Gaussian perturbations with one contrastive sample ($M= 1$ in ED, $\kappa = 1$ in CNCE [3]), and $w= 1$ , the two approaches coincide.
>
> To further explore CNCE as a baseline, we conducted experiments on the 25-gaussian dataset for different choices of $M$ ($\kappa$ in CNCE) and different choices for $t$ ($0.5\epsilon^2$ in CNCE). Our results (see Figure 6 in the attached pdf) show that the performance of CNCE is comparable to Energy Discrepancy. On image data, we expect that CNCE performs similarly to Energy Discrepancy and that many of the techniques we use for ED like the latent EBM would also improve the experimental results for CNCE. We believe that both, CNCE as well as our proposed loss function are promising approaches for fast and accurate training of EBMs.
>
> Conceptually, Energy Discrepancy has the appeal of being closely connected to the original maximum likelihood estimation problem, which has the lowest asymptotic variance amongst all unbiased estimators. For this reason, Energy Discrepancy may be more data efficient if sufficient compute is available. Additionally, CNCE requires tractable likelihoods of the perturbation, while Energy Discrepancy only requires being able to simulate the perturbation. Thus, ED is applicable to different non-Gaussian perturbations in which the likelihood is intractable.
>
> > The authors said ED provides fast training (in Line 338). Is there any comparison about the running time?
> >
>
> **ANSWER:** The statement is based on complexity analysis. Energy Discrepancy requires $\mathcal O(M)$ evaluations of the energy net, while SM and CD methods require at least $\mathcal O(d)$ evaluations of the energy net to compute the gradient, where $d$ denotes the dimension.
>
> For the density estimation experiment, one step of the optimiser for ED, CD, and SM takes 0.006s, 0.018s, and 0.021s, respectively. Additionally, ED converges more quickly as shown in Figure 4 in our paper. The resulting run time can be compared in Figure 5 in the attached pdf. In the image modelling experiments, ED reduces the computational cost significantly as $M=16$ is used throughout all image modelling experiments irrespective of the dimension.
>
> **Questions:**
>
> > In the Equation between Line 187 and 188, is there $E_\theta(x^i)$ missing behind $w/M$?
> >
>
> **ANSWER:** In this equation, the energy term behind the $w/M$ cancels with the non-contrastive energy term, i.e. for each $\mathbf x^i$ we write the energy as $E_\theta(\mathbf x^i) = \log \exp(E_\theta(\mathbf x^i))$ and use $\log(a)+\log(b) = \log(ab)$ to take the logarithm outside of all other operations. This gives for each sample in the batch:
>
> $E_\theta(\mathbf{x}^i) + \log\left(\frac{w}{M}\exp(-E_\theta(\mathbf{x}^i)) + \frac{1}{M}\sum_{j=1}^M \exp(-E_\theta(\tilde{\mathbf{x}}^{i, j}) \right) = \log\left(\frac{w}{M} + \frac{1}{M}\sum_{j=1}^M \exp(E_\theta(\mathbf x^i)-E_\theta(\tilde{\mathbf{x}}^{i, j})\right)$
>
> We will add the identity to the appendix and hope this makes things clearer.
>
> [1] Song, Yang and Ermon, Stefano: Generative Modeling by Estimating Gradients of the Data Distribution, NeurIPS 2019
>
> [2] Zengyi Li, Yubei Chen, Friedrich T. Sommer: Learning Energy-Based Models in High-Dimensional Spaces with Multi-scale Denoising Score Matching, 2019
>
> [3] Ceylan, Ciwan, and Michael U. Gutmann. "Conditional noise-contrastive estimation of unnormalised models." *ICML*, 2018.

---

> > ### Comment · Reviewer_NEwe · 2023-08-17
> >
> > Thanks for the authors' responses, which address my issues.
> > I would like to raise my score to 7.

---

> > > ### Author Response · Authors · 2023-08-17
> > >
> > > We are delighted to hear that we could address your concerns and your decision to increase your score! Thank you for your review of our work.

---

### Official Review · Reviewer_BJCK · 2023-07-20

**Soundness:** 3 good
**Presentation:** 3 good
**Contribution:** 3 good
**Rating:** 6
**Confidence:** 4

**Summary:**

The authors propose a new loss function for training energy-based models, which they dub "energy discrepancy."  The aim is to provide a viable alternative to contrastive divergence and score matching based methods that suffer from near-sightedness -- these approaches lack global information and can have difficulties fitting well-separated Gaussians.  The energy discrepancy method seeks to overcome this difficulty by perturbing the distributions to increase the mass of the low probability regions separating the peaks of the distribution.  This is done at different noise levels and then integrated.  The proposed loss is theoretically justified and then validated experimentally in both synthetic and real-world settings.

**Strengths:**

The approach is well-motivated for the most part and the presentation is mostly clear.  The idea is novel to me and experimentally improves over existing approaches.

**Weaknesses:**

I'm not certain that the approach is potentially as practical as it appears.  My main concern is that the w-stabilisation procedure seems to be doing a lot of heavy lifting and that is tailored to the Gaussian case.  Given other discussion in this work, I don't think this is a huge problem.  However, the case for acceptance would be bolstered if a broader set of applications were considered.

**Questions:**

-  There is a bit of magic in the w-stabilisation procedure.  I read the explanation in Appendix B, but I feel that more explanation is needed in the main text.  In particular, does this approach generalize beyond the Gaussian case?  This makes me think that the proposed approach is really quite specific (at least if you want to be able to do it in practice).    Does this need to be included as a limitation?

-  Is it possible to include some discrete setting experiments as well?

Minor typos:

- "normalisation of EBMs, also known as partition function" sounds funny.  Maybe "normalisation constant"?
- The sentence in lines 30-33 doesn't really make sense.
- "i.e." and "e.g." must always be followed by a comma.
- "line out in" -> "outline in"?
- Lots of "which" usage that I think requires a comma.



**Limitations:**

Yes, but see above comments.

---

> ### Author Rebuttal · Authors · 2023-08-08
>
> Thanks a lot for your valuable comments. Your suggestions are very helpful in further improving the work, and we are refining the manuscript accordingly.
>
> > My main concern is that the w-stabilisation procedure seems to be doing a lot of heavy lifting and that is tailored to the Gaussian case. Given other discussion in this work, I don't think this is a huge problem. However, the case for acceptance would be bolstered if a broader set of applications were considered.
> >
>
> **ANSWER:** Thank you for raising this concern. The $w$-stabilisation is critical to scale up our experiments to high dimensions and significantly reduce the computational cost. The stabilisation is, however, not restricted to the Gaussian case. To support our case, we have added experiments using a Bernoulli perturbation in various discrete settings. We observe a similar effect of the $w$-stabilisation in discrete spaces (see Figure 4 in the attached pdf).
>
> Additionally, it is possible to avoid stabilisation in certain use cases. For example, it is feasible to estimate densities using a large value for $M$ such as $M=512$ in low-dimensional data (see e.g. figure 21 in the appendix). The stabilisation is also not needed if the energy function is sufficiently regular. This typically applies for models that are not deep architectures and that have interpretable parameters, such as products of Gaussian experts with finite variance.
>
> > There is a bit of magic in the w-stabilisation procedure. I read the explanation in Appendix B, but I feel that more explanation is needed in the main text. In particular, does this approach generalize beyond the Gaussian case? This makes me think that the proposed approach is really quite specific (at least if you want to be able to do it in practice). Does this need to be included as a limitation?
> >
>
> **ANSWER:** Thanks for the comment. We will extend the main text to make the w-stabilisation more intuitive. The w-stabilisation can be generalised beyond the Gaussian case and we choose the same stabilisation successfully in the discrete setting which uses a Bernoulli perturbation. We included a study on the effect of w for the Bernoulli perturbation in Figure 4 on the attached PDF. The main idea behind the stabilisation is the following:
>
> In theory, the difference between the positive energy term and the contrastive energy term is bounded, and thus Energy Discrepancy has an existing minimum at $\exp(-U^\ast)\propto p_\mathrm{data}$. In practice, Energy Discrepancy requires approximating the contrastive energy term with samples, which we call “contrastive samples”.  At the edge of the data support, it can happen at random that all contrastive samples have high energy. The resulting parameter gradient encourages energy landscapes for which the energies of contrastive samples go to infinity. We visualise this in Figure 2 in our paper. The w-stabilisation effectively adds the unperturbed data point to the set of contrastive samples. If all contrastive samples have unusually high energies, the log-sum-exp operation is dominated by the energy of the unperturbed data point which does not blow up because it is minimised in the non-contrastive term at the same time. This heuristic argument applies to all perturbations.
>
> We agree that further theoretical work on the effect of the w-stabilisation and its optimal value would be of great interest and it is conceivable that different stabilisation procedures could improve the effectiveness of our method. However, at this stage of the work, the w-stabilisation has several big advantages:
>
> - It is simple, adds no computational complexity to the optimisation, and there is numerical evidence for its effectiveness
> - The $w$-parameter requires little tuning, and experimental results are of similar quality for all values that are of the order of 1.
> - It is theoretically justified by Theorem 3, which we have generalised to other perturbations in the meantime
>
> We hope we could address your concerns and are happy to discuss further.
>
> > Is it possible to include some discrete setting experiments as well?
> >
>
> **ANSWER:** We have included new experiments on discrete data using the Bernoulli perturbation in the attached PDF, following the setting in [1]. The experiments are conducted on predicting the connectivity matrix of an Ising model (Figure 1), learning the density of 2D data mapped via grey codes to the discrete space $\\{0, 1\\}^{32}$ (Figure 2), as well as binary MNIST, Polyglot, and Silhouettes data sets (Figure 3). It is noteworthy that there were no latent variable models used in the discrete image modelling while achieving competitive results. We leave it as future work to find suitable perturbations for other structured data like graphs and text.
>
> [1] Dinghuai Zhang et al.: Generative Flow Networks for Discrete Probabilistic Modeling, ICML 2022

---

### Official Review · Reviewer_BMfu · 2023-07-21

**Soundness:** 3 good
**Presentation:** 3 good
**Contribution:** 3 good
**Rating:** 6
**Confidence:** 2

**Summary:**

In this paper, the authors consider the problem of training energy-based models. Due to the limitations of two current approaches known as approximate maximum likelihood methods (which might lead to malformed estimators of the energy function) and score-based methods (which fail to resolve global features in the distribution without big data), the authors propose a novel loss function for this task named Energy Discrepancy (ED) to overcome those issues with rigorous theoretical guarantee. Moreover, they also conduct several experiments to demonstrate the efficacy of the ED loss function in learning low-dimensional data distributions compared to two previous methods. However, the authors point out that their approach does not work for high-dimensional data due to some manifold hypothesis.

**Strengths:**

1. Originality: Energy Discrepancy (ED) is a novel loss for training enerygy-based models.

2. Quality: Theoretical results are solid and associated with rigorous proof, but I do not spend time double-checking all of them. Additionally, a lot of experiments are carried out to empirically justify the effectiveness of the proposed loss function ED. This significantly strengthens the contributions of the paper.

3. Clarity: The paper is well written and organized, which makes it easy to follow.

4. Significance: Although the ED loss function work for only low-dimensional data distributions, this work lays the basis for alleviating the issues of maximum likelihood methods and score-based methods for training energy-based models.

**Weaknesses:**

1. Clarity:  There are some places that the authors introduce results without necessary intuitions. For instance, in line 96, they directly suggest using Gaussian kernels without explanation.

**Questions:**

1. In equation (2), why is the score-matching loss defined between $p_{\text{data}}$ and $E_{\theta}$ rather than between $p_{\text{data}}$ and $p_{\theta}$?

2. In the beginning of section 3, can we use other kernels rather than Gaussian kernels to perturb $p_{\text{data}}$ and $p_{\theta}$?

3. In equation (5), is the energy discrepancy $ED_q$ a proper metric?

4. In Theorem 2, the authors should either briefly introduce the definition of Wasserstein distance or cite relevant papers to that distance so that readers from different communities can understand.


**Limitations:**

The limitations are discussed in Section 7 of the paper.

---

> ### Author Rebuttal · Authors · 2023-08-08
>
> Thank you very much for your constructive comments. Here is our response to your questions:
>
> > There are some places that the authors introduce results without necessary intuitions. For instance, in line 96, they directly suggest using Gaussian kernels without explanation.
> >
>
> **ANSWER:** Thank you for making us aware of this. The intuition for Gaussian kernels is two-fold: Firstly, Gaussian kernels are the simplest choice as they are easy to sample from and allow easier calculations for the analysis of the method. They are often used to apply noise in score-based methods [1], contrastive methods [2], and as spread noise [3]. Consequently, this is the first perturbation we tried. Secondly, we observed empirically that Gaussian perturbations work well in practice. They allow us to make the connection to score matching which we describe in our motivation in equation (3) as well as maximum likelihood estimation which characterises Gaussian perturbations as optimal among a large class of Markov transition kernels. We will include an additional sentence to motivate our choice of perturbation in line 96 and leave other choices for follow-up work.
>
> > In equation (2), why is the score-matching loss defined between $p_{\mathrm{data}}$ and $E_\theta$ rather than between $p_\mathrm{data}$ and $p_\theta$?
> >
>
> **ANSWER:** Thank you for pointing out that this notation is unclear. This notation was chosen for the following reasons:
>
> 1. Unlike the Fisher divergence from which score matching is derived, the score-matching objective in equation (2) is not a statistical divergence between $p_\mathrm{data}$ and  $p_\mathrm{ebm}$, but an estimation criterion for the energy function. For this reason, we treat score-matching as a functional of the learned energy function $U$ or $E_\theta$, respectively and make this explicit in the notation.
> 2. We also reuse the notation in equation (3) where we find this notation more concise than the alternatives.
> 3. Finally, this notation is consistent with our notation for Energy Discrepancy, which we treat as a functional of the energy function in the same way.
>
> We are going to edit the sentence that precedes the equation to make the chosen notation more logical.
>
> > In the beginning of section 3, can we use other kernels rather than Gaussian kernels to perturb $p_\mathrm{data}$ and $p_\theta$?
> >
>
> **ANSWER:** The proposed Energy Discrepancy can indeed be defined for other kernel functions. The connection to score matching made at the beginning of section 3, however, holds only for Markov transition kernels associated with SDEs of the form $\mathrm d\mathbf x_t = \mathbf a(\mathbf x_t)\mathrm dt + \mathrm d\mathbf w_t$.
>
> In practice, Gaussian kernels are favourable due to easy sampling and many analytical tools available, so we conducted our analysis and experiments for this choice of kernel, first. Additionally, Theorem 2 shows that Gaussian kernels are optimal among possible Markov transition kernels in the sense that the maximum likelihood objective can be approximated with a Gaussian kernel of sufficiently large variance.
>
> One example of a non-Gaussian kernel for Energy Discrepancy is the Bernoulli perturbation that can be applied in discrete spaces (see section B.3 in the appendix). The additional experimental results in the attached PDF (Figure 1-4) demonstrate that this perturbation is effective in training EBMs in various discrete settings. We will explore other kernels in future work.
>
> > In equation (5), is the energy discrepancy a proper metric?
> >
>
> **ANSWER:** Energy Discrepancy is not a metric but is designed as a criterion for density estimation, similar to maximum likelihood estimation. This means that for Energy Discrepancy, $p_\mathrm{data}$ is always fixed while $p_\theta$ is learned and the role of the two arguments can not be interchanged. In particular, ED is not symmetric. When Energy Discrepancy is minimised in $p_\theta$, the minimum is attained at the ground truth $p_\theta = p_\mathrm{data}$.
>
> While not a metric, Energy Discrepancy can be related to a statistical divergence called KL contraction, which is a weaker notion of distance. (see section A.6 in the appendix).
>
> > In Theorem 2, the authors should either briefly introduce the definition of Wasserstein distance or cite relevant papers to that distance so that readers from different communities can understand.
> >
>
> **ANSWER:** Thank you for pointing this out, we will refer to [4] in our revision.
>
> [1] Song, Yang and Ermon, Stefano: Generative Modeling by Estimating Gradients of the Data Distribution, NeurIPS 2019
>
> [2] Michael Gutmann and Aapo Hyvärinen: Noise-contrastive estimation: A new estimation principle for unnormalized statistical models, AISTATS 2010
>
> [3] Mingtian Zhang, Peter Hayes, Tom Bird, Raza Habib, David Barber: Spread Divergence, ICML 2020
>
> [4] Peyre, Gabrial and Cuturi, Marco: Computational Optimal Transport, Foundations and Trends in Machine Learning 2019

---

> > ### Comment · Reviewer_BMfu · 2023-08-17
> >
> > Dear authors,
> >
> > Thanks for your thorough rebuttal. The authors has already addressed my concerns toward the paper. Therefore, I will maintain my score of 6 subject to the fact that these changes will be included in the revision of this paper.

---

> > > ### Author Response · Authors · 2023-08-17
> > >
> > > Dear Reviewer,
> > >
> > > We are glad to hear that we could address your concerns! The changes will be included in the next revision. Thank you for your assessment of our work.

---

### Author Rebuttal · Authors · 2023-08-09

We thank all reviewers for their constructive and extensive comments that help us to improve this work.

We would first like to summarise the paper according to the reviewers:

- Our work proposes a new practical, easy to implement, and fast training technique for energy-based models that does not require MCMC sampling. Reviews agree that this approach is novel:

    > **BMfu:** “Energy Discrepancy (ED) is a novel loss for training enerygy-based models.”; **BJCK:** “The idea is novel to me and experimentally improves over existing approaches.”; **NEwe:** “This paper proposes a new training criteria, which does not rely on possibly ill-conditioned score functions and time-consuming MCMC samples.”; **M9yD:** “I think this is very good paper that proposes a novel optimization method in enery-based models.”; **37sQ:** “The proposed algorithm is easy to implement.”
    >

    Furthermore, the reviewers agree that the contribution is significant:

    > **BMfu:** “Although the ED loss function work for only low-dimensional data distributions, this work lays the basis for alleviating the issues of maximum likelihood methods and score-based methods for training energy-based models.”; **M9yD:** “The experiments show that the energy discrepancy could achieve better performance than score matching and contrastive divergence in image generation.”
    >
- Our work introduces theoretical guarantees demonstrating the validity of our approach. Furthermore, we draw connections to score matching and maximum likelihood estimation. The reviews reflect the soundness of our approach as follows:

    > **NEwe:** “The authors also build connections between the proposed method and score-matching estimates or MLEs.”; **37sQ:** “ED interpolates between the losses of score matching and maximum-likelihood estimation. Theoretical derivation is rigorous.”
    >

We now summarise the concerns that were raised most frequently by reviewers and explain how we addressed them in our rebuttal:

- **Experiments on discrete spaces**

    Reviewer ****BJCK**** is interested in additional experiments on discrete spaces. We added three new experiments to the attached pdf regarding different discrete settings: Using an Energy Discrepancy based on a Bernoulli perturbation, we learn the connectivity matrix of an Ising model, two dimensional densities that are mapped into $\\{0, 1\\}^{32}$ via grey codes, as well as various image data sets with binary pixel values (MNIST, Polyglot, Silhouettes). Our results are competitive at a low computational cost.

    Reviewer ****NEwe**** is interested in experiments on data space, directly. We would like to refer to the new experiments on discrete data (Figure 1, 2, 3 on the attached PDF) to demonstrate that Energy Discrepancy is capable of learning high-dimensional distributions on pixel space, directly. However, we also discussed why we think that for contrastive learning methods like ours, latent variables or other types of hybrid models are necessary to model many image data sets due to the manifold hypothesis.

- **Concerns regarding the w-stabilisation:**

    Reviewer ****BJCK**** is concerned that the w-stabilisation only applies to the Gaussian case and that more motivation is needed in the main text. To respond to this concern, we are going to add a new paragraph to the main paper that motivates the w-stabilisation and how it applies to **any** type of perturbation. The main idea of the stabilisation is to control the variance of a log-sum-exp operation in the sample approximated loss functional. To support our case that this stabilisation applies to other perturbations, we have added ablation studies (see Figure 4 in the attached PDF) comparing empirical results with and without $w$ in the case of a Bernoulli perturbation.

    Reviewer ****37sQ**** is interested if Energy Discrepancy works if the stabilisation term goes to zero. Here, we would like to refer to the appendix of our paper. Figure 21 shows a comparison of learned energies for various choices of $w$ and $M$. One can see that Energy Discrepancy works for all values of $w$ if the number of perturbed samples $M$ is large. Introducing even a very small stabilisation $w$ reduces the number of required samples drastically.

---

### Decision · Program_Chairs · 2023-09-21

**Decision:**

Accept (poster)

**Comment:**

This paper proposes a new loss function for learning energy-based models that can be evaluated without computing scores or running long MCMC chains. Experiments show that the proposed method is promising. The reviewers all recognize the contribution as novel and substantial. All reviewers vote in favor of accepting this work and indicate that their concerns have been sufficiently addressed during author discussion. I think the proposed method is an interesting addition to the generative modeling toolbox and I am happy to support accepting this work.